# ORAK: A FOUNDATIONAL BENCHMARK FOR TRAINING AND EVALUATING LLM AGENTS ON DIVERSE VIDEO GAMES

**Dongmin Park**[1][†][*] **Minkyu Kim**[1][†][*] **Beongjun Choi**[1][†][*] **Junhyuck Kim**[1][†], **Keon Lee**[1][†], **Jonghyun Lee**[1][†], **Inkyu Park**[1][†], **Byeong-Uk Lee**[1][†], **Jaeyoung Hwang**[1][†], **Jaewoo Ahn**[1,2][†], **Ameya S. Mahabaleshwarkar**[3], **Bilal Kartal**[3], **Pritam Biswas**[3], **Yoshi Suhara**[3], **Kangwook Lee**[1,4,5], **Jaewoong Cho**[1]

[1]KRAFTON, [2]Seoul National University, [3]NVIDIA, [4]University of Wisconsin-Madison, [5]Ludo Robotics

## ABSTRACT

Large Language Model (LLM) agents are reshaping the game industry, by enabling more intelligent and human-preferable characters. Yet, current game benchmarks fall short of practical needs: they lack evaluations of diverse LLM capabilities across various game genres, studies of agentic modules crucial for complex gameplay, and fine-tuning datasets to adapt pre-trained LLMs into gaming agents. To fill these gaps, we present **Orak**, a benchmark for training and evaluating LLM agents across 12 popular video games spanning all major genres. Using a plug-and-play interface built on Model Context Protocol (MCP), Orak supports systematic and reproducible studies of agentic modules in varied game scenarios. We further release a fine-tuning dataset of expert LLM gameplay trajectories covering multiple genres, turning general LLMs into effective game agents. Orak offers a united evaluation framework, including game leaderboards, LLM battle arenas, and ablation studies of input modality, agentic strategies, and fine-tuning effects, establishing a foundation towards versatile gaming agents. Code and datasets are available at https://github.com/krafton-ai/Orak and https://huggingface.co/datasets/KRAFTON/Orak.

## 1 INTRODUCTION

Large Language Model (LLM) agents are revolutionizing various industries (Wang et al., 2024a), and well-established benchmarks play a key role in unlocking their abilities on complex tasks, *e.g.*, Coding (Hendrycks et al., 2021; Zhuo et al., 2024), Web Search (Liu et al., 2023; Pan et al., 2024), and Scientific Research (Mühlbacher et al., 2024; Zhang et al., 2025). In the game industry, there is growing interest in using LLM agents to enhance user game experiences, *e.g.*, more intelligent non-player characters (NPCs), monsters, or companions (NVIDIA, 2025). Also, games that simulate dynamic and uncertain real-world scenarios can serve as effective test-beds for evaluating agents' high-level decision-making, *i.e.*, system 2 reasoning (Li et al., 2025). In response, many benchmarks have been proposed to assess the ability of LLMs by playing games (Hu et al., 2024b; Ahn et al., 2025).

While these benchmarks have effectively utilized games to evaluate LLMs' general capabilities, they exhibit three major limitations: 1) they often rely on *text-only* games or *2D-grid* simulators rather than complex real video games, 2) they offer insufficient assessment of *agentic* modules, such as self-reflection, memory, and tool use, which are essential to complex gameplay, and 3) they lack *fine-tuning* datasets necessary to adapt pre-trained LLMs into effective gameplay agents, which significantly hinder the adoption of LLM agents in real-world video games.

To this end, we present **Orak**, a foundational benchmark designed to evaluate LLM agents across diverse video games. As shown in Figure 1, Orak includes 12 video games played by millions to billions of users worldwide: *Street Fighter III*, *Super Mario*, *Ace Attorney*, *Her Story*, *Pokémon Red*, *Darkest Dungeon*, *Minecraft*, *Stardew Valley*, *StarCraft II*, *Slay the Spire*, *Baba Is You*, and *2048*.

---

[*]Equal contribution. [†]Core contribution.

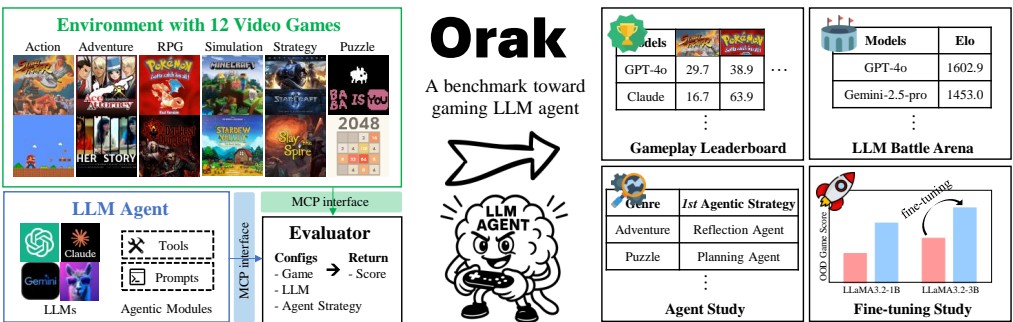

Figure 1: Overview of Orak, a benchmark designed to train and evaluate LLM agents across 12 video games across genres. Using MCP as a plug-and-play interface, it ensures systematic assessment, supporting gameplay leaderboards, battle arenas, and studies on agentic modules and fine-tuning.

These games span 6 major game genres, *i.e.*, action, adventure, role-playing, simulation, strategy, and puzzle, enabling a *comprehensive* assessment of key abilities required for versatile gameplay; action games enable testing fine-grained player control, adventure games challenge long-term memory and error handling, and strategy/puzzle games require complex logical reasoning and multi-step planning. With the use of *real* video games, Orak also ensures evaluation on rich, dynamic environments with varying stages, levels, and story-driven quests, which are even challenging for humans.

To enable consistent evaluation of rapidly evolving LLMs, we introduce a *plug-and-play* interface using Model Context Protocol (MCP) (Hou et al., 2025), allowing LLMs to seamlessly interact with agentic modules in gameplay. Each game environment and agentic module package operates as an independent MCP server, providing game mechanics (*e.g.*, retrieving game states, executing game steps) or agentic strategies (*e.g.*, reflection, planning) as callable tools to LLMs. During evaluation, LLM interacts with these servers by sequentially retrieving game states, performing action inference using agentic modules, and executing game steps, which enables streamlined evaluation across diverse games and supports controlled studies of various agentic modules.

In addition, we release a fine-tuning dataset that aims to transform pre-trained LLMs into effective gaming agents. The dataset consists of game interaction trajectories generated by expert LLMs, *e.g.,* GPT-4o, using core agentic strategies on all games in Orak. These trajectories encapsulate meta-knowledge on how and when to use the agentic strategies to play various genres of games, leading to more resource-efficient and effective gaming agents.

Our benchmark offers comprehensive evaluation dimensions, including game score leaderboards, competitive LLM battle arenas, and ablation studies of visual input state, agentic strategies, and fine-tuning effects. Extensive experiments on Orak with 15 LLMs reveal that (1) proprietary LLMs achieve superior performance across games with significant gaps from open-source LLMs, (2) their performance gap becomes narrow in battle scenarios, (3) proprietary LLMs benefit from extended agentic workflows, while open-source LLMs show limited gains, (4) models fail to extract sufficient value from visual inputs yet, and (5) fine-tuning enables effective transfer of gameplay meta-knowledge from larger LLMs to smaller ones, leading to generalization in intra-game, out-of-distribution (OOD) game, and non-game unseen scenarios like math and web interaction. We believe that Orak not only establishes a foundation for building gaming agents but also serves as a critical benchmark for evaluating general LLMs on realistic, long-horizon decision-making tasks.

## 2 RELATED WORK

**Playing Games by LLMs.** Many works have explored the use of LLMs for gameplay. Early efforts focused on *text-based* games such as Jericho (Hausknecht et al., 2020), Zork (Tsai et al., 2023), and TextCraft (Prasad et al., 2023), where LLMs navigate textual environments with reasoning. Subsequent work shifted toward *2D-grid* games, including Chess (Feng et al., 2023), NetHack (Küttler et al., 2020), and Crafter (Hafner, 2021), where spatial reasoning and puzzle-solving skills became more important. More recently, several studies have applied LLMs, combined with *agentic* workflows, to play more complex *video* games, such as Minecraft (Fan et al., 2022; Wang et al., 2023), Civilization (Qi et al., 2024), Pokémon (Hu et al., 2024c), and StarCraft (Ma et al., 2024). However,

| Benchmarks | Game Domain | Full Genre | # Games | Model Type | Agent Ablation | Fine-tuning Set |
|---|---|---|---|---|---|---|
| GAMA-bench (Huang et al., 2024) | Text | ✗ | 8 | LLM | ✗ | ✗ |
| GameBench (Costarelli et al., 2024) | Text | ✗ | 9 | LLM | ✗ | ✗ |
| GameArena (Hu et al., 2024a) | Text | ✗ | 6 | LLM | ✗ | ✗ |
| SmartPlay (Wu et al., 2023) | Text/2D-grid | ✗ | 6 | LLM | ✗ | ✗ |
| Balrog (Paglieri et al., 2024) | Text/2D-grid | ✗ | 6 | LLM/VLM | ✗ | ✗ |
| LVLM-Playground (Wang et al., 2025a) | 2D-grid | ✗ | 6 | VLM | ✗ | ✗ |
| Cradle (Tan et al., 2024) | Video | ✗ | 4 | VLM | ✗ | ✗ |
| V-MAGE (Zheng et al., 2025) | Video | ✗ | 5 | VLM | ✗ | ✗ |
| DSGBench (Tang et al., 2025) | Video | ✗ | 6 | LLM | ✗ | ✗ |
| LMGame-Bench (Hu et al., 2025) | Video | ✗ | 6 | LLM/VLM | ✗ | ✗ |
| **Orak (Ours)** | Video | ✓ | 12 | LLM/VLM | ✓ | ✓ |

Table 1: Game Benchmark Comparison. 'Full Genre' means whether six major genres are fully covered. 'Model Type' indicates whether the benchmark supports LLMs or VLMs. Unlike prior benchmarks, Orak is the *only* benchmark that fully covers all major genres, supports both LLMs/VLMs, provides ablation studies for agent modules, and releases a fine-tuning set.

these approaches rely on manually customized agentic workflows for each specific game, limiting their usability toward developing a general gaming agent.

**Evaluation Benchmarks for LLMs with Games.** As gameplay requires complex cognitive abilities, *e.g.*, context understanding, logical reasoning, and error handling, several recent benchmarks have sought to evaluate LLMs or Vision Language Models (VLMs) on games (Hu et al., 2024b). GAMA-Bench (Huang et al., 2024), GameBench (Costarelli et al., 2024), GameArena (Hu et al., 2024a), and SmartPlay (Wu et al., 2023) focus on *text-based* games, assessing LLMs' ability to navigate and reason on textual environments. Barlog (Paglieri et al., 2024) and LVLM-Playground (Wang et al., 2025a) are mainly based on *2D-grid* games, such as TicTacToe and Chess, to evaluate the spatial and visual reasoning capabilities of LLMs/VLMs. Using video games, Cradle (Tan et al., 2024) evaluates VLMs in 1 adventure and 3 simulation games, V-MAGE (Zheng et al., 2025) assesses VLMs on 5 action games, DSGBench (Tang et al., 2025) validates LLMs on 6 strategic games, and LMGame-Bench (Hu et al., 2025) mainly focus on puzzle games. Despite their contributions, existing benchmarks lack full coverage of game genres, omit in-depth studies on agentic modules, often rely on visual inputs, and under-explore the way to align pre-trained LLMs into versatile game agents. Table 1 summarizes the key characteristics of game benchmarks.

**Fine-tuning LLMs toward Agents.** To enhance agent capability, many efforts have proposed agentic strategies, e.g., chain-of-thought reasoning (Wei et al., 2022; Yao et al., 2023), self-reflection (Shinn et al., 2023; Park et al., 2023), hierarchical task planning (Huang et al., 2022b;a), and skill libraries (Wang et al., 2023). Complementing these advancements, efforts have focused on fine-tuning strategies tailored to LLM agents, which span two main directions: data-centric approaches, which involve fine-tuning on curated expert demonstrations (Chen et al., 2023; Zeng et al., 2023; Chen et al., 2024); and framework-oriented approaches, which focus on learning from agentic interactions (Feng et al., 2024; Chen et al., 2025; Putta et al., 2024). Notably, FireAct (Chen et al., 2023) emphasizes unified data formatting and CodeAct (Wang et al., 2024b) highlights the importance of high-quality curation of training trajectories. However, fine-tuning methods for game-playing agents remain largely explored. Unlike structured tasks in web, programming, or math domains, games involve large, dynamic, and partially observable state spaces, which requires agents to generalize across a variety of situations and learn diverse behavior patterns, posing unique challenges.

## 3 ORAK

We propose Orak, a benchmark designed to evaluate LLM agents across diverse video games. Figure 2 illustrates the evaluation pipeline of Orak, which is self-explanatory with code structures. By integrating the MCP interface with game environments and agentic modules, Orak enables *systematic* and *plug-and-play* evaluation of backbone LLMs with agentic strategies across various games. For evaluation, the game score is obtained by simply configuring the game, LLM backbone, and agentic strategy in `eval.py`. At each game step, the game observation is retrieved, the specified agent strategy is executed by the LLM, and the resulting action is applied to the game. This loop continues until the game ends or reaches the maximum step limit, after which the game score is recorded. Note that, with MCP interface, users can readily customize their agentic strategy, *i.e.*, calling a single

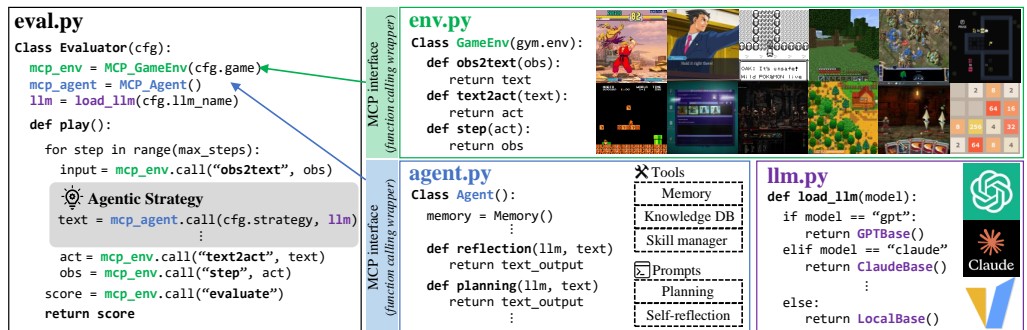

Figure 2: Evaluation pipeline of Orak. Game scores are computed via `eval.py` by simply configuring game, LLM backbone, and agentic strategy. Orak supports two types of submissions: (1) customizing `llm.py` with new backbone LLMs, and (2) customizing `agent.py` with new agentic strategies. The agentic strategies are callable by LLMs via MCP interface in `eval.py` (in grey box).

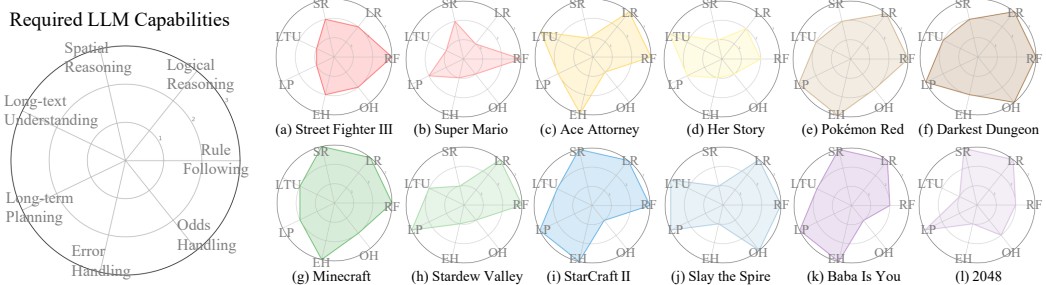

Figure 3: LLM capabilities required to play 12 games in Orak. The color theme (red, yellow, etc) represents game genres. See Appendix A for genre categorization details.

agentic module or multiple agentic modules sequentially. In the following section, we describe the 12 game environments in Orak, along with the submission guidelines for the benchmark leaderboard.

## 3.1 GAME ENVIRONMENTS

**LLM Capabilities Required.** LLM agents require diverse capabilities to play games. Figure 3 summarizes the capability levels needed for each game in Orak, measured using principled criteria adopted from the context of game design (Wu et al., 2023; Koster, 2013):

- **Rule Following (RF).** The level is measured by the extent to which adherence to rules is required for gameplay (1: single rule, 2: fewer than 5 rules, 3: 5 or more rules).
- **Logical Reasoning (LR).** The number of LLM's reasoning hops required to determine an in-game action (1: 1 hop, 2: 1 to 3 hops, 3: 3 or more hops).
- **Spatial Reasoning (SR).** The level of spatial understanding required for gameplay (1: not necessary, 2: required in specific situations, 3: critical to core gameplay).
- **Long-text Understanding (LTU).** The extent of long-context comprehension required for gameplay (1: a few lines, 2: a few paragraphs, 3: longer than one page with 500+ words).
- **Long-term Planning (LP).** The extent to which strategic planning is required (1: not necessary, 2: planning for up to 3 sequential actions, 3: essential to plan more than 3 sequential actions).
- **Error Handling (EH).** The extent to which error correction is required during gameplay (1: not necessary, 2: requires a one-step rollback, 3: requires multi-step rollback and re-planning).
- **Odds Handling (OH).** The extent to which understanding randomness is required for gameplay (1: not necessary, 2: randomness exists in game, 3: randomness is critical to core gameplay).

The capability level is measured on a scale of 1 to 3 by 8 human participants, and the moderate value is reported. Since most video games are designed to require various cognitive abilities even for humans, they tend to require high levels of LLM capabilities in many dimensions. For example, action games, *Street Fighter III* and *Super Mario* in red, require spatial reasoning and rule following, more than long-context understanding and planning, while adventure games, *Ace Attorney* and *Her Story* in yellow, emphasize long-text understanding and logical reasoning due to the need to comprehend long storylines. More detailed analysis of required LLM capabilities for each game is in Appendix B.

**Game Description.** For each game, we provide a brief description of (1) the game state, (2) the action space given to LLMs, and (3) the evaluation task and metric. More detailed explanations of each environment are elaborated in Appendices C-N.

**(a) Street Fighter III** (Capcom, 1997) is a 2D side-scrolling fighting action game with 20 unique characters, each with distinctive skills. (1) *Game state*: The player character, opponent character, remaining time, each player's health, super-bar, stun-bar gauges, and the distance between the two characters. (2) *Action space*: 15-20 discrete actions: 'move closer', 'low punch', 'high kick', etc. (3) *Evaluation task*: Beating the game bot; performance is measured by the number of stages cleared.

**(b) Super Mario** (Christian Kauten, 2018) is a side-scrolling game where the player controls Mario to avoid obstacles, defeat enemies, and reach the flag. (1) *Game state*: The positions (x,y) and sizes of obstacles and enemies extracted from the current game state. (2) *Action space*: Mario keeps moving to the right, and LLM decides the jump level, discretized in 6 bins. (3) *Evaluation task*: Reaching out to the final flag; performance is measured by the horizontal distance that Mario travels before dying.

**(c) Ace Attorney** (Capcom, 2001) is a courtroom adventure game where players act as defense attorneys, gathering evidence and witnesses to prove their client's innocence. (1) *Game state*: Dialogue history, collected evidence, court records profiles, etc. (2) *Action space*: Player's courtroom actions: advancing dialogue, accessing court records, pressing witnesses, and presenting evidence. (3) *Evaluation task*: Performance is measured by response correctness and total steps taken.

**(d) Her Story** (Barlow, 2015) is an interactive adventure game where players explore police interview clips to uncover a hidden truth. (1) *Game state*: History of queries and search results with metadata for the first 5 clips (visual description, date, viewing status, and transcript if played). (2) *Action space*: Searching for clips with keywords, or selecting a video to play. (3) *Evaluation task*: Uncover the truth; Performance is measured by the number of distinct video clips viewed to complete the game.

**(e) Pokémon Red** (Game Freak, 1996) is a role-playing game where a player explores, collects Pokémon, and battles other trainers to progress the storyline. (1) *Game state*: Player's location, party Pokémon (species, level, HP, status), inventory, battle state, and screen text. (2) *Action space*: Choosing high-level tools or low-level joypad actions. (3) *Evaluation task*: Defeat Brock, the first gym leader; Progress measured by how many of 12 predefined storyline flags are triggered.

**(f) Darkest Dungeon** (Red Hook Studios, 2016) is a role-playing game where heroes explore dungeons while managing stress and resources. (1) *Game state*: Party status (character stats, health, stress, and status effects), available skills, and enemy encounters. (2) *Action space*: Combat actions like 'attack', 'heal', and 'swap'. (3) *Evaluation task*: Complete the first expedition; Performance is measured by the sum of the successful combats, the survived heroes, and their remaining stress.

**(g) Minecraft** (Mojang Studios, 2011) is an open-ended sandbox game where players explore a world, gather resources, and survive by placing and breaking blocks. (1) *Game state*: The player's position, inventory, health status, the nearby blocks and biome, etc. (2) *Action space*: Executable JavaScript code within the Mineflayer environment (contributors, 2013). (3) *Evaluation task*: Crafting a target item; performance is measured by whether the item is collected in the inventory.

**(h) Stardew Valley** (ConcernedApe, 2016) is an open-ended life simulation game where players farm, fish, mine, and explore. (1) *Game state*: Player's location, energy, inventory, crop status, soil status, date, and weather. (2) *Action space*: Leaving the house, entering the house, sleeping, buying seeds, tilling soil, watering, harvesting, and selling crops. (3) *Evaluation task*: Earning the most money by harvesting crops within the first 13 in-game days; performance is measured by total profit.

**(i) StarCraft II** (Blizzard Entertainment, 2010) is a real-time strategy game where players gather resources, construct buildings, train units, and command armies to defeat opponents. (1) *Game state*: Resource levels, unit/building counts, production queues, research progress, and observed enemy info. (2) *Action space*: 72 discrete actions, including unit training, building, research, and strategic operations. (3) *Evaluation task*: Beating built-in AI bots; performance is measured by the win rate.

**(j) Slay the Spire** (MegaCrit, 2017) is a deck-building roguelike game where players ascend a multi-floor tower, battling enemies and building decks. (1) *Game state*: Player's class, deck, hand, health, relics, energy, enemies' intents and statuses, and current floor. (2) *Action space*: Playing a card during combat, ending the turn, and selecting a card reward after combat. (3) *Evaluation task*: Defeating the final boss at the top floor; performance is measured by the number of floors reached.

**(k) Baba Is You** (Hempuli, 2019) is a puzzle game where a player manipulates the rules by moving word tiles on a board. (1) *Game state*: Coordinates of text and object tiles, and active rules. (2)

*Action space*: A single movement 'up', 'down', 'left', and 'right', or a sequence of such moves. (3) *Evaluation task*: Solving the first stage; if the stage is not cleared, partial credit is awarded based on sub-goals (*e.g.*, breaking the 'Wall Is Stop' rule).

**(l) 2048** (Cirulli, 2014) is a sliding tile puzzle game that aims to combine numbered tiles on a 4×4 grid board to create a tile of the value 2048. (1) *Game state*: The current configuration of the 4×4 grid, where each cell contains either a number (power of 2) or is empty. (2) *Action space*: Four discrete actions; 'up', 'down', 'left', and 'right'. (3) *Evaluation task*: Creating the 2048 tile; performance is measured by the normalized progress toward creating the 2048 tile.

## 3.2 SUBMISSION GUIDELINE TO BENCHMARK LEADERBOARD

**Models.** Participants can submit new pre-trained LLMs, VLMs, or their fine-tuned versions. Unless otherwise specified, all models will be evaluated under the default agentic strategies for each game specified in Section 5.1.

**Agentic Strategies.** Participants can also submit new agentic modules and strategies. All submitted strategies should follow a consistent structure across games, *e.g.*, by maintaining a fixed sequence such as reasoning, planning, and using a specific tool.

## 4 FINE-TUNING: ALIGNING PRE-TRAINED LLMS INTO GAME AGENTS

| | Data example from Reflection module | | Data example from Action module |
|---|---|---|---|
| $X^{\text{ref}}$ | Analyze Mario's past action using state difference and provide critiques for improving his action. You should only respond in the format as below: ### Self-reflection (Describe self-reflection here) | $X^{\text{act}}$ | (Retrieve the self-reflection from memory) Analyze the current game state, and decide the best action. You should only respond in the format as below: ### Action Jump Level: n (where n is an integer from 0 to 6) |
| $S$ | ### Past Game State Mario at (100,100), Bricks at (120, 100), (120, 150) ### Current Game State Mario at (100,100), Bricks at (120, 100), (120, 150) | $S$ | ### Game State Mario at (100,100) Position of all objects: - Bricks at (120, 100), (120, 150) . . . |
| $Y^{\text{ref}}$ | ### Self-reflection Mario is blocked by bricks. Jump higher to get past. | $Y^{\text{act}}$ | ### Action Jump Level: 6. |

Table 2: Fine-tuning data examples when playing *Supermario* with 'reflection' agent.

We collect the fine-tuning dataset from expert LLMs, *e.g.*, GPT-4o and o3-mini, playing all 12 games in Orak using several agentic modules. This dataset with environment interaction trajectories encapsulates meta-knowledge on how to use the agentic strategies to solve diverse game genres.

**Data Format.** We denote LLMs' gameplay *trajectory* as $\mathcal{T} = \{\tau_1, \ldots, \tau_T\}$, where $T$ is the number of game steps, and $\tau_t$ denotes the sequence of LLM inferences executed by agentic strategies at game step $t$. Each LLM inference sequence $\tau$ is represented as $\tau = \{(X^{a_i}, S, Y^{a_i})\}_{i=1}^n$, where $a_i \in \{\text{'reflection', 'planning', \ldots, 'action'}\}$ is the $i$-th agentic module in sequence, $X^a$ is the prompt for agentic module $a$, $S$ is the game state, and $Y^a$ is the corresponding response of LLM. Table 2 shows detailed data examples of $\tau$.

**Data Selection.** For each game in Orak, we collect gameplay trajectories of expert LLMs, until we have more than 1000 LLM inference sequence $\tau$ in all $\mathcal{T}$ collected. To ensure high-quality data, we sort the collected $\mathcal{T}$ in terms of the game score and select the trajectories with the highest game scores until the number of selected $\tau$ exceeds 300. All selected trajectories follow the 'reflection-planning-action' sequence, so that we have around 300 samples for each agent module. By performing data selection on all 12 games, our fine-tuning set consists of approximately 11k samples.

**Data Augmentation.** To enhance the linguistic diversity, we augment each data sample $\tau$ by paraphrasing. We prompt GPT-4o to rephrase the game prompt $X^a$ while preserving all game-related information, generating 10 augmented samples for each sample $\tau$. See Appendix O for prompting details and the effect of the number of augmentations.

Our fine-tuning dataset is mainly for supervised fine-tuning (SFT). While dynamic data extraction from the environment could enable reinforcement learning fine-tuning, we leave this for future work.

| Genre | Action | | Adventure | | RPG | | Simulation | | Strategy | | Puzzle | | Avg |
|---|---|---|---|---|---|---|---|---|---|---|---|---|---|
| Games | SF3 | SuperMario | AceAttorney | HerStory | Pokémon | DarkestD | Minecraft | Stardew | StarCraft2 | SlaySpire | BabaIsYou | 2048 | Rank |
| Llama-3.2-1B | 0.0±0.0 | 18.7±8.6 | 1.3±2.2 | 2.1±1.2 | 0.0±0.0 | 0.0±0.0 | 0.0±0.0 | 0.0±0.0 | 0.0±0.0 | 0.0±0.0 | 6.7±11.5 | 0.0±0.1 | 13.5 |
| Llama-3.2-3B | 13.3±5.8 | 31.8±10.1 | 4.6±1.3 | 4.2±1.1 | 0.0±0.0 | 47.5±39.2 | 0.0±0.0 | 0.0±0.0 | 0.0±0.0 | 0.0±0.0 | 20.0±0.0 | 0.3±0.2 | 11.4 |
| Qwen-2.5-3B | 20.0±0.0 | 23.4±14.1 | 20.0±17.4 | 1.2±1.1 | 0.0±0.0 | 44.8±22.2 | 0.0±0.0 | 0.0±0.0 | 0.0±0.0 | 0.0±0.0 | 13.3±11.5 | 0.1±0.1 | 12.1 |
| Qwen-2.5-7B | 16.7±11.5 | 27.2±9.6 | 9.3±0.2 | 8.5±2.0 | 0.0±0.0 | 88.8±2.0 | 0.0±0.0 | 0.0±0.0 | 0.0±0.0 | 5.0±0.0 | 20.0±0.0 | 0.6±0.4 | 10.5 |
| Minitron-4B | 16.7±11.5 | 24.4±6.0 | 35.7±4.5 | 4.6±2.3 | 0.0±0.0 | 0.0±0.0 | 0.0±0.0 | 0.0±0.0 | 0.0±0.0 | 0.0±0.0 | 20.0±0.0 | 0.1±0.0 | 11.6 |
| Minitron-8B | 23.3±5.8 | 31.3±12.8 | 29.9±3.6 | 8.2±1.8 | 0.0±0.0 | 63.8±30.4 | 0.0±0.0 | 0.0±0.0 | 0.0±0.0 | 0.0±0.0 | 20.0±0.0 | 0.7±0.7 | 10.1 |
| Llama3.3-70B | **33.3**±28.9 | 32.0±11.8 | 53.9±23.4 | 46.4±10.8 | 16.7±14.4 | 87.2±8.2 | 0.0±0.0 | 40.9±37.2 | 0.0±0.0 | 5.0±0.0 | 20.0±0.0 | 1.4±1.3 | 7.3 |
| Qwen2.5-72B | 26.7±5.8 | 29.9±8.7 | 14.9±8.5 | 41.0±1.7 | 27.8±9.6 | 84.0±6.6 | 0.0±0.0 | 18.0±16.7 | 0.0±0.0 | 9.0±5.3 | 20.0±0.0 | 0.2±0.1 | 8.8 |
| GPT-4o-mini | 16.7±11.5 | 28.8±8.8 | 28.4±2.8 | 21.2±5.6 | 0.0±0.0 | 81.3±5.8 | 46.0±7.0 | 16.1±27.8 | 75.0±50.0 | 3.3±2.9 | 13.3±11.5 | 1.1±1.0 | 9.2 |
| GPT-4o | 29.7±14.3 | 34.1±14.2 | 85.3±1.5 | 64.2±5.2 | 38.9±9.6 | 93.4±1.5 | 71.0±7.0 | 81.4±4.8 | **100.0**±0.0 | 23.6±22.1 | 20.0±0.0 | 5.6±1.5 | 3.6 |
| GPT-5 | 19.5±11.0 | 31.6±10.5 | 59.1±26.5 | **74.9**±1.7 | **88.9**±4.8 | 92.9±0.7 | 73.0±7.0 | **92.3**±8.6 | 25.0±50.0 | 26.2±19.4 | **100.0**±0.0 | 10.2±2.1 | 3.6 |
| o3-mini | **33.3**±15.3 | 34.9±14.6 | **91.7**±1.5 | 66.5±3.6 | 0.0±0.0 | 89.0±2.1 | 75.0±0.0 | 55.1±16.0 | 25.0±50.0 | 15.0±0.0 | 73.3±46.2 | **25.3**±7.3 | 4.0 |
| Gemini-2.5-pro | 13.3±11.5 | **38.0**±14.4 | 55.7±3.4 | 67.5±3.3 | 83.3±0.0 | **93.7**±1.6 | 75.0±0.0 | 59.2±10.1 | **100.0**±0.0 | **51.9**±31.9 | 73.3±46.2 | 5.1±2.5 | **3.5** |
| Claude-3.7 | 16.7±11.5 | 31.7±8.2 | 81.9±1.6 | 62.9±2.6 | 63.9±19.2 | 89.9±2.5 | 75.0±0.0 | 53.6±20.9 | 50.0±57.7 | 15.0±0.0 | 46.7±46.2 | 5.3±2.7 | 5.1 |
| Deepseek-R1 | 20.0±0.0 | 28.7±13.2 | 83.3±1.5 | 67.2±3.9 | 75.0±0.0 | 91.7±1.1 | 41.7±0.0 | 66.1±11.5 | 50.0±57.7 | 24.9±17.1 | 20.0±0.0 | 11.5±3.4 | 5.1 |
| Human (Novice) | 20.0±20.0 | 100.0±0.0 | 87.8±2.6 | 72.6±2.8 | 86.1±12.7 | 90.3±2.0 | 70.8±0.0 | 69.8±19.9 | 33.3±57.7 | 52.9±23.2 | 100.0±0.0 | 22.7±10.4 | - |

Table 3: Performance of LLMs on Orak with default agentic strategies. The best scores for each game are highlighted in bold, and the average LLM rankings across all games are reported.

# 5 EXPERIMENT

## 5.1 EXPERIMENT SETUP

**Models.** We validate the performance of 15 LLMs using states provided in text format. The models include 8 open-source LLMs: LLaMA-3.2-1B/3B (Grattafiori et al., 2024), LLaMA-3.3-72B (Grattafiori et al., 2024), Qwen-2.5-3B/7B/72B (Yang et al., 2024), and Minitron-4B/8B (Sreenivas et al., 2024), and 7 proprietary LLMs: GPT-5/4o/4o-mini (Achiam et al., 2023), o3-mini, Gemini-2.5-pro (Team et al., 2023), Claude-3.7-sonnet (Anthropic, 2025), and DeepSeek-R1 (Guo et al., 2025). In addition, we study the effects of incorporating image inputs on 5 multi-modal LLMs: Qwen2.5-vl-7B/32B (Bai et al., 2025), GPT-4o, Gemini-2.5-pro, and Claude-3.7-sonnet.

**Default Agentic Strategies.** For each game in Orak, we select the most effective agentic strategy over GPT-4o as the default agent strategy. Specifically, For *Street Fighter III*, *HerStory*, *Darkest Dungeon*, and *2048*, we use a 'zero-shot' action inference agent. For *Super Mario*, *Pokémon Red*, *Stardew Valley*, *StarCraft II*, *Slay the Spire*, and *Baba Is You*, we use a 'reflection-planning' agent that sequentially performs self-reflection, subtask planning, and action inference, integrated with memory at each game step. For *AceAttorney*, we use a 'reflection' agent. For *Minecraft*, we use a 'skill-management' agent that further includes knowledge retrieval and skill management in the reflection-planning-action agent, following (Wang et al., 2023).

**Metrics and Implementation Details.** For each game, we report the normalization score rather than the absolute score, *i.e.*, the game score is normalized by the maximum game score. We report the average score of 3 to 20 trials for each game. We also report the average score of 3 human novices. More detailed metrics and LLM hyperparameter configurations can be found in Appendices C-N.

## 5.2 LLM GAMEPLAY PERFORMANCE

Table 3 shows the gameplay performance of LLMs on Orak with default agentic strategies. Overall, proprietary LLMs outperform open-source LLMs across all games in most cases. Gemini-2.5-pro performs the best on average, ranking first in 5 out of 12 games, with the best average ranking of 3.5. gpt-5/4o forms the second-best group with an average ranking of 3.6. GPT-5 shows superiority in puzzle games, *i.e.*, *Baba Is You* and *2048*, which require strong mathematical and logical reasoning, and spatial understanding abilities. Most small open-source LLMs, size under 8B, show almost zero score on complex games, i.e., *Pokémon-red*, *Minecraft*, *Stardew Valley*, *StarCraft II*, and *Slay the Spire*. Mid-sized open-source models, Llama3.3-70B and Qwen2.5-72B, show advances in moderately hard games, such as *Stardew Valley* and *Slay the Spire*, but their performance remains far below that of proprietary LLMs. See Appendices C-N for more detailed results on each game.

## 5.3 LLM ARENA

Among 12 games in Orak, *Street Fighter III* and *StarCraft II* support two-player competitive modes. For *Street Fighter III*, we conduct pairwise battles among 8 LLMs with 'zero-shot' agent. Each pair

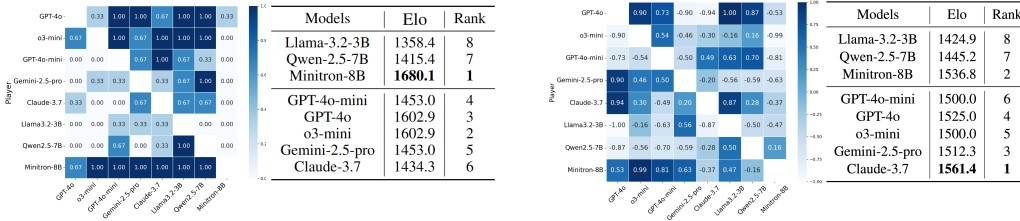

(a) *Street Fighter III* Arena.      (b) *StarCraft II* Arena.

Figure 4: Match outcomes and Elo ratings for LLMs in two competitive environments.

| Models | Agent | Action | | Adventure | | RPG | | Simulation | | Strategy | | Puzzle | | Avg |
|---|---|---|---|---|---|---|---|---|---|---|---|---|---|---|
| | Strategies | SF3 | SuperMario | AceAttorney | HerStory | Pokémon | DarkestD | Minecraft | Stardew | StarCraft2 | SlaySpire | BabaIsYou | 2048 | Rank |
| LLaMA-3B | Zeroshot | $13.3_{\pm5.8}$ | $21.2_{\pm8.2}$ | $5.7_{\pm3.2}$ | $4.2_{\pm1.1}$ | $0.0_{\pm0.0}$ | $47.5_{\pm39.2}$ | $0.0_{\pm0.0}$ | $0.0_{\pm0.0}$ | $0.0_{\pm0.0}$ | $0.0_{\pm0.0}$ | $20.0_{\pm0.0}$ | $0.3_{\pm0.2}$ | 6.4 |
| | Reflection | $\mathbf{30.0}_{\pm17.3}$ | $32.4_{\pm8.6}$ | $4.6_{\pm1.3}$ | $4.4_{\pm1.3}$ | $0.0_{\pm0.0}$ | $47.3_{\pm39.0}$ | $0.0_{\pm0.0}$ | $0.0_{\pm0.0}$ | $0.0_{\pm0.0}$ | $0.0_{\pm0.0}$ | $20.0_{\pm0.0}$ | $0.0_{\pm0.0}$ | 5.6 |
| | Planning | $20.0_{\pm0.0}$ | $27.0_{\pm8.4}$ | $4.6_{\pm1.3}$ | $5.2_{\pm1.0}$ | $0.0_{\pm0.0}$ | $56.3_{\pm23.6}$ | $0.0_{\pm0.0}$ | $0.0_{\pm0.0}$ | $0.0_{\pm0.0}$ | $0.0_{\pm0.0}$ | $20.0_{\pm0.0}$ | $0.1_{\pm0.2}$ | 6.2 |
| | Ref-Plan | $16.7_{\pm20.8}$ | $31.8_{\pm10.1}$ | $3.8_{\pm0.0}$ | $5.4_{\pm0.4}$ | $0.0_{\pm0.0}$ | $57.0_{\pm31.6}$ | $0.0_{\pm0.0}$ | $0.0_{\pm0.0}$ | $0.0_{\pm0.0}$ | $0.0_{\pm0.0}$ | $20.0_{\pm0.0}$ | $0.1_{\pm0.2}$ | 6.1 |
| GPT-4o | Zeroshot | $29.7_{\pm14.3}$ | $29.6_{\pm9.2}$ | $49.9_{\pm1.3}$ | $\mathbf{64.2}_{\pm5.2}$ | $33.3_{\pm0.0}$ | $\mathbf{93.4}_{\pm1.5}$ | $0.0_{\pm0.0}$ | $34.5_{\pm29.9}$ | $50.0_{\pm57.7}$ | $24.7_{\pm9.2}$ | $20.0_{\pm0.0}$ | $5.6_{\pm4.0}$ | 3.3 |
| | Reflection | $23.3_{\pm20.8}$ | $32.3_{\pm15.5}$ | $\mathbf{85.3}_{\pm1.5}$ | $61.5_{\pm0.4}$ | $36.1_{\pm4.8}$ | $85.2_{\pm10.3}$ | $\mathbf{50.0}_{\pm0.0}$ | $15.6_{\pm4.5}$ | $0.0_{\pm0.0}$ | $\mathbf{41.3}_{\pm19.6}$ | $20.0_{\pm0.0}$ | $3.5_{\pm2.9}$ | 3.0 |
| | Planning | $\mathbf{30.0}_{\pm26.5}$ | $29.4_{\pm12.5}$ | $52.7_{\pm0.5}$ | $59.4_{\pm5.7}$ | $33.3_{\pm0.0}$ | $82.0_{\pm8.6}$ | $13.0_{\pm0.0}$ | $54.9_{\pm20.2}$ | $50.0_{\pm57.7}$ | $35.3_{\pm9.2}$ | $20.0_{\pm0.0}$ | $6.0_{\pm5.5}$ | 3.1 |
| | Ref-Plan | $23.3_{\pm20.8}$ | $\mathbf{34.1}_{\pm14.2}$ | $52.8_{\pm0.5}$ | $62.0_{\pm4.9}$ | $\mathbf{38.9}_{\pm9.6}$ | $91.6_{\pm2.5}$ | $\mathbf{50.0}_{\pm0.0}$ | $\mathbf{81.4}_{\pm4.8}$ | $\mathbf{100.0}_{\pm0.0}$ | $36.0_{\pm25.5}$ | $20.0_{\pm0.0}$ | $\mathbf{7.0}_{\pm5.7}$ | 2.2 |

Table 4: Ablation study for agentic modules. 'Ref-Plan' refers to the 'Reflection-Planning' agent.

competed in three rounds, and the agent winning 2 out of 3 rounds was declared the winner. To ensure a fair comparison, both agents were assigned the same character, Ken, in all matches. Figure 4(a) shows the relative win rates and Elo ratings. Interestingly, different from the result in Section 5.2, Minitron-8B consistently outperforms all other LLMs and shows the best Elo rating. This may imply that when multiple agents are involved in the environment, adversarial actions can change the game dynamics. For *StarCraft II*, we conduct pairwise battles among 7 LLMs, with each pair competing in a single round. Both agents were assigned the same race, Protoss, in all matches. As in Figure 4(b), Claude-3.7-Sonnet shows the best Elo rating, while GPT-4o and Minitron-8B form the second group.

## 5.4 ABLATION STUDY FOR AGENTIC MODULES

Table 4 shows the ablation results of Llama-3.2-3B and GPT-4o across 4 agent strategies. Interestingly, the impact of adding agentic modules to gameplay performance *differs* between the two LLMs. For GPT-4o, the inclusion of agentic modules consistently improves gameplay performance; 'reflection-planning' agent achieves the best average ranking of 2.2, followed by 'reflection' and 'planning' agents with rankings of 3.1 and 3.3, and 'zero-shot' agent with the lowest ranking of 3.4. However, Llama-3.2-3B does not follow this trend. The 'reflection' agent shows the highest average ranking of 5.6, while 'reflection-planning' follows with 6.1, although it adds the planning module. This indicates that, for relatively smaller LLMs like Llama-3.2-3B, adding agentic modules may increase the complexity of the prompt, hindering their decision-making accuracy. These results suggest that the optimal agentic strategy may depend on the inherent capability of the LLM. See Appendices C-N for more detailed ablation analysis for each game.

## 5.5 EFFECT OF VISUAL INPUT

**Setup.** To study the effect of visual input, we divide games into two groups: Group 1 (Table 5) includes games where the provided textual game state can entirely be derived from a visual screenshot. Group 2 (Table 6) comprises games where the textual state contains information beyond what is visible in the current frame (e.g., abstracted inventory lists or off-screen character/item details); for this group, we exclude evaluation on *Image-only* input types for fairness. *Minecraft* was excluded since the environment does not provide the visionary view of a bot that LLM plays.

**Result.** As shown in Table 5, relying solely on *Image-only* input led to a substantial drop in performance. This was consistently reflected across all models, with average ranks deteriorating significantly compared to *Text-only* input. In contrast, as shown in Table 5& 6, utilizing *Both* text and visual inputs produced mixed effects on game scores and model ranks. For instance, in games like *Street Fighter III* (Group 1), since on-screen visual details are challenging to fully convey textually, adding visual context significantly benefited Claude, increasing its score by 16.6. Conversely, in narrative-heavy games such as *Ace Attorney* (Group 2), the same approach often proved detrimental; GPT-4o's score, for example, dropped by 31.8, and its average rank in that group fell from 2.9

| Models | Input | SF3 | SuperMario | Stardew | BabaIsYou | 2048 | Rank |
|---|---|---|---|---|---|---|---|
| Qwen2.5-7B | Text | $0.0_{\pm0.0}$ | $26.0_{\pm8.2}$ | $0.0_{\pm0.0}$ | $13.3_{\pm11.5}$ | $0.1_{\pm0.1}$ | 12.8 |
| | Image | $3.3_{\pm5.8}$ | $25.1_{\pm10.1}$ | $0.0_{\pm0.0}$ | $0.0_{\pm0.0}$ | $0.2_{\pm0.3}$ | 13.5 |
| | Both | $3.3_{\pm5.8}$ | $25.6_{\pm8.0}$ | $0.0_{\pm0.0}$ | $0.0_{\pm0.0}$ | $0.4_{\pm0.5}$ | 12.7 |
| Qwen2.5-32B | Text | $16.7_{\pm5.8}$ | $32.0_{\pm10.2}$ | $15.2_{\pm22.7}$ | $20.0_{\pm0.0}$ | $0.2_{\pm0.3}$ | 4.3 |
| | Image | $33.3_{\pm20.8}$ | $25.8_{\pm10.7}$ | $0.0_{\pm0.0}$ | $13.3_{\pm11.5}$ | $0.4_{\pm0.3}$ | 5.1 |
| | Both | $33.3_{\pm15.3}$ | $29.7_{\pm6.6}$ | $26.9_{\pm13.9}$ | $20.0_{\pm0.0}$ | $0.2_{\pm0.3}$ | 3.4 |
| GPT-4o | Text | $29.7_{\pm14.3}$ | $34.1_{\pm14.1}$ | $81.4_{\pm4.8}$ | $20.0_{\pm0.0}$ | $5.6_{\pm1.5}$ | 1.7 |
| | Image | $23.7_{\pm15.9}$ | $27.1_{\pm13.7}$ | $0.0_{\pm0.0}$ | $6.7_{\pm11.5}$ | $1.8_{\pm1.1}$ | 5.3 |
| | Both | $24.3_{\pm14.5}$ | $27.1_{\pm10.2}$ | $41.9_{\pm22.1}$ | $20.0_{\pm0.0}$ | $5.4_{\pm4.5}$ | 3.0 |
| Gemini-2.5 | Text | $13.3_{\pm11.5}$ | $38.0_{\pm13.4}$ | $59.2_{\pm10.1}$ | $73.3_{\pm46.2}$ | $5.1_{\pm2.5}$ | 5.2 |
| | Image | $16.7_{\pm11.5}$ | $28.5_{\pm10.7}$ | $7.6_{\pm9.0}$ | $20.0_{\pm0.0}$ | $5.5_{\pm2.4}$ | 7.4 |
| | Both | $20.0_{\pm10.0}$ | $40.9_{\pm9.6}$ | $60.0_{\pm6.0}$ | $86.7_{\pm23.1}$ | $3.1_{\pm2.6}$ | 4.0 |
| Claude-3.7 | Text | $16.7_{\pm11.5}$ | $28.7_{\pm13.2}$ | $53.6_{\pm20.9}$ | $46.7_{\pm46.2}$ | $5.3_{\pm2.7}$ | 5.8 |
| | Image | $23.3_{\pm11.5}$ | $25.6_{\pm6.4}$ | $0.0_{\pm0.0}$ | $20.0_{\pm0.0}$ | $8.4_{\pm4.0}$ | 8.0 |
| | Both | $33.3_{\pm5.8}$ | $22.6_{\pm6.3}$ | $49.8_{\pm1.0}$ | $20.0_{\pm0.0}$ | $6.7_{\pm0.9}$ | 6.2 |

Table 5: Modality comparison (Group 1).

| Models | Input | AceAttorney | HerStory | Pokémon | DarkestD | StarCraft2 | SlaySpire | Rank |
|---|---|---|---|---|---|---|---|---|
| Qwen2.5-7B | Text | $10.0_{\pm0.0}$ | $3.2_{\pm1.5}$ | $2.8_{\pm4.8}$ | $82.7_{\pm1.5}$ | $0.0_{\pm0.0}$ | $0.0_{\pm0.0}$ | 9.1 |
| | Both | $17.6_{\pm13.1}$ | $2.5_{\pm0.8}$ | $2.8_{\pm4.8}$ | $81.8_{\pm3.0}$ | $0.0_{\pm0.0}$ | $0.0_{\pm0.0}$ | 9.3 |
| Qwen2.5-32B | Text | $71.9_{\pm21.5}$ | $15.4_{\pm2.8}$ | $27.8_{\pm9.6}$ | $89.9_{\pm1.5}$ | $0.0_{\pm0.0}$ | $0.0_{\pm0.0}$ | 7.1 |
| | Both | $68.6_{\pm10.0}$ | $17.0_{\pm4.6}$ | $25.0_{\pm14.4}$ | $91.0_{\pm3.0}$ | $0.0_{\pm0.0}$ | $0.0_{\pm0.0}$ | 7.0 |
| GPT-4o | Text | $85.3_{\pm1.5}$ | $64.2_{\pm5.2}$ | $38.9_{\pm9.6}$ | $93.4_{\pm1.5}$ | $100.0_{\pm0.0}$ | $23.6_{\pm22.1}$ | 2.9 |
| | Both | $53.5_{\pm1.7}$ | $40.6_{\pm29.5}$ | $41.7_{\pm8.3}$ | $92.2_{\pm3.0}$ | $50.0_{\pm57.7}$ | $23.6_{\pm22.1}$ | 5.0 |
| Gemini-2.5 | Text | $55.7_{\pm3.4}$ | $67.5_{\pm2.3}$ | $83.3_{\pm0.0}$ | $93.7_{\pm1.6}$ | $100.0_{\pm0.0}$ | $51.9_{\pm31.9}$ | 2.1 |
| | Both | $52.6_{\pm0.8}$ | $64.9_{\pm2.4}$ | $83.3_{\pm0.0}$ | $92.2_{\pm1.8}$ | $100.0_{\pm0.0}$ | $26.2_{\pm19.4}$ | 3.2 |
| Claude-3.7 | Text | $81.9_{\pm1.6}$ | $62.9_{\pm2.6}$ | $63.9_{\pm19.2}$ | $89.9_{\pm2.5}$ | $50.0_{\pm57.7}$ | $15.0_{\pm6.0}$ | 4.8 |
| | Both | $71.3_{\pm17.3}$ | $63.6_{\pm3.1}$ | $72.2_{\pm4.7}$ | $90.1_{\pm5.7}$ | $50.0_{\pm57.7}$ | $9.7_{\pm4.6}$ | 4.3 |

Table 6: Modality comparison (Group 2).

| Models | Finetune | Intra-Game | | | | | OOD-Game | | Non-Game | | |
|---|---|---|---|---|---|---|---|---|---|---|---|
| | | SF3 | DarkestD | StarCraft2 | SlaySpire | BabaIsYou | SuperMario | 2048 | Math500 | WebShop-E | WebShop-H |
| Llama-3.2-1B | ✗ | $0.0_{\pm0.0}$ | $0.0_{\pm0.0}$ | $0.0_{\pm0.0}$ | $0.0_{\pm0.0}$ | $20.0_{\pm0.0}$ | $18.7_{\pm8.6}$ | $0.1_{\pm0.1}$ | $24.1_{\pm1.6}$ | $2.7_{\pm0.5}$ | $1.9_{\pm0.8}$ |
| | ✓ | $42.0_{\pm16.4}$ | $93.4_{\pm2.6}$ | $0.0_{\pm0.0}$ | $8.0_{\pm3.5}$ | $20.0_{\pm0.0}$ | $26.7_{\pm12.3}$ | $2.8_{\pm1.8}$ | $25.6_{\pm1.7}$ | $0.5_{\pm0.0}$ | $0.0_{\pm0.0}$ |
| Llama-3.2-3B | ✗ | $12.0_{\pm11.0}$ | $87.2_{\pm9.5}$ | $0.0_{\pm0.0}$ | $0.0_{\pm0.0}$ | $20.0_{\pm0.0}$ | $31.8_{\pm10.1}$ | $0.1_{\pm0.2}$ | $29.6_{\pm1.8}$ | $0.0_{\pm0.0}$ | $0.0_{\pm0.0}$ |
| | ✓ | $40.0_{\pm7.1}$ | $92.0_{\pm0.2}$ | $0.0_{\pm0.0}$ | $10.7_{\pm1.2}$ | $20.0_{\pm0.0}$ | $34.4_{\pm7.0}$ | $3.1_{\pm2.5}$ | $36.1_{\pm0.9}$ | $8.4_{\pm0.3}$ | $12.6_{\pm0.8}$ |

Table 7: Generalization performance of LLMs fine-tuned on expert gameplay trajectories from Orak.

(*Text-only*) to 5.0. This highlights that the impact of combining modalities varies considerably, with some scenarios showing improved ranks or scores while others demonstrated a decline.

## 5.6 EFFECT OF FINE-TUNING

**Setup.** To study the effect of fine-tuning, we consider three types of generalization: *Intra-game*, *OOD-game*, and *Non-game*. Intra-game generalization evaluates whether an LLM can adapt to *unseen scenarios* within the same game, *e.g.*, new stages or characters. OOD-game generalization evaluates whether fine-tuning on a specific set of games enables the model to act better on *unseen games*. Non-game generalization evaluates whether our gameplay fine-tuning set helps the model to perform better on non-game tasks like math or web navigation. For Intra-game and Non-game generalizations, we fine-tune LLMs on all 12 games, while for OOD-game generalization, we split 12 games in to 10 training games and 2 test games, i.e., *Super Mario* and *2048*. Additionally, to study the effect of data quality and scaling, we perform fine-tuning on high-score (original), low-score (samples with the lowest game score), and their mixed datasets (doubling the data volume). See Appendix P for detailed fine-tuning configurations and unseen scenario details in intra-game generalization.

**Intra-game Generalization.** As shown in Table 7, fine-tuned Llama-3.2-1B/3B show better generalization to unseen scenarios than the pretrained ones; in 3 out of 5 games, both fine-tuned models outperform their pretrained counterparts. The performance gain largely comes from the model learning to generate valid actions more reliably after fine-tuning, especially in environments where the pretrained model frequently failed to act meaningfully. However, this approach appears insufficient to significantly enhance the spatial reasoning abilities of smaller models. This is evident in *Baba Is You*, where fine-tuned models struggle to construct the winning condition required to score above 20.0. Also, since StarCraft II requires complex strategic reasoning to defeat enemies, the strategic patterns learned from the fine-tuning set were not sufficient for the model to win on new maps.

**OOD-game Generalization.** Similarly, despite being trained exclusively on trajectories from different games, fine-tuned models perform significantly better on OOD games like *Super Mario* and *2048*. This suggests that fine-tuning on trajectories shaped by reflection and planning enables models to learn transferable decision-making routines, as the underlying capabilities required for these behaviors are shared across games. Specifically, the improvement in *2048* likely benefits from its structural similarity to *Baba Is You*, a training game that also uses 2D grid layout and discrete move actions (up, down, left, right). More fine-tuning analysis with varying OOD gaps are in Appendix Q.

**Non-game Generalization.** More interestingly, LLMs fine-tuned on our gameplay trajectories sometimes show improved performance in non-game tasks like Math500 and WebShop (Yao et al., 2022). In Table 7, LLaMA-3.2-3B consistently improves on both Math500 and Webshop. In particular, on WebShop-E (easy) and -H (hard), it shows a dramatic gain from 0.0% to 8.4% or 12.6%, indicating our fine-tuning set is more effective for agentic tasks that require decision-making abilities similar

| Data | SF3 | DarkestD | StarCraft2 | SlaySpire | BabaIsYou |
|---|---|---|---|---|---|
| No fine-tuning | 12.0±11.0 | 87.2±9.5 | 0.0±0.0 | 0.0±0.0 | **20.0**±0.0 |
| high-score (110K) | **40.0**±7.1 | **92.0**±0.2 | 0.0±0.0 | **10.7**±1.2 | **20.0**±0.0 |
| low-score (110K) | 18.0±16.4 | 40.0±0.0 | 0.0±0.0 | 8.7±2.3 | **20.0**±0.0 |
| mixed (220K) | 18.0±14.8 | 40.0±0.0 | 0.0±0.0 | **10.7**±1.2 | **20.0**±0.0 |

(a) Effect of fine-tuning data quality and scale.

| Model | Hard (Original) | | Easy | |
|---|---|---|---|---|
| | Paused | Real-time | Paused | Real-time |
| gpt-4o-mini (19.6 sec) | 75.0±0.0 | 0.0±0.0 | **100.0**±0.0 | 0.0±0.0 |
| gpt-4o (27.2 sec) | **100.0**±0.0 | 0.0±0.0 | **100.0**±0.0 | **100.0**±0.0 |
| gemini-2.5-pro (99.2 sec) | **100.0**±0.0 | 0.0±0.0 | **100.0**±0.0 | 0.0±0.0 |

(b) Real-time game performance on *StarCraft II*.

Table 8: Effect of (a) fine-tuning data quality/scale and (b) latency-aware real-time evaluation.

to gameplay. In contrast, the smaller model, Llama-3.2-1B, shows no observable generalization to non-game tasks, which suggests that the extent of generalization may depend on model capacity.

**Effect of Data Quality and Scaling.** Table 8(a) shows the effect of data quality and scaling in fine-tuning generalization. Overall, high-score dataset alone is the most effective, while the low-score dataset contributes little or even degrades performance in certain cases (e.g., *Darkest Dungeon*). Simply scaling up the dataset by mixing high- and low-score samples does not yield a clear improvement, indicating that data quality, rather than sheer quantity, plays a more critical role in achieving better generalization.

## 5.7 LATENCY-AWARE EVALUATION IN REAL-TIME GAMEPLAY

Among the 12 games in Orak, 3 games, including *Street Fighter III*, *Super Mario*, and *Starcraft II*, require pausing for evaluation. Here, we conduct a latency-aware evaluation on *StarCraft II* by enabling real-time gameplay. As shown in Table 8(b), when switching from the paused setting to the real-time mode, all models failed (0 score) under the hard level (the original setup), indicating that timing-sensitive games are strongly affected by inference latency. When the difficulty level was reduced to easy, all models achieved perfect scores under the paused setting, but only GPT-4o maintained perfect performance under real-time conditions. The average response latencies per step with the reflection–planning agent were 19.6 s for GPT-4o-mini, 27.2 s for GPT-4o, and 99.2 s for Gemini-2.5-pro. These results suggest that GPT-4o provides the best trade-off between latency and gameplay accuracy among the three evaluated models.

## 6 DISCUSSION

**Conclusion.** In this paper, we introduce Orak, a benchmark designed to train and evaluate LLM agents across diverse video games. Orak enables comprehensive assessments of LLM capabilities required to play most game genres. Through a plug-and-play interface powered by MCP, it allows consistent evaluation of rapidly evolving LLMs over various agentic modules. In addition, we release a fine-tuning dataset of game interaction trajectories of top-performing LLMs, which can effectively transform pre-trained LLMs into gaming agents. With the comprehensive game set and user-friendly interface, Orak sets a new foundation for game-based LLM evaluation, driving progress towards versatile and high-performing gaming agents.

**Limitations.** We note that Orak provides environments favorable for LLM reasoning and decision-making by pre-processing game states into structured text where information unnecessary to gameplay is hidden. This may offer insights for providers seeking to deploy resource-efficient LLMs on games. We leave a complementary direction, providing the full in-game states with rich uncurated texts to LLMs, for future work. Further discussions, including cost and licensing, are provided in Appendix U.

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

# Orak: A Foundational Benchmark for Training and Evaluating LLM Agents on Diverse Video Games
## (Supplementary Material)

## A    GAME GENRE CATEGORIZATION

Video games are generally categorized into six widely recognized genres, *i.e.*, Action, Adventure, Role-Playing, Simulation, Strategy, and Puzzle, each characterized by distinct gameplay structures (Wikipedia contributors, 2025). **Action** games emphasize responsiveness and precise physical control. **Adventure** games focus on narrative exploration, interactive dialogues, and clue-based progression grounded in logical inference. **Role-Playing** games focus on character progression, stat-driven combat mechanics, and quest-based narrative development. **Simulation** games present system-driven environments in which players manage complex and interdependent variables such as time, resources, and procedural systems. **Strategy** games are a genre that emphasizes planning, resource management, decision-making, and tactical execution. **Puzzle** games revolve around rule-based problem solving, pattern recognition, and logical or spatial reasoning, typically within a clearly defined system of constraints.

In Table 9, we categorize each of the 12 video games in Orak into one of the six major genres based on the primary gameplay characteristics and the genre information provided by the respective game publishers. Some games belong to one or more genres due to hybrid gameplay mechanics.

| Game | Action | Adventure | Role-Playing | Simulation | Strategy | Puzzle |
|---|:---:|:---:|:---:|:---:|:---:|:---:|
| Street Fighter III | ○ | | | | | |
| Super Mario | ○ | | | | | |
| Ace Attorney | | ○ | | | | |
| Her Story | | ○ | | | | |
| Pokémon Red | | △ | ○ | | | |
| Darkest Dungeon | | △ | ○ | | △ | |
| Minecraft | | | △ | ○ | | |
| Stardew Valley | | △ | △ | ○ | | |
| StarCraft II | △ | | | | ○ | |
| Slay the Spire | | | | | ○ | |
| Baba Is You | | | | | | ○ |
| 2048 | | | | | | ○ |

Table 9: Genre categorization of the 12 games in Orak. ○ denotes the *main* genre and △ indicates the *secondary* genre.

**(a) Street Fighter III** (Capcom, 1997) is classified as an *Action* game, as it primarily relies on responsiveness, frame-precise input, and physical dexterity. Its gameplay requires fast reflexes and mastery of complex input sequences.

**(b) Super Mario** (Christian Kauten, 2018) is categorized as *Action* game. The core gameplay emphasizes precise timing in jumping and movement, demanding moment-to-moment control in response to environmental hazards and enemy placements.

**(c) Ace Attorney** (Capcom, 2001) is labeled as an *Adventure* game due to its narrative-driven structure, reliance on clue collection, and logical deduction. Players progress by interacting with characters and uncovering story elements through investigative mechanics, with minimal emphasis on reflex-based input.

**(d) Her Story** (Barlow, 2015) similarly fits within the *Adventure* category. Though more experimental in form, it shares a strong focus on narrative discovery through a search-based interface, requiring players to piece together a fragmented story using non-linear exploration and deductive reasoning.

**(e) Pokémon Red** (Game Freak, 1996) is classified as a *Role-Playing* game, as its primary mechanics involve turn-based combat, character progression, stat management, and quest-driven exploration.

Additionally, *Adventure* was assigned as a secondary genre to reflect the game's emphasis on world exploration, interaction with non-player characters, and sequential progression through narrative landmarks.

**(f) Darkest Dungeon** (Red Hook Studios, 2016) is similarly assigned *Role-Playing* as the main genre, supported by stat-driven character development and progression systems. However, it also includes significant *Strategy* elements, as players must carefully manage resources, form party compositions, and make tactical decisions in turn-based combat. *Adventure* was further added as a secondary genre due to its dungeon-crawling structure and emphasis on risk-driven exploration.

**(g) Minecraft** (Mojang Studios, 2011) is categorized as a *Simulation* game due to its open-ended, system-driven mechanics, including resource gathering, crafting, and environmental manipulation. It also exhibits *Role-Playing* traits through its progression systems and player-driven narrative development, warranting secondary classification.

**(h) Stardew Valley** (ConcernedApe, 2016) is also assigned to the *Simulation* genre based on its emphasis on time management, farming systems, and interrelated mechanics spanning multiple in-game variables. It incorporates *Role-Playing* through relationship-building and character progression, and *Adventure* through dungeon exploration, seasonal events, and quest-based interactions.

**(i) StarCraft II** (Blizzard Entertainment, 2010) is classified as a *Strategy* game, consistent with its real-time strategic planning, resource allocation, and micromanagement mechanics. Given the importance of unit-level control, *Action* was added as a secondary genre to reflect the real-time, reflex-driven demands during gameplay.

**(j) Slay the Spire** (MegaCrit, 2017) is categorized solely as a *Strategy* game. The gameplay centers around deck-building, route optimization, and turn-based combat, requiring players to plan several moves.

**(k) Baba Is You** (Hempuli, 2019) is clearly identified as a *Puzzle* game, as its mechanics are centered on solving logic-based problems through the manipulation of in-game rules represented by words. The core loop involves constrained, rule-based problem solving and spatial reasoning.

**(l) 2048** (Cirulli, 2014) is also classified as a *Puzzle* game, characterized by deterministic mechanics, arithmetic pattern recognition, and constraint-based spatial logic within a fixed grid.

This genre classification provides a structured foundation for analyzing agent cognition and gameplay dynamics across diverse games.

## B    REQUIRED LLM CAPABILITIES FOR GAMEPLAY

Looking at the Figure 3, most games demand advanced LLM capabilities across multiple dimensions, as they are designed to challenge a wide range of human cognitive skills. (1) **Action** games, *Street Fighter III* and *Super Mario* in red, require spatial reasoning and rule following, more than long-context understanding and planning. (2) **Adventure** games, *Ace Attorney* and *Her Story* in yellow, emphasize long-text understanding and logical reasoning due to the need to comprehend long storylines. (3) **Role-playing** games, *Pokémon Red* and *Darkest Dungeon* in brown, require strong long-term planning, logical reasoning, and rule-following abilities to understand game-specific rules and complete milestones of game tasks. (4) **Simulation** games, *Minecraft* and *Stardew Valley* in green, also require high levels of long-term planning and rule-following abilities. While *Minecraft* requires strong spatial reasoning and error handling, which are generally essential for simulation games, *Stardew Valley* gets lower scores for them because these abilities are not critical for its evaluation task; earning money by harvesting crops. (5) **Strategic** games, *StarCraft II* and *Slay the Spire* in blue, require various LLM abilities for gameplay. Notably, these two games are the only ones in Orak that require 5 different LLM capabilities rated at level 3, highlighting that recent strategic video games increasingly demand a wide range of cognitive skills for effective gameplay. (6) **Puzzle** games, *Baba Is You* and *2048* in purple, require high levels of spatial reasoning, logical reasoning, and long-term planning because puzzle games are typically designed to require complex problem-solving through multiple reasoning hops and spatial understanding.

## C  STREET FIGHTER III

### C.1  GAME DESCRIPTION FOR STREET FIGHTER III

**Environment.** *Street Fighter III* (Capcom, 1997) is a 2D competitive fighting game, known for precise controls, deep mechanics, and a diverse roster of characters. Each character features unique moves, combos, and super arts, requiring precise timing and strategic decision-making. Players aim to defeat their opponent through a mix of normal attacks, special moves, and advanced mechanics like parries and cancels. For implementation, we use *Diambra Arena* environment (DIAMBRA, 2025), a Docker-based platform designed for RL research. *Street Fighter III* is one of the environments supported by Diambra, which not only enables seamless extraction of

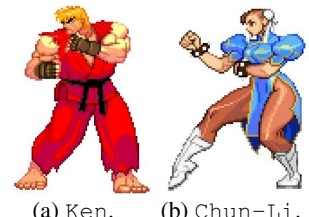

(a) Ken.       (b) Chun-Li.

Figure 5: Two of playable characters in *Street Fighter III*.

the game state—such as screenshots, health, timer, and super bar values—but also provides a straightforward interface for sending controller inputs. In this setting, the agent operates in a discrete action space that directly corresponds to various controller actions, including directional movement, attack buttons, and their combinations, enabling intuitive and fine-grained control of the in-game character.

The game supports both single-player and multi-player modes. In single-player mode, the player progresses through ten increasingly difficult stages, facing stronger opponents at each level. Each stage follows a *best-of-three* format, where the player must win two out of three matches to advance to the next stage. The game ends upon either completion of the final stage or defeat. In multi-player mode, two players compete in a best-of-three match, and the game is over once a winner is determined. For our default evaluation setting, agents play Ken in both modes (see Figure 5(a)). However, since the environment supports a variety of characters, we also conduct evaluations using Chun-Li to demonstrate intra-game generalization capabilities (see Figure 5(b)).

**Observation-to-Text Conversion.** The Diambra environment offers a convenient interface for extracting the game state from *Street Fighter III*. Through this interface, we obtain the latest game frame at a resolution of 224×384, along with key state information such as remaining time, player and opponent health, super bar gauge, super count, stun bar gauge, and stun status. However, a critical aspect in fighting games is understanding the relative positions of the characters, which is not directly provided by the Diambra environment. To address this limitation, we employ a lightweight YOLOv11 object detection model (Khanam & Hussain, 2024) to extract the relative positions of the

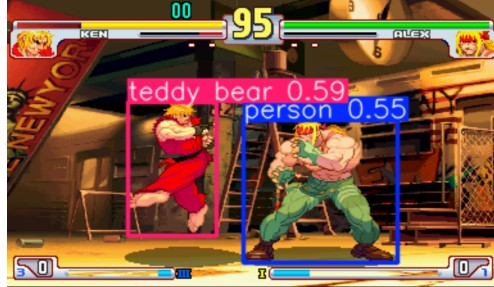

Figure 6: Character detection using YOLOv11 model (Khanam & Hussain, 2024) in *Street Fighter III*.

two characters from the game frame. Based on the computed distance, we classify the spatial relationship into three discrete categories—*very close*, *close*, and *far*—and incorporate this information into the agent's user prompt.

**Action Space.** In the Diambra environment, the native action space is defined on a per-frame basis and consists of 18 discrete actions. These are composed of:

- **Idle action (1 total)**: Idle (No action)

- **Movement actions (8 total)**: Left, Left+Up, Up, Up+Right, Right, Right+Down, Down, Down+Left

- **Attack actions (9 total)**: Low Punch, Medium Punch, High Punch, Low Kick, Medium Kick, High Kick, Low Punch+Low Kick, Medium Punch+Medium Kick, High Punch+High Kick

To enable more strategic and temporally consistent behavior, we use a higher-level action space that abstracts these frame-level controls into semantically meaningful commands. Each high-level action is mapped to a predefined sequence of low-level controller inputs, often spanning multiple frames. The high-level action space is divided into two categories:

**System prompt (Planning)**

You are a helpful AI assistant integrated with 'Street Fighter III: 3rd Strike' on the PC, assisting future decision-making. Your goal is to assist in long-term strategy planning to defeat your opponent. Based on the target task and the player's current progress, your role is to propose the most suitable subtask for the current situation. Your responses must be precise, concrete, and highly relevant to the player's objectives.

Subtask_reasoning: Decide whether the previous subtask is finished and whether it is necessary to propose a new subtask. The subtask should be straightforward, contribute to the target task and be most suitable for the current situation, which should be completed within a few actions. You should respond to me with:
1. How to finish the target task? You should analyze it step by step.
2. What is the current progress of the target task according to the analysis in step 1? Please do not make any assumptions if they are not mentioned in the above information. You should assume that you are doing the task from scratch.
3. (If previous subtask is provided) What is the previous subtask? Does the previous subtask finish? Or is it improper for the current situation? Then select a new one, otherwise you should reuse the last subtask.

**Game screenshot**

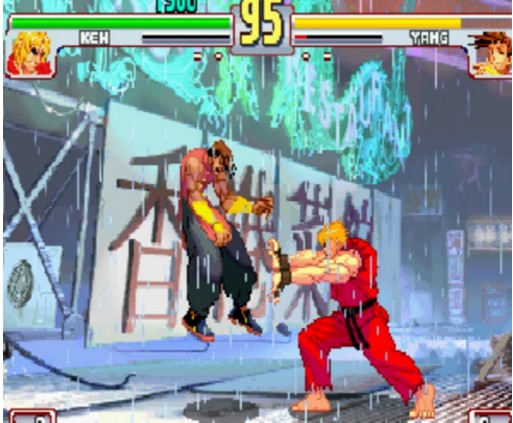

**User prompt (Planning)**

### Target task
Defeat the opponent

### Previous subtask for the task
The current subtask is to attempt another throw to break the opponent's guard and reduce their health.

### Last executed action
- Medium Punch+Medium Kick
- High Punch+High Kick

### Self reflection of the last executed action
The attempted throw to break the opponent's guard and reduce their health was unsuccessful. The opponent's health remained unchanged, and the distance between the players increased, indicating that the opponent likely evaded or countered the throw.

### Current state
You are playing Ken in Street Fighter 3. Your opponent is Yang. You are facing left.
Distance from Opponent: close
Time Remaining: 95
Health:
    Your Health: 161
    Opponent's Health: 134
Super Bar Guage:
    Your Super Bar Guage: 32
    Opponent's Super Bar Guage: 9
Super Count (Count of activated super move):
    Your Super Count: 0
    Opponent's Super Count: 0
Stun Bar Guage:
    Your Stun Bar Guage:  0
    Opponent's Stun Bar Guage: 10
IsStunned (0: not stunned, 1: stunned):
    Your Status: 0
    Opponent's Special Status: 0

You MUST respond in the format described below, and you should not output comments or other information.
### Subtask_reasoning
1. ...
2. ...
...
### Subtask
The current subtask is

Figure 7: Planning prompt for 'reflection-planning' agent playing *Street Fighter III*.

- **Character-agnostic actions (14 total)**: Move Closer, Move Away, Jump Closer, Jump Away, Super Attack, Low Punch, Medium Punch, High Punch, Low Kick, Medium Kick, High Kick, Low Punch+Low Kick, Medium Punch+Medium Kick, High Punch+High Kick
- **Character-specific actions**:
  - `Ken`: Fireball (Hadouken), Hurricane Kick, etc.
  - `Chun-Li`: Kikkoken, Hyakuretsukyaku, etc.

For example, if the character is positioned on the left side of the screen and the high-level action is 'Move Closer', the system issues 'Right' movement commands over four frames. If the action is 'Fireball', the corresponding low-level sequence would be 'Down'→'Down+Right'→'Right'→'Medium Punch'. Since the number of character-specific actions varies, the total size of the high-level action space differs depending on character, typically ranging around 20 actions.

## C.2    GAMEPLAY PROMPT FOR STREET FIGHTER III

Our implementation of *Street Fighter III* supports four types of agents: reflection, planning, reflection-planning, and zero-shot. Among these, we introduce prompts for the 'reflection-planning' agent in this subsection. Figures 7–9 present the prompts used by the reflection-planning agent for planning, action inference, and reflection, respectively. At each step, the agent plans to determine the subtask, using the planning module. Based on this subtask, the action inference module infers the optimal action to execute. Finally, the reflection module evaluates whether the executed action was successful.

**Planning prompt.** As shown in Figure 7, the system prompt provides detailed instructions for an agent to support strategic planning. It defines the assistant's role in proposing suitable subtasks based

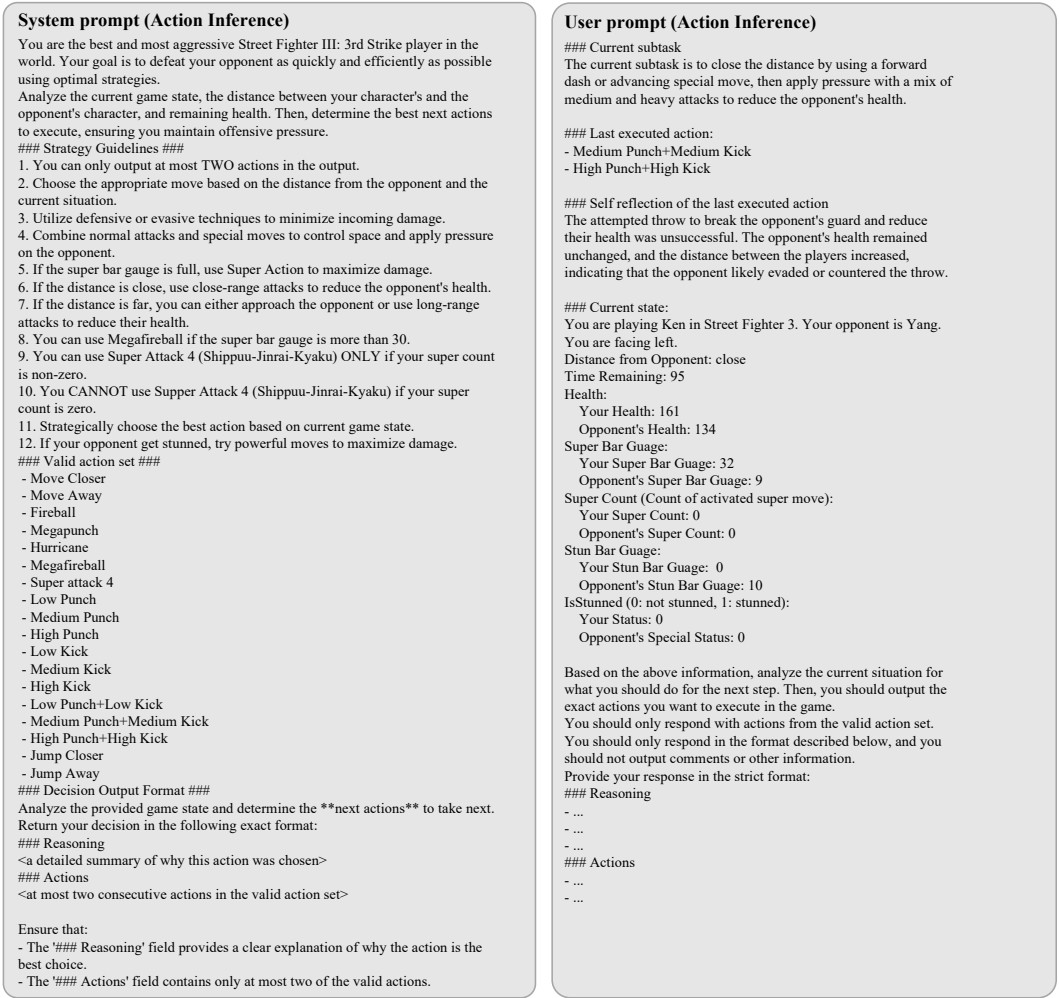

**System prompt (Action Inference)**

You are the best and most aggressive Street Fighter III: 3rd Strike player in the world. Your goal is to defeat your opponent as quickly and efficiently as possible using optimal strategies.
Analyze the current game state, the distance between your character's and the opponent's character, and remaining health. Then, determine the best next actions to execute, ensuring you maintain offensive pressure.
### Strategy Guidelines ###
1. You can only output at most TWO actions in the output.
2. Choose the appropriate move based on the distance from the opponent and the current situation.
3. Utilize defensive or evasive techniques to minimize incoming damage.
4. Combine normal attacks and special moves to control space and apply pressure on the opponent.
5. If the super bar gauge is full, use Super Action to maximize damage.
6. If the distance is close, use close-range attacks to reduce the opponent's health.
7. If the distance is far, you can either approach the opponent or use long-range attacks to reduce their health.
8. You can use Megafireball if the super bar gauge is more than 30.
9. You can use Super Attack 4 (Shippuu-Jinrai-Kyaku) ONLY if your super count is non-zero.
10. You CANNOT use Supper Attack 4 (Shippuu-Jinrai-Kyaku) if your super count is zero.
11. Strategically choose the best action based on current game state.
12. If your opponent get stunned, try powerful moves to maximize damage.
### Valid action set ###
 - Move Closer
 - Move Away
 - Fireball
 - Megapunch
 - Hurricane
 - Megafireball
 - Super attack 4
 - Low Punch
 - Medium Punch
 - High Punch
 - Low Kick
 - Medium Kick
 - High Kick
 - Low Punch+Low Kick
 - Medium Punch+Medium Kick
 - High Punch+High Kick
 - Jump Closer
 - Jump Away
### Decision Output Format ###
Analyze the provided game state and determine the **next actions** to take next.
Return your decision in the following exact format:
### Reasoning
<a detailed summary of why this action was chosen>
### Actions
<at most two consecutive actions in the valid action set>

Ensure that:
- The '### Reasoning' field provides a clear explanation of why the action is the best choice.
- The '### Actions' field contains only at most two of the valid actions.

**User prompt (Action Inference)**

### Current subtask
The current subtask is to close the distance by using a forward dash or advancing special move, then apply pressure with a mix of medium and heavy attacks to reduce the opponent's health.

### Last executed action:
- Medium Punch+Medium Kick
- High Punch+High Kick

### Self reflection of the last executed action
The attempted throw to break the opponent's guard and reduce their health was unsuccessful. The opponent's health remained unchanged, and the distance between the players increased, indicating that the opponent likely evaded or countered the throw.

### Current state:
You are playing Ken in Street Fighter 3. Your opponent is Yang.
You are facing left.
Distance from Opponent: close
Time Remaining: 95
Health:
    Your Health: 161
    Opponent's Health: 134
Super Bar Guage:
    Your Super Bar Guage: 32
    Opponent's Super Bar Guage: 9
Super Count (Count of activated super move):
    Your Super Count: 0
    Opponent's Super Count: 0
Stun Bar Guage:
    Your Stun Bar Guage: 0
    Opponent's Stun Bar Guage: 10
IsStunned (0: not stunned, 1: stunned):
    Your Status: 0
    Opponent's Special Status: 0

Based on the above information, analyze the current situation for what you should do for the next step. Then, you should output the exact actions you want to execute in the game.
You should only respond with actions from the valid action set.
You should only respond in the format described below, and you should not output comments or other information.
Provide your response in the strict format:
### Reasoning
- ...
- ...
- ...
### Actions
- ...
- ...

Figure 8: Action inference prompt for 'reflection-planning' agent playing *Street Fighter III*.

on the target task and the current game state. The user prompt includes: (1) the main goal of the game, (2) the previous subtask (generated by the recent planning module), (3) the last executed action, (4) a self-reflection on the last action (generated by the recent reflection module), (5) the current state, and (6) the expected output format for the subtask reasoning task.

**Action inference prompt.** As shown in Figure 8, the system prompt outlines strategic guidelines for playing `Ken`, a predefined set of valid actions, and the required output format. The user prompt contains: (1) the current subtask (provided by the recent planning module), (2) the last executed action, (3) the corresponding self-reflection (generated by the recent reflection module), (4) the current state, and (5) the expected output format for the action inference task.

**Reflection prompt.** As illustrated in Figure 9, the system prompt provides detailed instructions for an agent to perform reflection. The agent is required to analyze whether the last action was successful based on state transitions. The user prompt includes: (1) the target task, (2) the current subtask (generated by the recent planning module), (3) the last executed action, (4) the previous state, (5) the current state, and (6) the expected output format for the reflection task.

## C.3 EVALUATION METRIC FOR STREET FIGHTER III

**Single-Agent Play.** In the single-player mode, the agent faces a series of 10 stages against in-game rule-based bots. The game ends either when the player loses a stage or successfully clears all 10

**System prompt (Reflection)**

You are a helpful AI assistant integrated with 'Street Fighter III: 3rd Strike' on the PC, capable of analyzing in-game contexts and determining whether an executed action has taken effect. Your task is to evaluate the success of actions based on state changes and provide logical reasoning.
You need to answer the following questions step by step to derive reasoning based on the last action and the states.
1. What is the executed action and its desired result?
2. What is the difference between the two states? Compare every component.
3. Was the executed action successful? Provide reasoning.
4. (If the last action was not successful) What is the most probable cause? Give only one cause. You should summarize the reasoning in a clear and concise manner, providing a logical explanation for the success or failure of the last action.

**User prompt (Reflection)**

### Target task
Defeat the opponent

### Current subtask
The current subtask is to close the distance by using a forward dash or advancing special move, then apply pressure with a mix of medium and heavy attacks to reduce the opponent's health.

### Last executed action
- Megafireball
- Move Closer

### Previous state
You are playing Ken in Street Fighter 3. Your opponent is Yang.
You are facing left.
Distance from Opponent: close
Time Remaining: 95
Health:
    Your Health: 161
    Opponent's Health: 134
Super Bar Guage:
    Your Super Bar Guage: 32
    Opponent's Super Bar Guage: 9
Super Count (Count of activated super move):
    Your Super Count: 0
    Opponent's Super Count: 0
Stun Bar Guage:
    Your Stun Bar Guage:  0
    Opponent's Stun Bar Guage: 10
IsStunned (0: not stunned, 1: stunned):
    Your Status: 0
    Opponent's Special Status: 0

### Current state
You are playing Ken in Street Fighter 3. Your opponent is Yang.
You are facing left.
Distance from Opponent: close
Time Remaining: 94
Health:
    Your Health: 161
    Opponent's Health: 123
Super Bar Guage:
    Your Super Bar Guage: 35
    Opponent's Super Bar Guage: 9
Super Count (Count of activated super move):
    Your Super Count: 0
    Opponent's Super Count: 0
Stun Bar Guage:
    Your Stun Bar Guage:  0
    Opponent's Stun Bar Guage: 14
IsStunned (0: not stunned, 1: stunned):
    Your Status: 0
    Opponent's Special Status: 0

You should only respond in the format as described below.
### Self_reflection
1. ...
2. ...
3. ...
4. ...
### Self_reflection_summary
...

Figure 9: Reflection prompt for 'reflection-planning' agent playing *Street Fighter III*.

stages. Therefore, the evaluation metric can be straightforwardly defined as

$$\text{Score} = \text{Number of stages cleared by the agent} \times 10.$$

**Multi-Agent Play.** To evaluate models in a competitive multi-agent environment, we conduct pairwise matches between all agents and compute Elo ratings based on their win rates. For each pair, three games are played to obtain a reliable estimate of relative performance. The resulting win rate matrix and Elo scores are presented in Figure 4(a).

We adopt a Bradley-Terry model formulation (Bradley & Terry, 1952) for Elo estimation, where each model's rating is iteratively optimized using gradient ascent on the log-likelihood of observed outcomes. The gradient is computed based on the expected win probabilities derived from current ratings, using the standard Elo transformation:

$$P(i \text{ beats } j) = \frac{1}{1 + 10^{(R_j - R_i)/400}},$$

where $R_i$ and $R_j$ denote the Elo ratings of agent $i$ and agent $j$, respectively. The expected probability of $i$ defeating $j$ increases as the rating difference $R_i - R_j$ becomes larger. After optimization, we shift all Elo ratings so that their mean equals 1,500 for intuitive interpretation. If two models receive identical ratings, the one that won their head-to-head match is ranked higher.

### C.4 EXPERIMENTAL CONFIGURATION FOR STREET FIGHTER III

For all 6 open-source LLMs, including Llama-3.2-1B/3B, Qwen-2.5-3B/7B, and Minitron-4B/8B, we use a temperature of 0.0 and a repetition penalty of 1.0. During LLM inference, we pause the

| Models | SF3 | Rank |
|---|---|---|
| Random Agent | $10.0_{\pm 6.4}$ | 12 |
| Llama-3.2-1B | $0.0_{\pm 0.0}$ | 13 |
| Llama-3.2-3B | $13.3_{\pm 5.8}$ | 10.5 |
| Qwen-2.5-3B | $20.0_{\pm 0.0}$ | 4.5 |
| Qwen-2.5-7B | $16.7_{\pm 11.5}$ | 7.5 |
| Minitron-4B | $16.7_{\pm 11.5}$ | 7.5 |
| Minitron-8B | $23.3_{\pm 5.8}$ | 3 |
| GPT-4o-mini | $16.7_{\pm 11.5}$ | 7.5 |
| GPT-4o | $29.7_{\pm 14.3}$ | 2 |
| o3-mini | $\mathbf{33.3}_{\pm 15.3}$ | 1 |
| Gemini-2.5-pro | $13.3_{\pm 11.5}$ | 10.5 |
| Claude-3.7 | $16.7_{\pm 11.5}$ | 7.5 |
| Deepseek-R1 | $20.0_{\pm 0.0}$ | 4.5 |

Table 10: Gameplay score on *Street Fighter III*.

| Models | Agent | SF3 | Rank |
|---|---|---|---|
| Random Agent | - | $10.0_{\pm 6.4}$ | 9 |
| Llama-3B | Zero-shot | $13.3_{\pm 5.8}$ | 8 |
| | Reflection | $\mathbf{30.0}_{\pm 17.3}$ | 1.5 |
| | Planning | $20.0_{\pm 0.0}$ | 6 |
| | Ref-Plan | $16.7_{\pm 20.8}$ | 7 |
| GPT-4o | Zero-shot | $29.7_{\pm 14.3}$ | 3 |
| | Reflection | $23.3_{\pm 20.8}$ | 4.5 |
| | Planning | $\mathbf{30.0}_{\pm 26.5}$ | 1.5 |
| | Ref-Plan | $23.3_{\pm 20.8}$ | 4.5 |

Table 11: Ablation study for agentic modules on *Street Fighter III*.

| Models | Input | SF3 | Rank |
|---|---|---|---|
| Random Agent | - | $10.0_{\pm 6.4}$ | 10 |
| GPT-4o | Text | $\mathbf{29.7}_{\pm 14.3}$ | 2 |
| | Image | $23.7_{\pm 15.9}$ | 4 |
| | Both | $24.3_{\pm 14.5}$ | 3 |
| Gemini | Text | $13.3_{\pm 11.5}$ | 9 |
| | Image | $16.7_{\pm 11.5}$ | 7 |
| | Both | $20.0_{\pm 10.0}$ | 6 |
| Claude | Text | $16.7_{\pm 11.5}$ | 7 |
| | Image | $23.3_{\pm 11.5}$ | 5 |
| | Both | $\mathbf{33.3}_{\pm 5.8}$ | 1 |

Table 12: Comparison across modalities on *Street Fighter III*.

*Street Fighter III* game environment. We set the maximum number of game steps to 10,000. Due to the in-game time constraints, episodes do not reach the maximum of 10,000 steps. A single stage (best-of-three matches) is usually resolved within 100 to 200 steps, as rounds either timeout or end early when one player's health reaches zero.

## C.5  RESULT FOR STREET FIGHTER III

All reported results in Tables 10–13 are computed as the mean and standard deviation over five independent runs, using a 'zero-shot' agent for each model configuration. To establish a baseline for comparison, we additionally included a random agent that selects an arbitrary action uniformly at random. This agent was evaluated over 30 episodes, and its performance is summarized in tables.

**Single-Agent Play.** The performance across various LLMs was evaluated as presented in Table 10. The smallest model, Llama-3.2-1B, completely failed to comprehend the current game context and consistently ignore the required output format, performing even worse than a random agent. On the other hand, all other evaluated models surpassed the random agent, demonstrating varying levels of competence. Commercial LLMs generally outperformed their open-source counterparts, with GPT-4o and o3-mini standing out as notable examples.

To investigate the effectiveness of agentic modules, we conducted an ablation study shown in Table 11. For Llama-3.2-3B, the Reflection agent showed notably superior performance, indicating that reflective reasoning significantly aids in aligning actions to game dynamics, possibly by allowing the model to reassess and correct previous outputs based on feedback. Conversely, for GPT-4o, both the zero-shot and Planning agents performed remarkably well. This may suggest GPT-4o's inherent capability for generalization (zero-shot) and structured sequential reasoning (planning), enabling efficient decision-making without iterative reflection.

Input modality may significantly influence agent performance, since spatial information in fighting games is expecially important for gameplay. Since our implementation simplify character distance into three sparse levels, we expected image inputs would enhance spatial understanding and consequently improve performance. However, as shown in Table 12, GPT-4o surprisingly demonstrated decreased performance when using image inputs, possibly due to limited visual comprehension capabilities. In contrast, Gemini and Claude effectively leveraged visual data, improving their gameplay scores.

Lastly, Table 13 demonstrates intra-game generalization capabilities when fine-tuning models on `Ken`-specific gameplay data and evaluating on `Chun-Li` scenarios, which represent out-of-distribution conditions due to differences in action spaces (as previously detailed in Section C.1). Remarkably, the previously format-incompliant pretrained Llama-3.2-1B learned to follow the required output format effectively and exhibited excellent gameplay performance after fine-tuning, even in `Chun-Li` gameplay. A similar improvement was observed in Llama-3.2-3B, which even surpassed GPT-4o in performance. These results highlight the significant impact of targeted fine-tuning, demonstrating that relatively small-scale LLMs can achieve substantial gains in task-specific capability.

**Multi-Agent Play.** We evaluate agent performances in a multi-agent environment by conducting pairwise matches among 8 LLMs, all operating in a zero-shot setting. The results are provided in

| Model | Finetune | SF3 |
|-------|----------|-----|
| Llama-3.2-1B | ✗ | $0.0_{\pm 0.0}$ |
|  | ✓ | $\mathbf{42.0}_{\pm 16.4}$ |
| Llama-3.2-3B | ✗ | $12.0_{\pm 11.0}$ |
|  | ✓ | $\mathbf{40.0}_{\pm 7.1}$ |
| GPT-4o | ✗ | $10.0_{\pm 0.0}$ |

Table 13: Intra game generalization score of *Street Fighter III*.

Figure 4(a). Each pairwise matchup consisted of three independent games, with each game played in a best-of-three format to determine the winner. To ensure fair performance comparison, both agents used the same character—Ken—in all matches.

Notably, unlike the single-agent evaluation results in Table 10, Minitron-8B consistently outperforms all other models in the multi-agent arena and achieves the highest Elo rating. This divergence raises the possibility that the involvement of other intelligent agents could alter the game dynamics, perhaps due to increased strategic diversity or emergent adversarial behavior.

# D    SUPER MARIO

## D.1    GAME DESCRIPTION FOR SUPER MARIO

**Environment.**    *Super Mario* (1985 Super Mario Bros) (Christian Kauten, 2018) is a side-scrolling game where the player controls Mario to avoid obstacles, defeat monsters, and reach the flag. In this environment, Mario progresses through the game using directional key controls (*e.g.*, 'left' and 'right' keys) and jump actions. Mario should either destroy or traverse obstacles (*e.g.*, bricks, stairs, pipes), avoid or defeat monsters (*e.g.*, Goombas, Koopas) by jumping on them and avoid falling into pits. For implementation, we adopt the gym_super_mario_bros environment (Kauten, 2018), which is widely used in the reinforcement learning (RL) community. Specifically, we use the SuperMarioBros-v1 environment, where Mario plays within a 256×240-pixel screen with a black background, as shown in Figure 10(a). The environment consists of 8 worlds, each containing 4 stages. We use World 1, Stage 1 as our *default evaluation setting* (for Table 3). However, the environment supports evaluation across any world and stage. For example, we use World 3, Stage 1 for the evaluation of *intra-game generalization* (for Table 7).

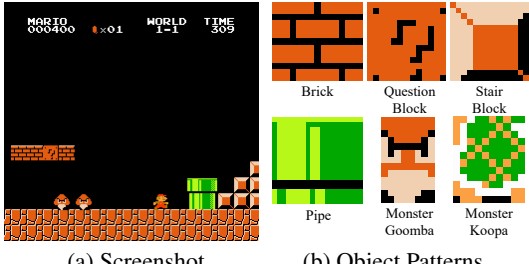

(a) Screenshot.        (b) Object Patterns.

Figure 10: Screenshot and assets of *Super Mario*.

**Observation-to-Text Conversion.** The environment only provides RGB image frames as observations. To convert visual game states into text input suitable for LLMs, we apply *visual pattern matching* to parse the exact location of each object on the frame. As shown in Figure 10(b), in the SuperMarioBros-v1 environment, the pixel-level visual patterns of objects remain stable across frames. By maintaining a set of pixel templates for all game objects as game assets, we perform 2D visual pattern matching to parse the presence and exact location of each object in the scene. These parsed object locations are converted to 2D coordinates $(x, y)$ and formatted as text, which is then passed to the LLM as part of its observation input.

**Action Space.** We constrain Mario to only move in the 'right' direction to simplify the control space. Mario can 'jump' at varying heights, and by constraining action spans 4 game frames (*i.e.*, frame skipping), Mario's jumping ability is discretized into 7 levels. Jump Level 0 corresponds to walking forward without jumping, while Jump Levels 1 through 6 represent increasing jump heights, with Level 6 being the highest possible jump. At each game step, the LLM chooses a jump level from 0 to 6, determining the jump height as Mario moves to the right.

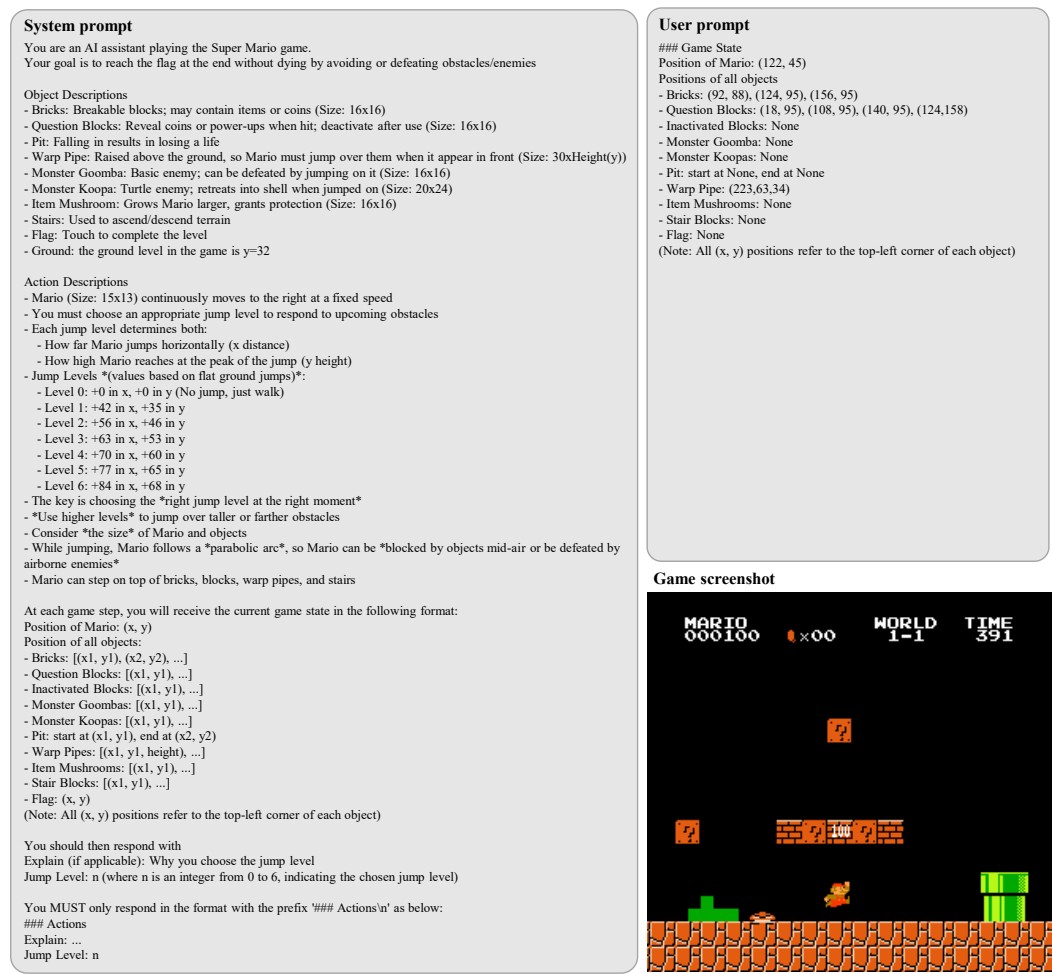

**System prompt**

You are an AI assistant playing the Super Mario game.
Your goal is to reach the flag at the end without dying by avoiding or defeating obstacles/enemies

Object Descriptions
- Bricks: Breakable blocks; may contain items or coins (Size: 16x16)
- Question Blocks: Reveal coins or power-ups when hit; deactivate after use (Size: 16x16)
- Pit: Falling in results in losing a life
- Warp Pipe: Raised above the ground, so Mario must jump over them when it appear in front (Size: 30xHeight(y))
- Monster Goomba: Basic enemy; can be defeated by jumping on it (Size: 16x16)
- Monster Koopa: Turtle enemy; retreats into shell when jumped on (Size: 20x24)
- Item Mushroom: Grows Mario larger, grants protection (Size: 16x16)
- Stairs: Used to ascend/descend terrain
- Flag: Touch to complete the level
- Ground: the ground level in the game is y=32

Action Descriptions
- Mario (Size: 15x13) continuously moves to the right at a fixed speed
- You must choose an appropriate jump level to respond to upcoming obstacles
- Each jump level determines both:
    - How far Mario jumps horizontally (x distance)
    - How high Mario reaches at the peak of the jump (y height)
- Jump Levels *(values based on flat ground jumps)*:
    - Level 0: +0 in x, +0 in y (No jump, just walk)
    - Level 1: +42 in x, +35 in y
    - Level 2: +56 in x, +46 in y
    - Level 3: +63 in x, +53 in y
    - Level 4: +70 in x, +60 in y
    - Level 5: +77 in x, +65 in y
    - Level 6: +84 in x, +68 in y
- The key is choosing the *right jump level at the right moment*
- *Use higher levels* to jump over taller or farther obstacles
- Consider *the size* of Mario and objects
- While jumping, Mario follows a *parabolic arc*, so Mario can be *blocked by objects mid-air or be defeated by airborne enemies*
- Mario can step on top of bricks, blocks, warp pipes, and stairs

At each game step, you will receive the current game state in the following format:
Position of Mario: (x, y)
Position of all objects:
- Bricks: [(x1, y1), (x2, y2), ...]
- Question Blocks: [(x1, y1), ...]
- Inactivated Blocks: [(x1, y1), ...]
- Monster Goombas: [(x1, y1), ...]
- Monster Koopas: [(x1, y1), ...]
- Pit: start at (x1, y1), end at (x2, y2)
- Warp Pipes: [(x1, y1, height), ...]
- Item Mushrooms: [(x1, y1), ...]
- Stair Blocks: [(x1, y1), ...]
- Flag: (x, y)
(Note: All (x, y) positions refer to the top-left corner of each object)

You should then respond with
Explain (if applicable): Why you choose the jump level
Jump Level: n (where n is an integer from 0 to 6, indicating the chosen jump level)

You MUST only respond in the format with the prefix '### Actions\n' as below:
### Actions
Explain: ...
Jump Level: n

**User prompt**

### Game State
Position of Mario: (122, 45)
Positions of all objects
- Bricks: (92, 88), (124, 95), (156, 95)
- Question Blocks: (18, 95), (108, 95), (140, 95), (124,158)
- Inactivated Blocks: None
- Monster Goomba: None
- Monster Koopas: None
- Pit: start at None, end at None
- Warp Pipe: (223,63,34)
- Item Mushrooms: None
- Stair Blocks: None
- Flag: None
(Note: All (x, y) positions refer to the top-left corner of each object)

**Game screenshot**

Figure 11: Action inference prompt for 'zero-shot' agent playing *Super Mario*.

## D.2 GAMEPLAY PROMPT FOR SUPER MARIO

Figure 11 shows the action inference prompt used by the 'zero-shot' agent for playing *Super Mario*. The system prompt contains most of the gameplay-specific knowledge. It includes (1) the main goal of the game, (2) detailed descriptions and sizes of each object, (3) explanations and safety notes for each action, and (4) the expected input-output format between the LLM and the environment. The user prompt provides the current game state as a list of all objects detected in the frame, represented by their top-left corner (x, y) coordinates obtained by visual pattern matching. Given this prompt, the LLM agent infers the appropriate jump level to advance safely toward the flag by avoiding obstacles and monsters.

## D.3 EVALUATION METRIC FOR SUPER MARIO

The goal of *Super Mario* is to reach the flag located at the right end of the stage. Since the gym_super_mario_bros environment provides Mario's current position on the map, we define the evaluation metric as the proportion of the distance traversed toward the flag before Mario dies. Formally, the normalized score is defined as:

$$\text{Score} = dist(x_{Mario}, x_{start})/dist(x_{flag}, x_{start}) \times 100,$$

where $x_{Mario}$, $x_{flag}$, and $x_{start}$ are the $x$ coordinate of Mario traversed before die, that of the flag, and that of the starting position on the map, respectively.

| Models | SuperMario | Rank |
|---|---|---|
| Llama-3.2-1B | $18.7_{\pm 8.6}$ | 12 |
| Llama-3.2-3B | $31.8_{\pm 10.1}$ | 4 |
| Qwen-2.5-3B | $23.4_{\pm 14.1}$ | 11 |
| Qwen-2.5-7B | $27.2_{\pm 9.6}$ | 9 |
| Minitron-4B | $24.4_{\pm 6.0}$ | 10 |
| Minitron-8B | $31.3_{\pm 12.8}$ | 6 |
| GPT-4o-mini | $28.8_{\pm 8.8}$ | 7 |
| GPT-4o | $34.1_{\pm 14.2}$ | 3 |
| o3-mini | $34.9_{\pm 14.6}$ | 2 |
| Gemini-2.5-pro | $\mathbf{38.0}_{\pm 14.6}$ | 1 |
| Claude-3.7 | $31.7_{\pm 8.2}$ | 5 |
| Deepseek-R1 | $28.7_{\pm 13.2}$ | 8 |

Table 14: Gameplay score on *Super Mario*.

| Models | Agent | SuperMario | Rank |
|---|---|---|---|
| Llama-3B | Zero-shot | $21.2_{\pm 8.2}$ | 8 |
| | Reflection | $32.4_{\pm 8.6}$ | 2 |
| | Planning | $27.0_{\pm 8.4}$ | 7 |
| | Ref-Plan | $31.8_{\pm 10.1}$ | 4 |
| GPT-4o | Zero-shot | $29.6_{\pm 9.2}$ | 5 |
| | Reflection | $32.3_{\pm 15.5}$ | 3 |
| | Planning | $29.4_{\pm 12.5}$ | 6 |
| | Ref-Plan | $\mathbf{34.1}_{\pm 14.2}$ | 1 |

Table 15: Ablation study for agentic modules on *Super Mario*.

| Models | Input | SuperMario | Rank |
|---|---|---|---|
| GPT-4o | Text | $34.1_{\pm 14.1}$ | 3 |
| | Image | $27.1_{\pm 13.7}$ | 6.5 |
| | Both | $27.1_{\pm 10.2}$ | 6.5 |
| Gemini | Text | $38.0_{\pm 13.4}$ | 2 |
| | Image | $28.5_{\pm 10.7}$ | 5 |
| | Both | $\mathbf{40.9}_{\pm 9.6}$ | 1 |
| Claude | Text | $28.7_{\pm 13.2}$ | 4 |
| | Image | $25.6_{\pm 6.4}$ | 8 |
| | Both | $22.6_{\pm 6.3}$ | 9 |

Table 16: Comparison across modalities on *Super Mario*.

### D.4 EXPERIMENTAL CONFIGURATION FOR SUPER MARIO

For all 6 open-source LLMs, including Llama-3.2-1B/3B, Qwen-2.5-3B/7B, and Minitron-4B/8B, we use a temperature of 1 and a repetition penalty of 1. During LLM inference, we pause the *Super Mario* game environment. We set the maximum number of game steps to 100. We run all experiments with 20 trials and report the average score with the standard deviation.

### D.5 RESULT FOR SUPER MARIO

As shown in Table 14, Gemini-2.5-pro achieves the highest score of 38.0 on *Super Mario*, followed by o3-mini with the score of 34.9. Among open-source LLMs, Llama-3.2-3B and Minitron-8B perform competitively, achieving scores of 31.8 and 31.3 respectively, which are comparable to the 31.7 score of Claude-3.7-sonnet. Qualitatively, most LLMs, including Gemini-2.5-pro, frequently fail to correctly estimate the *parabolic* jump trajectory when Mario is mid-air. This change of moves often results in Mario colliding with obstacles or being killed by monsters.

Table 15 shows the effect of reflection and planning modules on gameplay performance. Among these, the reflection module has a more pronounced impact. Specifically, when Mario is stuck in front of high obstacles such as warp pipes, the reflection module enables the agent to revise its previous low jump level decisions and select a higher jump level, allowing it to overcome the obstacle and proceed, thereby improving the final score. GPT-4o achieves the best score of 34.1 when both reflection and planning modules are used. In contrast, Llama-3.2-3B performs best when only the reflection module is used with a score of 32.4, indicating that Llama-3.2-3B may not benefit from the additional planning module or be easily disturbed by its response.

As shown in Table 16, using image-only input observations consistently underperforms compared to text-only inputs across all models, including GPT-4o, Gemini-2.5-pro, and Claude-3.7-sonnet. This suggests that relying solely on visual input makes it more challenging for models to extract detailed information from the game scene or to perform spatial reasoning, *i.e.*, estimating distances between objects. In contrast, when both text and image inputs are provided, GPT-4o and Claude-3.7-sonnet show a performance drop, while Gemini-2.5-pro shows improved performance. This indicates that multimodal input can be beneficial when the model effectively integrates complementary information from both modalities.

## E ACE ATTORNEY

### E.1 GAME DESCRIPTION FOR ACE ATTORNEY

**Environment.** *Ace Attorney* (Capcom, 2001) is a courtroom adventure game where players act as defense attorneys, gather evidence, and cross-examine witnesses. We target the first episode of *Phoenix Wright: Ace Attorney Trilogy* on *Steam* (see Figure 12). We define four subtasks: one three-question multiple-choice quiz where the player selects the correct answer, and three cross-examination tasks where the player presses witnesses for more details

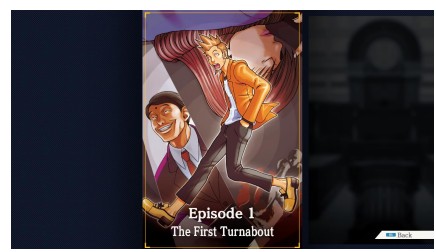

Figure 12: Screenshot of Episode 1: The First Turnabout.

**System prompt**

You are an AI defense attorney in an interactive Ace Attorney-style trial. The game advances screen-by-screen based on your choices, and your goal is to win by managing dialogue and evidence effectively. **ONLY** perform actions permitted by the currently visible screen.

Responsibilities:
- Monitor dialogue for cues to review evidence or profiles.
- Choose the best options in multiple-choice scenarios.
- Cross-examine witnesses to detect contradictions and present evidence.

Gameplay Guidelines:
- Press "Ok" to continue dialogue and "Tab" to access the Court Record.
- **ONLY** access the Court Record when absolutely necessary: if the "Last Court Record" is None or if the "Last Check Time" is significantly outdated relative to the current dialogue.
- All actions must be based solely on the on-screen dialogue. The on-screen dialogue is defined as the very last entry in the "Current State"'s [Recent Conversations], which is marked as [**The Conversation Currently on Screen**].
- There are two types of important dialogue: (1) regular dialogue (with no color formatting) and (2) testimony for Cross-Examination, displayed in green (color=#00f000).
- The final goal of the game is to identify contradictions between the on-screen testimony and the Court Record, and to present evidence proving that the false testimony is being shown.

**IMPORTANT (Cross-Examination Eligibility):** You may perform Cross-Examination actions ("Press at the moment of testimony" or "Present the selected evidence") only when both conditions below are met:
   1. The testimony is displayed in color=#00f000 and the "Current State" includes **Cross-Examination!**
   2. The most recent testimony (marked as [**The Conversation Currently on Screen**]) clearly relates to a contradiction you have either suspended or confirmed.

**IMPORTANT (Action Strategy):** When both Cross-Examination Eligibility conditions are satisfied, use either of the following two Cross-Examination actions: If you need additional hints or clarification, press "Press at the moment of testimony" (represented by "Hold it!" in [Recent Conversations]). However, if you are confident and ready to expose false testimony, wait until the contradictory on-screen testimony appears, then press "Tab", select the appropriate evidence, and execute "Present the selected evidence" (represented by "Objection!" in [Recent Conversations]).
- **DO NOT** use these actions for merely suspicious or ambiguous discrepancies. Trigger them only when there is a definitive contradiction—such as when the testimony directly and logically conflicts with the actual record.
- **DO NOT** repeat actions that are already recorded in "Last Decisions" on the same on-screen testimony.
- Only the on-screen testimony ([**The Conversation Currently on Screen**]) can trigger the actions. Even if your analysis or long-term memory indicates a contradiction, continue pressing "Ok" until the corresponding testimony appears on the screen.
- To return to a previous testimony and display it on screen, press the "Left" key.

Additional Notes:
- Constantly assess the dialogue for cues and adapt your strategy as new evidence emerges.
- Remember that not every piece of testimony contains a contradiction; only initiate cross-examination when there is clear and definitive evidence of inconsistency.
- **IMPORTANT:** Only select an action from the candidate list by responding solely with the **INTEGER** number corresponding to the selected option.

**User prompt**

Current State:
[Recent Conversations]
[2025-04-02 17:18:30] Alias: Yes! Er... yes, Your Honor?
...
[**The Conversation Currently on Screen** - 2025-05-23 12:44:13]
Bravo: Open the Court Record with <color=#ff0000>      </color>, then point out <color=#ff0000>contradictions</color> in the testimony!

Last Court Record:
**Last Check Time**: 2025-05-23 12:44:13
**Court Record - Evidence**:
1: Attorney's Badge - No one would believe I was a defense attorney if I didn't carry this.
...
5: Blackout Record - Electricity to Ms. Foxtrot's building was out from noon to 6PM on the day of the crime.
**Court Record - Profile**:
1: Bravo (Age: 27) - Chief Attorney at Bravo & Co. My boss, and a very good defense attorney.
...
5: Echo (Age: 36) - Discovered Ms. Foxtrot's body. Newspaper salesman who saw Delta flee the scene.

Last Decisions:
None

**Possible Options** (Active Key Types):
1: Ok
2: Tab

Please respond using the following format:
### Reasoning
[Your step-by-step reasoning here.]

### Actions
[**ONLY** output the **INTEGER** number corresponding to the correct option from the **Possible Options**.]

**Game screenshot**

Figure 13: Action inference prompt for 'zero-shot' agent playing *Ace Attorney*.

or presents evidence to expose contradictions. We use Harmony (Pardeike, 2025) with a BepInEx plugin (Contributors, 2025a) to hook the game's source code at launch, capturing states such as dialogue text, arrow-button visibility, keyboard inputs, and Court Record entries. The hooks save states as `.txt` or `.json` files and monitor a command `.txt` file for inputs, injecting them into the game in real time.

**Observation-to-Text Conversion.** All states remain in text form and require only minimal post-processing. We map speaker indices to character names (*e.g.*, index '2' → 'Phoenix Wright') using a predefined mapping and replace original names with arbitrary aliases to prevent contamination (*e.g.*, 'Phoenix Wright' → 'Alias'). We also convert Court Records and multiple-choice candidate options—originally stored in `.json`—into continuous descriptive text when they exist.

**Action Space.** The basic actions include pressing 'Ok' to progress the dialogue and 'Tab' to access the Court Records. For multiple-choice questions, the action space expands to include candidate options. During cross-examinations, additional actions become available: 'Left' to return to the previous dialogue, 'Press' to question the witness further, selecting relevant evidence from the Court Records, and 'Present' to introduce it during the examination. Each option carries an index, and the model returns only the corresponding integer.

## E.2    GAMEPLAY PROMPT FOR ACE ATTORNEY

Figure 13 presents the zero-shot agent's action-inference prompt for *Ace Attorney*. The system prompt specifies (1) the game's goal, (2) procedures for dialogue, multiple-choice questions, and cross-examinations, (3) rules for accessing Court Records and presenting evidence, and (4) the expected I/O format between the LLM and the environment. The user prompt lists recent conver-

| Models | AceAttorney | Rank |
|---|---|---|
| Llama-3.2-1B | $1.3_{\pm2.2}$ | 12 |
| Llama-3.2-3B | $4.6_{\pm1.3}$ | 11 |
| Qwen-2.5-3B | $20.0_{\pm17.4}$ | 9 |
| Qwen-2.5-7B | $9.3_{\pm0.2}$ | 10 |
| Minitron-4B | $35.7_{\pm4.5}$ | 6 |
| Minitron-8B | $29.9_{\pm3.6}$ | 7 |
| GPT-4o-mini | $28.4_{\pm2.8}$ | 8 |
| GPT-4o | $85.3_{\pm1.5}$ | 2 |
| o3-mini | $\mathbf{91.7}_{\pm1.5}$ | 1 |
| Gemini-2.5-pro | $55.7_{\pm3.4}$ | 5 |
| Claude-3.7 | $81.9_{\pm1.6}$ | 4 |
| Deepseek-R1 | $83.3_{\pm1.5}$ | 3 |

Table 17: Gameplay score on *Ace Attorney*.

| Models | Agent | AceAttorney | Rank |
|---|---|---|---|
| Llama-3B | Zero-shot | $5.7_{\pm3.2}$ | 5 |
| | Reflection | $4.6_{\pm1.3}$ | 6.5 |
| | Planning | $4.6_{\pm1.3}$ | 6.5 |
| | Ref-Plan | $3.8_{\pm0.0}$ | 8 |
| GPT-4o | Zero-shot | $49.9_{\pm1.3}$ | 4 |
| | Reflection | $\mathbf{85.3}_{\pm1.5}$ | 1 |
| | Planning | $52.7_{\pm0.5}$ | 3 |
| | Ref-Plan | $52.8_{\pm0.5}$ | 2 |

Table 18: Ablation study for agentic modules on *Ace Attorney*.

| Models | Input | AceAttorney | Rank |
|---|---|---|---|
| GPT-4o | Text | $\mathbf{85.3}_{\pm1.5}$ | 1 |
| | Both | $53.5_{\pm1.7}$ | 5 |
| Gemini | Text | $55.7_{\pm3.4}$ | 4 |
| | Both | $52.6_{\pm0.8}$ | 6 |
| Claude | Text | $\mathbf{81.9}_{\pm1.6}$ | 2 |
| | Both | $71.3_{\pm17.3}$ | 3 |

Table 19: Comparison across modalities on *Ace Attorney*.

sations—highlighting the conversation currently visible on the screen—and provides timestamped Court Record entries. Using this prompt, the agent detects contradictions, selects the correct actions, and manages dialogue and evidence to advance the trial.

### E.3 EVALUATION METRIC FOR ACE ATTORNEY

Each subtask begins and ends at fixed points, with screenshots at both start and end and the preceding conversation history provided, and yields a reward $r_i$ and a step count $t_i$. In the multiple-choice task (MC), $r_i$ is the number of correct answers out of three, while in cross-examination tasks CE1, CE2, and CE3, $r_i$ is 1 for a pass and 0 for a fail; any failure incurs a maximum step count of 50. We normalize each reward and step count against fixed benchmarks $\bar{r}_i$ and $\bar{t}_i$, namely $(3, 25)$ for MC, $(1, 11)$ for CE1, $(1, 3)$ for CE2, and $(1, 4)$ for CE3, by computing

$$p_i = \frac{r_i}{\bar{r}_i}, \quad s_i = \frac{\bar{t}_i}{t_i},$$

and assign each task a difficulty weight $w_i \in \{1, 4, 2, 3\}$ reflecting MC, CE1, CE2, and CE3 respectively. These weights were set proportional to each task's difficulty and the minimum number of steps required (the benchmarks). We then form weighted averages

$$A = \frac{\sum_i w_i\, p_i}{\sum_i w_i}, \quad B = \frac{\sum_i w_i\, s_i}{\sum_i w_i},$$

and combine them into a composite score

$$\text{Score} = 100\big(\alpha\, A + (1 - \alpha)\, B\big),$$

with $\alpha = 0.7$ (70% for accuracy, 30% for efficiency). All experiments were repeated three times, and we report the sample mean and standard deviation of score.

### E.4 EXPERIMENTAL CONFIGURATION FOR ACE ATTORNEY

For all six open-source LLMs (Llama-3.2-1B/3B, Qwen-2.5-3B/7B, and Minitron-4B/8B), we set the temperature to 0.7, apply a repetition penalty of 1, cap game steps at 50, and limit conversation history to the most recent 20 exchanges.

### E.5 RESULT FOR ACE ATTORNEY

As shown in Table 17, o3-mini tops the leaderboard with a score of 91.7, followed by GPT-4o with a score of 85.3, Deepseek-R1 with a score of 83.3, and Claude-3.7-sonnet with a score of 81.9. Among open-source models, Gemini-2.5-pro and Minitron-4B score 55.7 and 35.7, respectively, while smaller Llama and Qwen variants remain below 20. Notably, Qwen-2.5-3B completes the first cross-examination in one out of three trials, where other open-source models struggle. Models scoring around 30 often misinterpret key details and repeat the same mistakes until they reach the maximum number of steps. Models below 20 sometimes reason correctly but mostly select irrelevant options, whereas the lowest performers fail to follow the required format and cannot advance further.

Table 18 compares our reflection and planning modules. The base Llama-3.2-3B model scores 5.7; adding reflection or planning alone yields approximately 4.6, and combining both drops the

score to 3.8. Smaller models like Llama-3.2-3B often focus on incorrect details and produce flawed reasoning when reflection or planning is faulty. GPT-4o scores highest with reflection alone (85.3), since reflection prevents repeated errors and helps spot contradictions. Planning alone adds little, and combining it with reflection lowers the score to approximately 52.7, as it attempts to resolve contradictions not visible on-screen.

Table 19 compares input modalities. GPT-4o and Gemini-2.5-pro score 85.3 vs. 53.5 and 55.7 vs. 52.6 for text-only vs. multimodal inputs, respectively, while Claude-3.7-sonnet falls from 81.9 to 71.3. Because we supply complete text descriptions of every on-screen element and its context, adding visual input brings no improvement—confirming that text alone suffices to play *Ace Attorney*.

# F    HER STORY

## F.1    GAME DESCRIPTION FOR HER STORY

**Environment.** *Her Story* (Barlow, 2015) is an interactive adventure game where players explore police interview clips to uncover a hidden truth. The player begins the game by accessing an old desktop interface, where a program called `L.O.G.I.C. Database` is open. The player can enter keywords into the database to retrieve up to five video clips whose transcripts contain the searched word. By watching these clips and gathering clues, the player repeatedly formulates new queries to reconstruct the underlying story. To interface the game with our code, we use Harmony (Pardeike, 2025) together with Unity Doorstop (Contributors, 2025b) to log internal game states to a `.txt` file. Specifically, each line in the file is a JSON object representing a snapshot of the game's state at a given moment. Each object contains an event type and its associated metadata. The following examples illustrate key elements of the logged game state. For clarity, we omit auxiliary metadata that are present in the actual logs but are only used for debugging or UI-related purposes, such as video IDs, screen resolution, and UI element positions.

- `Load title screen`: {"status": "title"}
- `Load L.O.G.I.C. Database`: {"status": "start_game"}
- `Query`: {"status": "query", "keyword": *keyword*}
- `Get query result`: {"status": "query_result", "num_total": *number of clips containing the keyword*, "num_visible": *number of clips shown*, "video": *list of* {"new": *1 if not viewed, otherwise 0*, "session": *recording date*, "outfit": *visual description of the thumbnail*}}
- `Open video panel`: {"status": "open_detail"}
- `Close video panel`: {"status": "close_detail"}
- `Play video`: {"status": "play_video", "script": *transcript of the video*}
- `Close video`: {"status": "close_video"}

**Observation-to-Text Conversion.** We aggregate the game states and convert them into textual observations. Each observation includes summary information such as the number of clips containing the keyword and the number of clips shown. It also contains per-clip metadata, including the recording date, thumbnail description, viewing status, and the transcript if the clip was viewed after the query.

**Action Space.** The original *Her Story* game is designed as a point-and-click interface, where the player interacts with the game by typing keywords into a search bar, clicking on retrieved clip thumbnails, navigating panels, and controlling playback. These interactions rely on low-level input mechanisms such as mouse movements and keyboard input. To reduce complexity, we abstract these low-level interactions into two high-level actions: searching with a keyword and playing the retrieved video clip.

- `Search [keyword]`: Returns all video clips whose transcripts contain the exact word. It consists of three low-level GUI actions: (1) clicking the search bar, (2) typing the keyword, and (3) pressing Enter to submit the query.
- `Play Video [i]`: Plays the $i$-th video from the current search result list. It consists of four low-level GUI actions: (1) clicking the thumbnail of the video clip to open the panel, (2) clicking the thumbnail within the panel to start playback, (3) either waiting until the video finishes or pressing the ESC key to exit playback, and (4) clicking the Exit button to close the panel.

## F.2 GAMEPLAY PROMPT FOR HER STORY

Figure 14 shows the action inference prompts used by the 'zero-shot' agent to play *Her Story*. The system prompt provides instructions covering: (1) the main goal of the game, (2) the type of information each video clip may contain, (3) behavioral rules the agent should follow—such as avoiding repeated keywords in the search history—and (4) the expected input-output format between the LLM and the environment. The user prompt includes: (1) the last executed action, (2) the current game state, and (3) the search history.

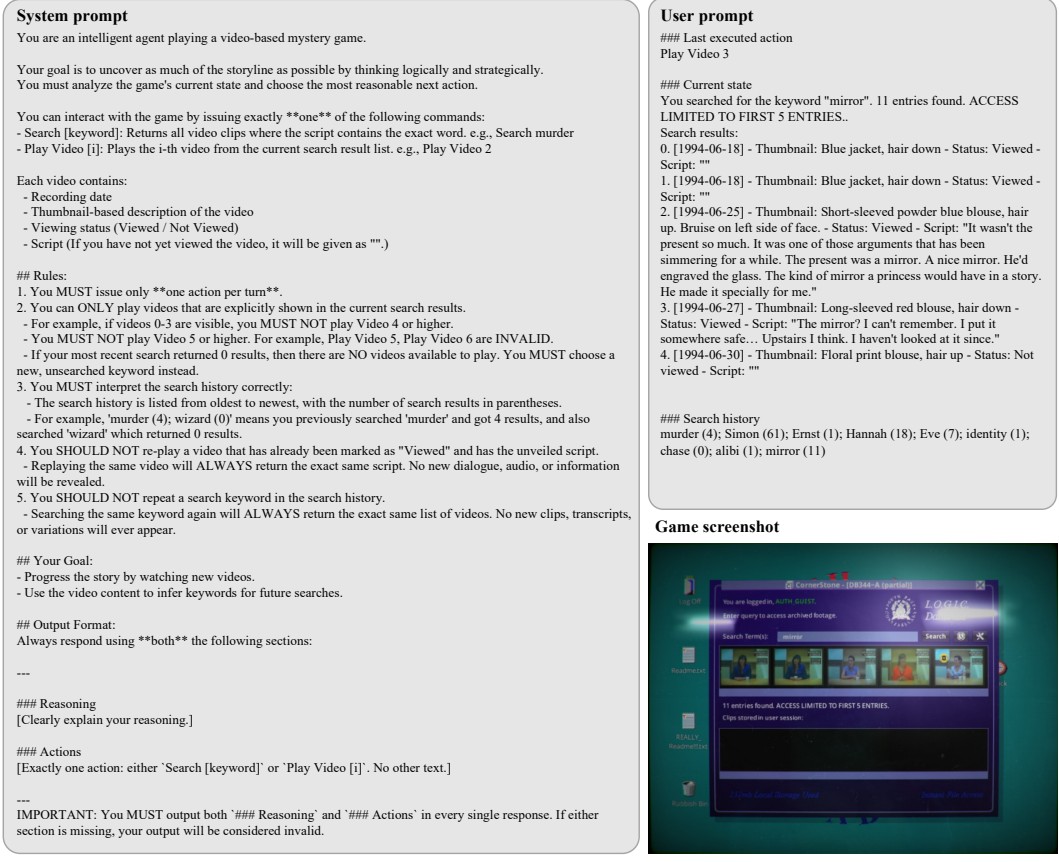

Figure 14: Action inference prompt for 'zero-shot' agent playing *Her Story*.

## F.3 EVALUATION METRIC FOR HER STORY

To uncover the truth behind the case, the player actively explores the video archive by issuing queries and watching clips. We evaluate this behavior by counting how many unique clips the player has viewed from the full archive of 271 clips. This aligns with the original game design, where unlocking achievements and reaching the ending depend on the number of clips watched. Specifically, we define the score as:

$$\text{Score} = \left( \frac{\text{Number of distinct video clips viewed}}{271} \right) \times 100.$$

## F.4 EXPERIMENTAL CONFIGURATION FOR HER STORY

For all LLMs, we use a temperature of 0.3 and a repetition penalty of 1. We limit the maximum number of interactions with the game environment to 400 steps. We run each experiment three times and report the average score along with the standard deviation.

| Models | HerStory | Rank |
|---|---|---|
| Llama-3.2-1B | $2.1_{\pm1.2}$ | 11 |
| Llama-3.2-3B | $4.2_{\pm1.1}$ | 10 |
| Qwen-2.5-3B | $1.2_{\pm1.1}$ | 12 |
| Qwen-2.5-7B | $8.5_{\pm2.0}$ | 7 |
| Minitron-4B | $4.6_{\pm2.3}$ | 9 |
| Minitron-8B | $8.2_{\pm1.8}$ | 8 |
| GPT-4o-mini | $21.2_{\pm5.6}$ | 6 |
| GPT-4o | $64.2_{\pm5.2}$ | 4 |
| o3-mini | $66.5_{\pm3.6}$ | 3 |
| Gemini-2.5-pro | $\mathbf{67.5}_{\pm3.3}$ | 1 |
| Claude-3.7 | $62.9_{\pm2.6}$ | 5 |
| Deepseek-R1 | $67.2_{\pm3.9}$ | 2 |

Table 20: Gameplay score on *Her Story*.

| Models | Agent | HerStory | Rank |
|---|---|---|---|
| Llama-3B | Zero-shot | $4.2_{\pm1.1}$ | 8 |
| | Reflection | $4.4_{\pm1.3}$ | 7 |
| | Planning | $5.2_{\pm1.0}$ | 6 |
| | Ref-Plan | $5.4_{\pm0.4}$ | 5 |
| GPT-4o | Zero-shot | $\mathbf{64.2}_{\pm5.2}$ | 1 |
| | Reflection | $61.5_{\pm0.4}$ | 3 |
| | Planning | $59.4_{\pm5.7}$ | 4 |
| | Ref-Plan | $62.0_{\pm4.9}$ | 2 |

Table 21: Ablation study for agentic modules on *Her Story*.

| Models | Input | HerStory | Rank |
|---|---|---|---|
| GPT-4o | Text | $64.2_{\pm5.2}$ | 3 |
| | Both | $40.6_{\pm29.5}$ | 6 |
| Gemini | Text | $\mathbf{67.5}_{\pm3.3}$ | 1 |
| | Both | $64.9_{\pm2.4}$ | 2 |
| Claude | Text | $62.9_{\pm2.6}$ | 5 |
| | Both | $63.6_{\pm3.1}$ | 4 |

Table 22: Comparison across modalities on *Her Story*.

## F.5 RESULT FOR HER STORY

Table 20 summarizes the gameplay scores of different LLMs on *Her Story*. Commercial LLMs outperform open-source models in this task. Gemini-2.5-pro achieves the highest score of 67.3, followed by Deepseek-R1 (66.9), o3-mini (66.3), GPT-4o (64.0), Claude-3.7-sonnet (62.6), and GPT-4o-mini (21.1). The low score of GPT-4o-mini is due to its repeated search of keywords such as `alibi`, `evidence`, and `witness`, even after those queries have already been issued. This redundancy reduces its ability to discover new clips. Among open-source LLMs, all models score below 10. Most exhibited similar failure patterns: they fail to play unseen clips, repeatedly issue the same keywords, or search using entire sentences (e.g., *for the name "Simon"*, *for keywords related to Luna's*) rather than meaningful single words.

Table 21 reports the effect of incorporating reflection and planning modules on gameplay performance. For Llama-3.2-3B, the 'reflection-planning' agent achieves the best performance, followed by 'planning', 'reflection', and 'zero-shot' agents. This suggests that such modules can improve performance for weaker base models. In contrast, for GPT-4o, the 'zero-shot' agent achieves the highest score, and adding reflection or planning modules does not lead to further improvements. These modules tend to produce information that may be redundant for stronger models like GPT-4o, which can already generate effective queries without additional reasoning support.

Table 22 compares gameplay performance across different input modalities. In our multimodal setup, the visual input corresponds to the main screen of the `L.O.G.I.C. Database` interface, as shown in Figure 14. For GPT-4o and Gemini-2.5-pro, using both text and image inputs underperforms compared to the text-only setting. In particular, for GPT-4o, performance varies significantly depending on the random seed when using multimodal inputs. We observe issues such as repeated use of the same keywords and the inclusion of quotation marks around keywords (e.g., `"murder"`), which often lead to failed queries. In contrast, Claude-3.7-sonnet shows slightly better performance in the multimodal setting. We speculate two reasons why the visual input fails to improve performance in most cases: (1) most of the useful visual elements (e.g., clip lists, thumbnails) are already represented in the text observation, and (2) additional information that visual input could provide—such as the interviewee's gestures or expressions—is only available when the video is actually played, and thus not present in the static visual input used in our setup.

## G POKÉMON RED

### G.1 GAME DESCRIPTION FOR POKÉMON RED

**Environment.** *Pokémon Red* (Game Freak, 1996) is a role-playing game where the player navigates the Kanto region to catch and train creatures called Pokémon, battle other trainers, and ultimately defeat the Elite Four and the Champion. The player explores various environments, including towns, routes, caves, and buildings, encountering wild Pokémon and other characters. The gameplay loop involves exploring these areas, engaging in turn-based battles with Pokémon, and managing a team of up to six Pokémon. For implementation, we utilize the PyBoy (Baekalfen) emulator to run the game. Specifically, our evaluation focuses on a segment where the player starts in Pallet Town

and progresses towards Viridian City, encountering wild Pokémon and trainers. The game screen resolution is 160×144 pixels.

**Observation-to-Text Conversion.** Instead of relying on visual pattern matching, we directly access the game's internal memory via the PyBoy emulator to extract relevant game state information. This includes detailed map information, the player's current coordinates, information about the player's party (e.g., Pokémon, their HP), encountered opponent Pokémon information, and the player's inventory of items. This rich set of information is then formatted as text to serve as the observation input for the LLM.

**Action Space.** The fundamental action space consists of the Game Boy buttons: 'up', 'down', 'left', 'right', 'a', 'b', 'start', and 'select'. Additionally, we define a set of higher-level tools to facilitate more complex interactions:

- `move_to(x, y)`: Finds and executes a path to the specified map coordinates $(x, y)$.
- `interact_with_object(object_name)`: Interacts with a specified object in the environment.
- `warp_with_warp_point(warp_point_coord)`: Uses a specified warp point to move to a different location.
- `overworld_map_transition(direction)`: Transitions to an adjacent map in the given direction.
- `continue_dialog()`: Advances the current dialogue.
- `select_move_in_battle(move_name)`: Selects and uses a specific move in a Pokémon battle.
- `switch_pkmn_in_battle(pkmn_name)`: Switches to a different Pokémon in the player's party during a battle.
- `run_away()`: Attempts to flee from a wild Pokémon battle.
- `use_item_in_battle(item_name)`: Uses a specified item during a Pokémon battle.

At each step, the LLM can choose up to five consecutive fundamental actions or invoke one of the provided tools.

### G.2 GAMEPLAY PROMPT FOR POKÉMON RED

Figure 15 shows the system prompt used by the agent for playing *Pokémon Red*, which provides the LLM with the necessary game rules, action space details (including basic controls and available tools), information about different game states, and the expected input/output format. The system prompt guides the LLM on how to interpret the game state and decide on the next action or tool to use to achieve the overarching goals of becoming the Champion and completing the Pokédex.

Figure 16 illustrates an example of the user prompt provided to the LLM. This includes the recent history of actions and their outcomes, the current game state (map information, player position, inventory, party, screen text, etc.), any recent critique on the agent's actions, the current sub-task (if any), and relevant memory entries. Based on this information, the LLM infers the next action or tool to use, following the guidelines set in the system prompt.

### G.3 EVALUATION METRIC FOR POKÉMON RED

The goal in our defined segment of *Pokémon Red* is to progress through a series of key storyline milestones. We define 12 predefined storyline flags, and our evaluation metric is the percentage of these flags achieved by the agent. The 12 flags are: Exit Red's House, Encounter Professor Oak, Choose a starter Pokémon, Finish the first battle with the Rival, Arrive in Viridian City, Receive Oak's parcel, Deliver Oak's parcel to Professor Oak, Obtain the Town Map, Purchase a Poké Ball, Catch a new Pokémon, Arrive in Pewter City, Defeat Pewter Gym Leader Brock. The final score is calculated as the percentage of these 12 flags that have been successfully triggered within a given episode or evaluation period. Formally,

$$\text{Score} = \Big(\frac{\text{Number of flags achieved}}{12}\Big) \times 100.$$

**System prompt**

You are Action Inference for a Pokémon Red LLM agent.

Goal: Determine optimal tool use or low-level action(s) to execute `Next_subtask` (or inferred goal) based on current state and rules.

Core Rules Reminder:
- Main Goals: Become Champion, complete Pokédex.
- Controls: A=Confirm/Interact, B=Cancel/Back, Start=Menu, D-Pad=Move. Use for manual actions/menuing if tools don't cover.
- Game States: Current state dictates valid actions/tools.
  - *Title:* Only pressing 'a' is allowed. Select 'CONTINUE', not 'NEW GAME'. DON'T QUIT!
  - *Field:* Move, interact, menu (use nav/interaction tools).
    - Prioritize revealing '?' tiles, unless blocked/interrupted by NPCs or progression gates. However, if important objects or warp points are discovered, consider investigating them instead.
    - In field state, presence of [Interacted Dialog Buffer] means dialog just ended — do not use `continue_dialog.`
  - *Dialog*:* Advance: `continue_dialog` or `B`. Choices: D-Pad(move cursor '▶'), `A` (confirm), `B` (option/name cancel).
    - If D-Pad unresponsive with selection box: press `B` to advance dialog.
    - Looped/long dialog: press `B` repeatedly to exit.
    - Press `B` to delete incorrect characters in the nickname.
    - Finalize name input if cursor '▶' is on '⊮' and 'A' is pressed.
    - Extract critical info from dialog for goals/progression.
  - *Battle:* Use battle tools (moves, items, switch, run). Trainer battles: no running.
- Map Understanding:
  - Map: `[Full Map]` grid (X right, Y down; (0,0)=top-left), `[Notable Objects]` list w/ coords.
  - Walkability (CRITICAL): 'O', 'G', 'WarpPoint', '~'(w/ Surf) = Walkable. 'X', 'Cut', '-', '|', 'TalkTo', 'SPRITE', 'SIGN', '?', Ledges ('D','L','R') = Unwalkable.
  - Interactable with 'A' (CRITICAL): 'TalkTo', 'SPRITE', 'SIGN'.
  - Prioritize paths uncovering '?' (unexplored) tiles.
  - Interact: From adjacent walkable tile, facing target.
- General Strategy:
  - Priorities: Info gathering (NPCs, signs, revealing '?' tiles), resource management (heal, buy), obstacle clearing, goal advancement. Use memory/dialog hints.
  - Exploration: Current (x,y) reveals area (x-4 to x+5, y-4 to y+4). Move to walkable tile near '?' region.
  - Map Transitions: Only via tools `warp_with_warp_point` (needs 'WarpPoint' tile) or `overworld_map_transition` (needs walkable boundary for `overworld`-type maps).

# Manual Button Reference
- A: Confirm/Interact/Advance. Title state: use repeatedly to proceed.
- B: Cancel/Back. Can also advance some dialogs (see Dialog state rules).
- Start: Open/close main menu (Field state).
- D-Pad: Move character/cursor.

# AVAILABLE TOOLS (Use when applicable & valid)
### 1. Field State Tools (Note: `warp_with_warp_point`, `overworld_map_transition`, `interact_with_object` tools include movement; `move_to` not needed before them.)
- move_to(x_dest, y_dest): Move to WALKABLE `(x_dest, y_dest)`. Reveals '?' tiles around dest.
  - Usage: `use_tool(move_to, (x_dest=X, y_dest=Y))`
  - CRITICAL: Dest MUST be WALKABLE ('O','G'); NOT '?', 'X', 'TalkTo', 'SIGN', etc.
  - Not for 'WarpPoint' (use `warp_with_warp_point`) or interactables (use `interact_with_object`).
- warp_with_warp_point(x_dest, y_dest): Moves to 'WarpPoint' `(x_dest,y_dest)` & warps (includes `move_to`).
  - Usage: `use_tool(warp_with_warp_point, (x_dest=X, y_dest=Y))`
  - Needs 'WarpPoint' at coords.
- overworld_map_transition(direction): 'overworld' maps: move off edge to transition (includes `move_to`).
  - `direction`: 'north'|'south'|'west'|'east'
  - Usage: `use_tool(overworld_map_transition, (direction="DIR"))`
  - Needs walkable boundary tile.
- interact_with_object(object_name): Moves adjacent to `object_name` (from Notable Objects), faces, interacts ('A'). Includes `move_to`. Also handles its dialog; no `continue_dialog` needed after.
  - Usage: `use_tool(interact_with_object, (object_name="NAME"))`
### 2. Dialog State Tools
- continue_dialog(): Use ONLY if NO selection box ("▶") visible. Advances dialog ('A'/'B').
  - Usage: `use_tool(continue_dialog, ())`
  - For choices: use D-Pad + 'A', NOT this tool.
### 3 Battle State Tools
- select_move_in_battle(move_name): Select `move_name` (active Pokémon's move, UPPERCASE).
  - Usage: `use_tool(select_move_in_battle, (move_name="MOVE"))`
- switch_pkmn_in_battle(pokemon_name): Switch to `pokemon_name` (from Current Party).
  - Usage: `use_tool(switch_pkmn_in_battle, (pokemon_name="PKMN_NAME"))`
- use_item_in_battle(item_name, pokemon_name=None): Use `item_name` (from Bag) on optional `pokemon_name` (from Current Party).
  - Usage: `use_tool(use_item_in_battle, (item_name="ITEM", pokemon_name="PKMN_NAME"))`
- run_away(): Flee wild battle (not Trainer).
  - Usage: `use_tool(run_away, ())`
---
# INPUTS (`None` if absent)
1. `RecentHistory`: List[(action, resulting_state_summary)] (Always provided)
2. `CurrentGameState`: (obj) Map, Player, Objects, Inventory, Party, Screen Text (includes `screen.screen_type`). (Always provided)
3. `RecentCritique` (Opt): Feedback on last action.
4. `Next_subtask` (Opt): High-level goal (e.g., "Talk to Oak", "Explore Route 1 N").
5. `RelevantMemoryEntries`: List[str] Contextual facts. (Always provided)
---
# CORE LOGIC (Be Concise)
1. Infer Subtask (if `Next_subtask` is `None`): Define immediate step based on state/map/rules (e.g., "Inferred: move_to explore S", "Inferred: continue dialog").
2. Plan Action (Tool-First):
   - State Check: Identify `CurrentGameState.screen.screen_type`.
   - Tool Eval: Find best tool for state & subtask from `# AVAILABLE TOOLS`. Check preconditions (e.g., `move_to` walkability, battle tool state).
   - `move_to` Use (Field state): For nav >4-5 tiles or exploration, strongly prefer `move_to`. Target WALKABLE tile maximizing '?' reveal.
   - Other Tools: Use interact/warp/dialog/battle tools if conditions match.
   - Low-Level: Use Controls (A/B/Start/D-Pad) ONLY if no tool applies OR for precise menu/dialog choices/facing. Max 5 inputs.
   - Justify: Explain tool choice (state, subtask, map, rules). If `move_to` not used for nav, why (e.g., adjacent target, wrong state, no valid path). If LowLevel, why no tool?
3. `Lessons_learned`: Extract factual lessons (state changes, critique, map reveals).
4. Quit Check: Output `quit` only if main goal achieved.

# RESPONSE FORMAT (Strict Adherence Required)
### State_summary
<1-2 lines: Current state, location, status, immediate goal/intent.>

### Lessons_learned
<Lesson 1: e.g., "Fact: `move_to(X,Y)` revealed Pallet S. (X,Y) is 'O'.">
... (max 5 concise, factual lessons. No speculation.)

### Action_reasoning
1. Subtask: [Provided `Next_subtask` or "Inferred: [your inferred subtask]"]
2. ToolEval:
   - ToolChosen: [`<tool_name>` or "LowLevel" or "None"]
   - Justification: [Why this tool/approach (state=`screen_type`, subtask, map, rules)? If `move_to` for nav rejected, why? If LowLevel, why?]
3. Plan: [`use_tool(<tool_name>, <args>)` or `<low-level actions>`.]
4. RedundancyCheck: [How this avoids recent failure/stagnation.]

### Actions
<low-level1> | <low-level2> | ... (MAX 5)
OR
use_tool(<tool_name>, (<arg1>=val1, ...))
OR
quit

# RULES (Strictly follow)
- Cursor move & confirm: separate turns ALWAYS (e.g., 'up', then next turn 'a'; NOT 'up | a' in this response).
- Adhere to state-based tool/action validity.
- Be concise. Adhere strictly to format.

Figure 15: Action inference system prompt for 'zero-shot' agent playing *Pokémon Red*.

**User prompt**
Recent History:
Summary: The player successfully moved from position (3, 6) to (3, 2) within Red's House 2F. An interaction was completed with an object named "TalkToPrintRedSNESText." The player attempted to move to a warp point at (7, 1) but was initially unsuccessful due to the need to use a specific tool. Subsequently, the player successfully used the warp point to transition from Red's House 2F to Red's House 1F at position (7, 1).
Executed Action Sequence: (oldest)[use_tool(interact_with_object, (object_name="TalkToPrintRedSNESText"))
(isSuccess,Feedback):(True, 'Successfully Interact with TalkToPrintRedSNESText.')->use_tool(move_to, (x_dest=7, y_dest=1))->use_tool(move_to, (x_dest=7, y_dest=1))
(isSuccess,Feedback):(False, "The destination is 'WarpPoint'. Use 'warp_with_warp_point' tool.")->use_tool(warp_with_warp_point, (x_dest=7, y_dest=1))->use_tool(warp_with_warp_point, (x_dest=7, y_dest=1))
(isSuccess,Feedback):(True, 'Success to warp to RedsHouse1f (7, 1) using a warp point (7, 1) in RedsHouse2f')](latest)

Current State:
State: Field

[Filtered Screen Text]
N/A

[Selection Box Text]
N/A

[Enemy Pokemon]
- Not in battle

[Current Party]
No more Pokemons

[Badge List]
N/A

[Bag]
N/A

[Current Money]: ¥3000

[Map Info]
Map Name: RedsHouse1f, (x_max , y_max): (7, 7)
Map type: reds_house
Expansion direction: 0
Your position (x, y): (7, 1)
Your facing direction: right
Action instruction
 - up: (x, y) -> (x, y-1)
 - down: (x, y) -> (x, y+1)
 - left: (x, y) -> (x-1, y)
 - right: (x, y) -> (x+1, y)

[Full Map]
        (y=0)
   (x=0) 01234567 (x=7)
         +--------+
0 | ???XXXXX
1 | ???SOOOW
2 | ???OOOOO
3 | ???OOOOO
4 | ???XXSOO
5 | ???XXOOO
6 | ????????
7 | ????????
         (y=7)

[Notable Objects]
( 3,  1) SIGN_REDSHOUSE1F_TV
( 7,  1) WarpPoint
( 5,  4) SPRITE_MOM_1

Recent Critique:
None

Next Subtask:
None

Relevant Memory Entries:
1: Lesson 1: Fact: Successfully moved to (3, 2) in "RedsHouse2f".

Figure 16: Action inference user prompt for 'zero-shot' agent playing *Pokémon Red*.

## G.4    EXPERIMENTAL CONFIGURATION FOR POKÉMON RED

We configure all six open-source LLMs (Llama-3.2-1B/3B, Qwen-2.5-3B/7B, and Minitron-4B/8B) with a temperature of 0.1, a repetition penalty of 1, and a maximum of 1000 game steps. We conduct all experiments over three trials and report the mean score along with its standard deviation.

| Models | Pokémon Red | Rank |
|---|---|---|
| Llama-3.2-1B | $0.0_{\pm0.0}$ | 8.5 |
| Llama-3.2-3B | $0.0_{\pm0.0}$ | 8.5 |
| Qwen-2.5-3B | $0.0_{\pm0.0}$ | 8.5 |
| Qwen-2.5-7B | $0.0_{\pm0.0}$ | 8.5 |
| Minitron-4B | $0.0_{\pm0.0}$ | 8.5 |
| Minitron-8B | $0.0_{\pm0.0}$ | 8.5 |
| GPT-4o-mini | $0.0_{\pm0.0}$ | 8.5 |
| GPT-4o | $38.9_{\pm9.6}$ | 4 |
| o3-mini | $0.0_{\pm0.0}$ | 8.5 |
| Gemini-2.5-pro | $\mathbf{83.3}_{\pm0.0}$ | 1 |
| Claude-3.7 | $63.9_{\pm9.2}$ | 3 |
| Deepseek-R1 | $75.0_{\pm0.0}$ | 2 |

Table 23: Gameplay score on *Pokémon Red*.

| Models | Agent | Pokémon Red | Rank |
|---|---|---|---|
| Llama-3B | Zero-shot | $0.0_{\pm0.0}$ | 6.5 |
| | Reflection | $0.0_{\pm0.0}$ | 6.5 |
| | Planning | $0.0_{\pm0.0}$ | 6.5 |
| | Ref-Plan | $0.0_{\pm0.0}$ | 6.5 |
| GPT-4o | Zero-shot | $33.3_{\pm0.0}$ | 3.5 |
| | Reflection | $36.1_{\pm4.8}$ | 2 |
| | Planning | $33.3_{\pm0.0}$ | 3.5 |
| | Ref-Plan | $\mathbf{38.9}_{\pm9.6}$ | 1 |

Table 24: Ablation study for agentic modules on *Pokémon Red*.

| Models | Input | Pokémon Red | Rank |
|---|---|---|---|
| GPT-4o | Text | $38.9_{\pm9.6}$ | 6 |
| | Both | $41.7_{\pm8.3}$ | 5 |
| Gemini | Text | $\mathbf{83.3}_{\pm0.0}$ | 1.5 |
| | Both | $\mathbf{83.3}_{\pm0.0}$ | 1.5 |
| Claude | Text | $63.9_{\pm19.2}$ | 4 |
| | Both | $72.2_{\pm4.8}$ | 3 |

Table 25: Comparison across modalities on *Pokémon Red*.

## G.5 Result for Pokémon Red

Table 23 presents the gameplay scores on *Pokémon Red*. Gemini-2.5-pro achieves the highest score of 83.3, followed by Deepseek-R1 (75.0) and Claude-3.7 (63.9). GPT-4o achieves a score of 38.9. Notably, all the open-source LLMs evaluated (Llama-3.2-1B/3B, Qwen-2.5-3B/7B, and Minitron-4B/8B) recorded a score of 0.0, indicating significant challenges in playing *Pokémon Red*. Interestingly, o3-mini, which is expected to perform well due to its reasoning capabilities, also achieved a score of 0.0. We observed that o3-mini exhibited a tendency to rely on its pre-existing knowledge or intuition rather than adapting to the game environment, such as consistently moving downwards based on a likely incorrect assumption about the exit's location, leading to unproductive repeated actions. This highlights the difficulty some models face in grounding their reasoning within the specific context of the game.

Table 24 shows the impact of reflection and planning modules. For GPT-4o, the 'reflection-planning' agent achieved the highest score (38.9), followed by the 'reflection' agent (36.1), and then the 'planning' and 'zero-shot' agents (both at 33.3). This suggests that reflection plays a crucial role in improving performance in *Pokémon Red*. In contrast, Llama-3.2-3B consistently scored 0.0 across all agent configurations.

The comparison across input modalities is presented in Table 25. For Gemini-2.5-pro and Claude-3.7, using both text and image inputs resulted in performance equal to or better than using text input alone. This indicates that visual information can be beneficial in this environment. GPT-4o also showed a performance increase when both modalities were used (41.7) compared to text-only input (38.9), suggesting that incorporating image data aids the agent's decision-making.

## H Darkest Dungeon

### H.1 Game Description for Darkest Dungeon

**Environment.** *Darkest Dungeon* is a turn-based roguelike role-playing game where the player manages a roster of heroes as they explore procedurally generated dungeons filled with monsters, traps, and treasures (Red Hook Studios, 2016). Each hero has unique abilities and a stress level that influences their behavior during combat and exploration. Stress accumulates through continued exploration and battle, and heroes who reach high stress thresholds may develop afflictions that hinder or occasionally enhance their performance. The game emphasizes tactical positioning, turn-order strategy, and long-term roster management. Elements of randomness, such as attack accuracy, critical hits, and affliction outcomes, introduce uncertainty and require players to adapt their strategies dynamically. For implementation, we build our environment on top of a rule-based bot (Kgleken, 2023), replacing its rule-based combat logic with the decisions made by LLM agents. To access internal game states, we utilize the *Darkest Dungeon Save Editor* (Robojumper, 2023). Since the game does not support complete control via keyboard input, we employ the *Xbox 360 controller emulator* to inject actions from the LLM agent. We evaluate the agent's performance during the first embarkation mission after the tutorial, which we designate as our *default evaluation setting*. For consistency, we fix the party roster to include the `Plague Doctor`, `Vestal`, `Highwayman`, and `Crusader`, in that order, and equip the inventory with an additional 8 'Food' and 8 'Torches'. To ensure reproducibility, we provide a save file with this setup preconfigured. During the mission,

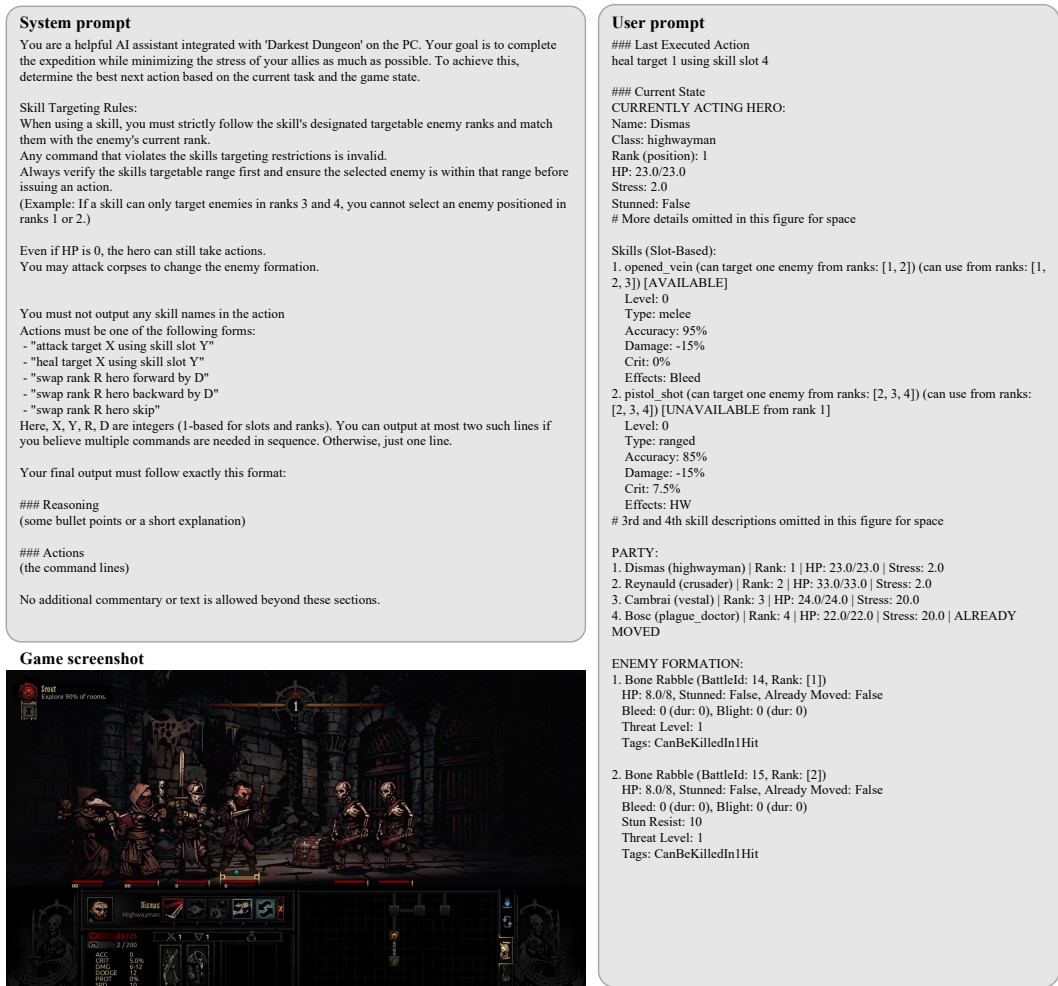

**System prompt**
You are a helpful AI assistant integrated with 'Darkest Dungeon' on the PC. Your goal is to complete the expedition while minimizing the stress of your allies as much as possible. To achieve this, determine the best next action based on the current task and the game state.

Skill Targeting Rules:
When using a skill, you must strictly follow the skill's designated targetable enemy ranks and match them with the enemy's current rank.
Any command that violates the skills targeting restrictions is invalid.
Always verify the skills targetable range first and ensure the selected enemy is within that range before issuing an action.
(Example: If a skill can only target enemies in ranks 3 and 4, you cannot select an enemy positioned in ranks 1 or 2.)

Even if HP is 0, the hero can still take actions.
You may attack corpses to change the enemy formation.

You must not output any skill names in the action
Actions must be one of the following forms:
 - "attack target X using skill slot Y"
 - "heal target X using skill slot Y"
 - "swap rank R hero forward by D"
 - "swap rank R hero backward by D"
 - "swap rank R hero skip"
Here, X, Y, R, D are integers (1-based for slots and ranks). You can output at most two such lines if you believe multiple commands are needed in sequence. Otherwise, just one line.

Your final output must follow exactly this format:

### Reasoning
(some bullet points or a short explanation)

### Actions
(the command lines)

No additional commentary or text is allowed beyond these sections.

**Game screenshot**

**User prompt**
### Last Executed Action
heal target 1 using skill slot 4

### Current State
CURRENTLY ACTING HERO:
Name: Dismas
Class: highwayman
Rank (position): 1
HP: 23.0/23.0
Stress: 2.0
Stunned: False
# More details omitted in this figure for space

Skills (Slot-Based):
1. opened_vein (can target one enemy from ranks: [1, 2]) (can use from ranks: [1, 2, 3]) [AVAILABLE]
    Level: 0
    Type: melee
    Accuracy: 95%
    Damage: -15%
    Crit: 0%
    Effects: Bleed
2. pistol_shot (can target one enemy from ranks: [2, 3, 4]) (can use from ranks: [2, 3, 4]) [UNAVAILABLE from rank 1]
    Level: 0
    Type: ranged
    Accuracy: 85%
    Damage: -15%
    Crit: 7.5%
    Effects: HW
# 3rd and 4th skill descriptions omitted in this figure for space

PARTY:
1. Dismas (highwayman) | Rank: 1 | HP: 23.0/23.0 | Stress: 2.0
2. Reynauld (crusader) | Rank: 2 | HP: 33.0/33.0 | Stress: 2.0
3. Cambrai (vestal) | Rank: 3 | HP: 24.0/24.0 | Stress: 20.0
4. Bosc (plague_doctor) | Rank: 4 | HP: 22.0/22.0 | Stress: 20.0 | ALREADY MOVED

ENEMY FORMATION:
1. Bone Rabble (BattleId: 14, Rank: [1])
    HP: 8.0/8, Stunned: False, Already Moved: False
    Bleed: 0 (dur: 0), Blight: 0 (dur: 0)
    Threat Level: 1
    Tags: CanBeKilledIn1Hit

2. Bone Rabble (BattleId: 15, Rank: [2])
    HP: 8.0/8, Stunned: False, Already Moved: False
    Bleed: 0 (dur: 0), Blight: 0 (dur: 0)
    Stun Resist: 10
    Threat Level: 1
    Tags: CanBeKilledIn1Hit

Figure 17: Action inference prompt for 'zero-shot' agent playing *Darkest Dungeon*.

dungeon exploration is handled by rule-based logic, while all combat decisions are delegated to the LLM agent.

**Observation-to-Text Conversion.** We convert the internal game state of *Darkest Dungeon* into a structured textual description suitable for LLM input. The observation includes combat-relevant details such as the active hero's stats, available skills, party composition, and enemy formation. For each hero, we format key attributes (e.g., HP, stress, position, status effects) along with skill availability and target constraints, using symbolic descriptors extracted from a parsed skill configuration file. Enemy information is similarly structured, including HP, rank, resistances, and threat indicators. The final text is composed of three parts: a detailed hero description with skill information, a party summary with basic stats, and an enemy formation breakdown.

**Action Space.** At each decision point, the agent chooses one of four action types: 'Attack', 'Heal', or 'Swap'. For 'Attack' and 'Heal' actions, the agent specifies a skill slot index and a target index corresponding to a specific enemy or ally. 'Swap' actions require the agent to provide the current hero's rank and a swap distance. To allow for complex plans (*e.g.*, swapping then healing), the agent may output up to two such structured command lines in sequence.

## H.2 GAMEPLAY PROMPT FOR DARKEST DUNGEON

Figure 17 shows the action inference prompt used by the 'zero-shot' agent for playing *Darkest Dungeon*. The system prompt encodes task-specific knowledge, including (1) the primary objective of completing expeditions while minimizing party stress, (2) strict targeting constraints for combat skills

| Models | DarkestD | Rank |
|---|---|---|
| Llama-3.2-1B | $0.0_{\pm 0.0}$ | 11.5 |
| Llama-3.2-3B | $47.5_{\pm 39.2}$ | 9 |
| Qwen-2.5-3B | $44.8_{\pm 22.2}$ | 10 |
| Qwen-2.5-7B | $88.8_{\pm 2.0}$ | 6 |
| Minitron-4B | $0.0_{\pm 0}$ | 11.5 |
| Minitron-8B | $63.8_{\pm 30.4}$ | 8 |
| GPT-4o-mini | $81.3_{\pm 5.8}$ | 7 |
| GPT-4o | $93.4_{\pm 1.5}$ | 2 |
| o3-mini | $89.0_{\pm 2.1}$ | 5 |
| Gemini-2.5-pro | $\mathbf{93.7}_{\pm 1.6}$ | 1 |
| Claude-3.7 | $89.9_{\pm 2.5}$ | 4 |
| Deepseek-R1 | $91.7_{\pm 1.1}$ | 3 |

Table 26: Gameplay score on *Darkest Dungeon*.

| Models | Agent | DarkestD | Rank |
|---|---|---|---|
| Llama-3B | Zero-shot | $47.5_{\pm 39.2}$ | 7 |
| | Reflection | $47.3_{\pm 39.0}$ | 8 |
| | Planning | $56.3_{\pm 23.6}$ | 6 |
| | Ref-Plan | $57.0_{\pm 31.6}$ | 5 |
| GPT-4o | Zero-shot | $\mathbf{93.4}_{\pm 1.5}$ | 1 |
| | Reflection | $85.2_{\pm 10.3}$ | 3 |
| | Planning | $82.0_{\pm 8.6}$ | 4 |
| | Ref-Plan | $91.6_{\pm 2.5}$ | 2 |

Table 27: Ablation study for agentic modules on *Darkest Dungeon*.

| Models | Input | DarkestD | Rank |
|---|---|---|---|
| GPT-4o | Text | $\mathbf{93.4}_{\pm 1.5}$ | 2 |
| | Image | $92.2_{\pm 3.0}$ | 3.5 |
| Gemini | Text | $\mathbf{93.7}_{\pm 1.6}$ | 1 |
| | Image | $92.2_{\pm 1.8}$ | 3.5 |
| Claude | Text | $89.9_{\pm 2.5}$ | 6 |
| | Image | $\mathbf{90.1}_{\pm 5.7}$ | 5 |

Table 28: Comparison across modalities on *Darkest Dungeon*.

based on hero and enemy rank positions, and (3) the expected format for issuing valid commands. The user prompt provides the current game state, including the acting hero's stats and available skills, a summary of the party, and the enemy formation. Some fields are omitted for brevity, but the format mirrors the actual prompt used during inference.

## H.3 EVALUATION METRIC FOR DARKEST DUNGEON

To evaluate the performance of an LLM agent in *Darkest Dungeon*, we define a composite scoring metric that reflects progress, survivability, and stress management throughout the expedition. The score consists of three components: (1) the proportion of room combats successfully cleared, weighted at 40 points, (2) the fraction of heroes who survive the entire mission (out of four), weighted at 30 points, and (3) the remaining stress capacity of the team, also weighted at 30 points. However, the latter two components are only counted if the stage is successfully cleared (i.e., if the first term reaches its full 40 points); otherwise, they are set to zero. To ensure fairness, the stress of any hero who dies before the end of the run is treated as the maximum stress (200) when computing the average team stress. The final score is computed as:

$$
\text{Score} = \begin{cases} 40 \cdot \left( \frac{\text{\# combats cleared}}{\text{\# total combats}} \right) + 30 \cdot \left( \frac{\text{\# heroes survived}}{4} \right) + 30 \cdot \left( 1 - \frac{\text{total stress}}{800} \right), & \text{if stage is cleared} \\ 40 \cdot \left( \frac{\text{\# combats cleared}}{\text{\# total combats}} \right), & \text{otherwise} \end{cases}
$$

## H.4 EXPERIMENTAL CONFIGURATION FOR DARKEST DUNGEON

For all 6 open-source LLMs, we use a temperature of 0.7 and a repetition penalty of 1, and set the maximum number of game steps to 200. We run all experiments with 3 trials and report the average score with the standard deviation.

## H.5 RESULT FOR DARKEST DUNGEON

Table 26 reports the gameplay scores of various models on *Darkest Dungeon*. We show that Gemini-2.5-pro achieves the highest score of 93.7, closely followed by GPT-4o (93.4) and Deepseek-R1 (91.7), demonstrating strong capabilities across combat decisions and roster management. Among open-source models, Qwen-2.5-7B performs best, achieving a score of 88.8. In contrast, smaller models such as Llama-3.2-1B and Minitron-4B fail to make meaningful progress, often producing invalid outputs and scoring 0.0. A closer analysis reveals that Llama-3.2-1B frequently fails to follow the correct action format, while Llama-3.2-3B and Qwen-2.5-3B tend to issue invalid commands, such as using unavailable skills from incorrect hero positions or targeting unreachable enemies. These models also overuse 'Swap' actions, leading to inefficient combat sequences. Notably, many small models become stuck in a loop when the 'Crusader' hero is affected by the 'Surprised!' status and repositioned to the back row. Since most of the Crusader's skills are unusable from that position, the models repeatedly attempt invalid actions or swap ineffectively, wasting turns and failing to recover from the disrupted formation.

Table 27 and Table 28 present ablation studies on agentic modules and input modalities in *Darkest Dungeon*. For agentic components, GPT-4o performs best in the zero-shot setting, with reflection and planning offering marginal or even negative impact on its performance. Llama-3.2-3B shows

small but consistent gains when equipped with the planning module, though overall improvements remain limited. This suggests that large models like GPT-4o already possess sufficient planning capabilities for this task, while smaller models benefit only modestly from explicit agentic prompting. In terms of modality (Table 28), we find that providing image input in addition to text yields minimal improvement. For all models, performance remains similar or only slightly better when image data is included, indicating that the agents primarily rely on the structured text input to make decisions.

## I  MINECRAFT

### I.1  GAME DESCRIPTION FOR MINECRAFT

**Environment.** *Minecraft* (Mojang Studios, 2011) is an open-ended sandbox game where players explore a world, gather resources, and survive by placing and breaking blocks. This environment is based on the Mineflayer JavaScript API (contributors, 2013). Using Mineflayer, the agent can control a Minecraft bot through high-level JavaScript commands. We use Minecraft version 1.19 for compatibility, and to ensure consistent evaluation, we fix the world seed to 42 and initialize the bot at coordinates (604, 100, -823). The bot starts in survival mode with no items and progressively crafts a target item using its in-game observations and JavaScript-based actions generated by the LLM. We select 8 target items with varying levels of crafting difficulty: 'crafting table', 'stone pickaxe', 'furnace', 'bucket', 'golden sword', 'diamond pickaxe', 'enchanting table', and 'nether portal'.

**Observation-to-Text Conversion.** The Mineflayer API provides the bot with its state and contextual information from the surrounding environment. Specifically, the bot receives textual observations in the form of: {Current biome, DayTime, Nearby blocks, Health status, Hunger status, Position, Equipped items, Inventory contents}.

**Action Space.** The action space consists of JavaScript code that interfaces with the Mineflayer API. Following Voyager (Wang et al., 2023), we expose the following set of *control primitives* to guide the LLM in generating valid and effective code actions for the bot using in-context learning.

- `exploreUntil(bot, direction, maxTime, callback)`: Moves the agent in a fixed direction for up to maxTime seconds, or until a custom stopping condition (defined in callback) is satisfied.
- `mineBlock(bot, name, count)`: Mines and collects up to count number of blocks with the specified name, within a 32-block radius.
- `craftItem(bot, name, count)`: Crafts the specified item using a nearby crafting table.
- `placeItem(bot, name, position)`: Places a block of the specified type at the given position.
- `smeltItem(bot, itemName, fuelName, count)`: Smelts the specified item using the provided fuel. Requires access to a nearby furnace.
- `KillMob(bot, mobName, timeout)`: Hunts and eliminates the specified mob within the time limit, and collects any resulting drops.
- `getItemFromChest(bot, chestPosition, itemsToGet)`: Navigates to the chest at the given location and retrieves the requested items.
- `depositItemIntoChest(bot, chestPosition, itemsToDeposit)`: Navigates to the given chest and deposits specified items into it.

### I.2  GAMEPLAY PROMPT FOR MINECRAFT

Figure 18 shows the action inference prompt used by the 'zero-shot' agent for playing *Minecraft*. The system prompt contains gameplay-specific knowledge and guidance for action inference using Mineflayer APIs. It includes (1) the main task of the game, (2) control primitives, which is Javascript code template that should be referred to, (3) game observation in text format, and (4) the expected output response format with reasoning and code. The user prompt provides the current game state provided by Mineflayer APIs. Given this prompt, the LLM agent infers the appropriate Javascript code for action to complete the target task.

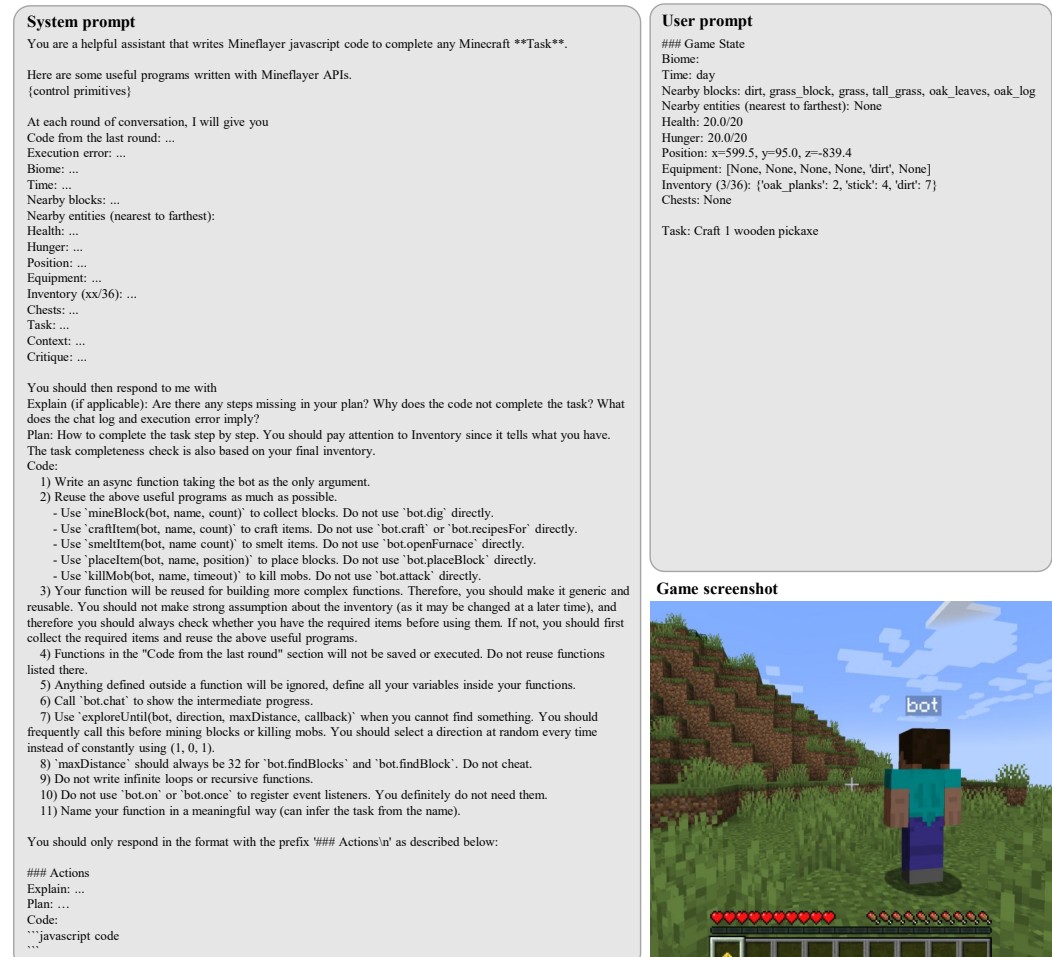

Figure 18: Action inference prompt for 'zero-shot' agent playing *Minecraft*.

## I.3 EVALUATION METRIC FOR MINECRAFT

We evaluate agent performance using the success rate of crafting target item. Since Mineflayer APIs provide access to the agent's inventory, we compute the success score by checking whether the target item appears in the inventory at each game step. If the item is successfully crafted and presented, the episode is marked as successful with a score of 100; otherwise, 0.

## I.4 EXPERIMENTAL CONFIGURATION FOR MINECRAFT

For all LLMs, we use a temperature of 1 and a repetition penalty of 1. The interaction with the game environment is limited to a maximum of 100 steps. We repeat all experiments 3 times to craft each target item, and report the average score and standard deviation.

## I.5 RESULT FOR MINECRAFT

As shown in Table 29, o3-mini, Gemini-2.5-pro, and Claude-3.7-sonnet performed the best on Minecraft, each obtaining a score of 75.0. All three models successfully crafted the following six items: 'crafting table', 'stone pickaxe', 'furnace', 'bucket', 'golden sword', and 'diamond pickaxe'. However, they all failed to craft the two most difficult items: 'enchanting table' and 'nether portal'. In contrast, all six open-source LLMs failed to craft any of the target items, resulting in a score of 0.0. Notably, five of the models, except Minitron-8B, failed to generate any executable JavaScript code compatible with the Mineflayer API. While Minitron-8B was able to generate valid code sometimes to move the bot and mine wood, it failed to craft even the simplest item, the crafting table.

| Models | Minecraft | Rank |
|---|---|---|
| Llama-3.2-1B | 0.0±0.0 | 9.5 |
| Llama-3.2-3B | 0.0±0.0 | 9.5 |
| Qwen-2.5-3B | 0.0±0.0 | 9.5 |
| Qwen-2.5-7B | 0.0±0.0 | 9.5 |
| Minitron-4B | 0.0±0.0 | 9.5 |
| Minitron-8B | 0.0±0.0 | 9.5 |
| GPT-4o-mini | 46.0±7.0 | 5 |
| GPT-4o | 71.0±7.0 | 4 |
| o3-mini | **75.0**±0.0 | 2 |
| Gemini-2.5-pro | **75.0**±0.0 | 2 |
| Claude-3.7 | **75.0**±0.0 | 2 |
| Deepseek-R1 | 41.7±0.0 | 6 |

Table 29: Gameplay score on *Minecraft*.

| Models | Agent | Minecraft | Rank |
|---|---|---|---|
| Llama-3B | Zero-shot | 0.0±0.0 | 6 |
| | Reflection | 0.0±0.0 | 6 |
| | Planning | 0.0±0.0 | 6 |
| | Ref-Plan | 0.0±0.0 | 6 |
| GPT-4o | Zero-shot | 0.0±0.0 | 6 |
| | Reflection | **50.0**±0.0 | 1.5 |
| | Planning | 13.0±0.0 | 3 |
| | Ref-Plan | **50.0**±0.0 | 1.5 |

Table 30: Ablation study for agentic modules on *Minecraft*.

| Seed | Buy Price | Sell Price | Growth Days | Notes |
|---|---|---|---|---|
| Parsnip Seeds | 20 | 35 | 4 | |
| Bean Starter | 60 | 40 | 10 | Regrows every 3 days after first harvest |
| Cauliflower Seeds | 80 | 175 | 12 | |
| Potato Seeds | 50 | 80 | 6 | 20% chance of extra yield |

Table 31: Comparison of available seeds in *Stardew Valley*.

As shown in Table 30, the reflection module, which encourages the model to generate improved code actions based on past failed attempts, significantly improved the performance of GPT-4o. However, the 'reflection-planning' agent achieved a score of 50.0, which is lower than the default 'skill-management' agent score of 71.0 in Table 29. This suggests that the skill management module, which is responsible for storing previously successful code actions and retrieving them when needed, plays a more substantial role in enhancing the performance of *Minecraft*.

## J  STARDEW VALLEY

### J.1  GAME DESCRIPTION FOR STARDEW VALLEY

**Environment.** *Stardew Valley* (ConcernedApe, 2016) is a life simulation and farming role-playing game. The player can engage in a variety of daily activities such as farming, fishing, mining, foraging, and socializing with villagers.

Our objective is to evaluate an LLM agent's ability to autonomously perform farming-related tasks that maximize monetary gain within the first 13 in-game days (i.e., until the Egg Festival on Spring 13). Specifically, we focus on harvesting crops and strategically earning money by predicting high-profit crops and interacting with the in-game environment. The character begins on Day 1 with 200 gold. Four types of seeds are available for purchase: parsnip seeds, bean starter, cauliflower seeds, and potato seeds. Each seed type differs in cost, selling price, days to harvest, and other characteristics. Table 31 summarizes the properties of each seeds. To make the task challenging and enforce planning under resource constraints, we manually set the player's maximum energy to 50 (default: 200). Using tools such as the hoe or watering can consumes 2 energy per use, limiting the number of tiles the agent can till or water in a single day. If the agent's energy drops to 0 or below, they start the next day with only 26 energy (instead of 50). Furthermore, if energy falls -15 or below, the player loses 10% of their current gold and still begins the following day with only 26 energy. This constraint encourages the agent to prioritize actions and manage resources efficiently.

We run the game on the *Steam* platform and use the modding tool SMAPI (Pathoschild, 2025) to extract in-game states and implement custom actions. To send keyboard and mouse inputs, we use the *pyautogui* library on macOS and the *AutoHotkey (AHK)* library on Windows, following prior work (Tan et al., 2024). Since many in-game actions such as planting or watering crops consist of multiple low-level actions (e.g., move up/down/right/left, switch tool, use tool), we define a set of high-level actions to abstract these into semantically meaningful units. Each high-level action is mapped to a predefined sequence of keyboard inputs and implemented via SMAPI scripts with custom keyboard bindings. We provide the save point used for our experiment to ensure reproducibility.

**Observation-to-Text Conversion.** After each high-level action is executed, we extract a JSON-formatted game state via SMAPI, which includes information such as player location, current inventory, crop states in the field, remaining energy, and money. This state is then serialized into a natural language description and passed to the LLM.

**Action Space.** We define a compact action space consisting of 8 high-level actions essential for solving the task. These actions abstract away low-level controls and are defined as follows:

- `till_soil(num_tiles)`: Tills *num_tiles* soil tiles to prepare them for planting. The tiles are selected in a fixed order starting from the pre-defined position.
- `plant_seeds()`: Plants all available seeds from the inventory into empty, tilled soil tiles. If the number of tilled tiles is less than the number of seeds, only the available plots are used.
- `water_seeds()`: Waters all planted crops that have not yet been watered on the current day.
- `harvest_crops()`: Harvests all crops that are fully grown and ready to be collected.
- `sell_item()`: Sells all harvested crops currently in the inventory.
- `buy_item(item_name, item_count)`: Opens the shop interface, selects the specified item, and attempts to purchase the specified quantity. If there is insufficient money, the agent buys as many units as possible.
- `get_out_of_house()`: Moves the character out of the house.
- `go_house_and_sleep()`: Navigates the character back to the house, enters it, moves to the bed, and interacts with it to end the day.

## J.2 GAMEPLAY PROMPT FOR STARDEW VALLEY

Figure 19 shows the action inference prompt used by the 'zero-shot' agent for playing *Stardew Valley*. The system prompt defines the agent's role as an in-game assistant tasked with selecting the best next action based on the current situation and target task. It specifies strict behavioral rules, such as only using actions from a predefined set, avoiding repeated failed actions, and formatting outputs as Python code (up to two actions). The valid action set includes available actions with clear descriptions and constraints. The user prompt provides the goal and detailed context, including crop stats, energy rules, and the current game state (location, inventory, weather, etc.). It also includes the last executed actions and requires the agent to valid action output.

## J.3 EVALUATION METRIC FOR STARDEW VALLEY

The objective in the *Stardew Valley* task is to maximize the amount of money earned during the first 13 in-game days, starting from Day 1. This period is a natural milestone in the game, as the Egg Festival takes place on Spring 13, where players can purchase high-reward crops such as Strawberry seeds. We evaluate performance based on the net profit earned by the end of Spring 13, calculated as the difference between the final gold amount and the initial amount of 200 gold. We normalize the score using a human expert baseline of $1156 - 200 = 956$ gold, which we assign a normalized score of 100. Formally, the normalized score is defined as

$$\text{Score} = (x_{\text{final}} - x_{\text{start}})/(x_{\text{oracle}} - x_{\text{start}}) = (x_{\text{final}} - x_{\text{start}})/956,$$

where $x_{\text{final}}$ is the agent's final gold amount, $x_{\text{start}} = 200$ is the starting gold, and $x_{\text{oracle}} = 1156$ is the human expert score.

## J.4 EXPERIMENTAL CONFIGURATION FOR STARDEW VALLEY

For all LLMs, we use a temperature of 0.0 and a repetition penalty of 1. The interaction with the game environment is limited to a maximum of 150 steps. For GPT-4o, we used GPT-4o-2024-05-13 model for Stardew Valley. During LLM inference, the game is paused to prevent in-game time from progressing, which could otherwise alter the environment (*e.g.*, a day transition). We perform each experiments three times and and present the average score with the standard deviation.

**System prompt**

You are a helpful AI assistant integrated with 'Stardew Valley' on the PC, equipped to handle various tasks in the game. Your goal is to determine the best next action based on the given task, controlling the game character to execute the appropriate actions from the available action set.

Analyze the current situation and provide the reasoning for what you should do for the next step to complete the task. Then, you should output the exact action you want to execute in the game.:

Reasoning: You should think step by step and provide detailed reasoning to determine the next action executed on the current state of the task.

Guidelines:
1. You should output actions in Python code format and specify any necessary parameters to execute that action. If the function has parameters, you should also include their names and decide their values. If it does not have a parameter, just output the action.
2. You can only output at most two actions in the output.
3. If you want to get out of the house, just use the skill get_out_of_house().
4. If you want to move to home and sleep, just use the skill go_house_and_sleep().
5. You MUST NOT repeat the previous action again if you think the previous action fails.
6. You MUST choose actions only from the given valid action set. Any action outside this set is strictly forbidden.
7. If you are at the FarmHouse, the task you MUST do is to leave the house and go to the farm.

### Valid action set in Python format
Function: get_out_of_house()
Description: Move the character out of the house. This function automates the action of moving the character out of the house by navigating through the door. This function only takes effect when the character is inside the house and in bed.

Function: go_house_and_sleep()
Description: Let the character move to house and enter the house and then move the character to the bed and interact with it to go to sleep. This function automates the action of moving the character to the bed and interacting with it to go to sleep.

Function: buy_item(item_name, item_count)
Description: This function opens the shop interface, selects the specified item, and buys the desired quantity. It can be executed from anywhere in the game world, ensuring seamless item acquisition. If item_name is not one of the available choices, the function will do nothing.

Parameters:
- item_name: The name of the item to be bought. (CHOICES: "Parsnip Seeds", "Bean Starter", "Cauliflower Seeds", "Potato Seeds")
- item_count: The number of items to be bought.

Function: sell_item()
Description: Sell all crops in the inventory. This function automatically opens the shop interface and sells all crops in the inventory. This function operates wherever the player is in the game world.

Function: till_soil(num_tiles)
Description: Till the soil. This function automatically till the given number of soil tiles located at the predefined position. This function only work when the character is in the farm area.

Parameters:
- num_tiles: Number of soil tiles to till.

Function: plant_seeds()
Description: This function plants all available seeds from the inventory into tilled soil. It operates under the assumption that there is a sufficient number of empty tilled soil plots. If there are fewer available plots than seeds, only the available plots will be used. The character must be in the farm area for this function to work. If no seeds are in the inventory, the function will do nothing.

Function: water_seeds()
Description: This function waters all planted seeds. This function only work when the character is in the farm area. If all plants are watered, this function will do nothing.

Function: harvest_crops()
Description: Harvest all crops which are ready to harvest. This function only work when the character is in the farm area.

**User prompt**

### Target task
Your task is to maximize profit before the morning of Spring 14th through strategical crop selection and cultivation. Each seed type has different growth times, purchase costs, and selling prices. 'Parsnip Seeds' grow in 4 days, costing 20g per seed, and selling for 35g. 'Bean Starter' takes 10 days to mature, cost 60g per seed, sell for 40g, and can be harvested every 3 days after maturity. 'Cauliflower Seeds' take 12 days, cost 80g, and sell for 175g. 'Potato Seeds' grow in 6 days, cost 50g, sell for 80g, and have a 20% chance to yield an extra crop. When harvested, crops have a chance to be of higher quality, which can be sold for a better price. You have 50 energy per day, and tilling soil or watering seeds consumes 2 energy per action. If your energy drops below 0, you will become exhausted, starting the next day with only 26 energy. If your energy drops to -15, you will pass out, losing 10% of your money and starting the next day with 26 energy. Tilled soil without crop may revert to untilled soil overnight with a certain probability, requiring re-tilling before planting new seeds. Your final score is determined by the money you have at the start of Spring 14th. Any crops that are not harvested by that time will not be counted, even if they are still growing. Do not buy and plant seeds if the crop cannot fully mature within the remaining time. Doing so will yield no returns and result in wasted resources. Always check the growth time before planting. To succeed, you must choose the most profitable seeds, till the soil, plant and care for them daily, harvest when ready, and sell them—then repeat the process to grow your earnings. Other actions, such as clearing debris, are not required. Crop cultivation is the sole method of earning money.

### Last executed action
```python
get_out_of_house()
plant_seeds()
```

### Current state
The player is located at Farm. The player has 0 gold and 26/50 energy remaining.

Today is spring 2. 11 days remaining. The weather is Raining.

Crops currently growing:
- Potato (Stack: 4, Days to harvest: 6, Watered: True)

Number of empty tilled soil tiles:
19

The toolbar contains the following items:
1. Axe (Stack: 1)
2. Hoe (Stack: 1)
3. Watering Can (Stack: 1)
4. Pickaxe (Stack: 1)
5. Scythe (Stack: 1)

You should only respond in the format described below, and you should not output comments or other information.
### Reasoning
1. ...
2. ...
3. ...
### Actions
```python
action(args1=x,args2=y)
```

**Game screenshot**

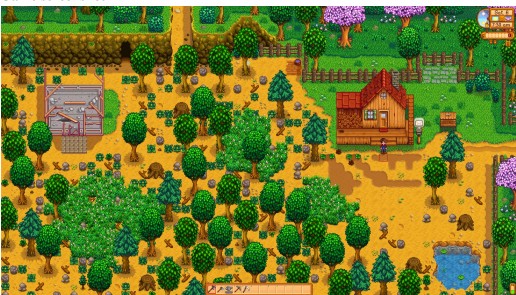

Figure 19: Action inference prompt for 'zero-shot' agent playing *Stardew Valley*.

| Models | Stardew | Rank |
|---|---|---|
| Llama-3.2-1B | $0.0_{\pm 0.0}$ | 10.5 |
| Llama-3.2-3B | $0.0_{\pm 0.0}$ | 10.5 |
| Qwen-2.5-3B | $0.0_{\pm 0.0}$ | 10.5 |
| Qwen-2.5-7B | $0.0_{\pm 0.0}$ | 10.5 |
| Minitron-4B | $0.0_{\pm 0.0}$ | 10.5 |
| Minitron-8B | $0.0_{\pm 0.0}$ | 10.5 |
| GPT-4o-mini | $16.1_{\pm 27.8}$ | 7 |
| GPT-4o | $81.4_{\pm 4.8}$ | 2 |
| GPT-5 | $\mathbf{92.3}_{\pm 8.6}$ | 1 |
| o3-mini | $55.1_{\pm 16.0}$ | 5 |
| Gemini-2.5-pro | $59.2_{\pm 10.1}$ | 4 |
| Claude-3.7 | $53.6_{\pm 20.9}$ | 6 |
| Deepseek-R1 | $66.1_{\pm 11.5}$ | 3 |

Table 32: Gameplay score on *Stardew Valley*.

| Models | Agent | Stardew | Rank |
|---|---|---|---|
| Llama-3B | Zero-shot | $0.0_{\pm 0.0}$ | 6.5 |
| | Reflection | $0.0_{\pm 0.0}$ | 6.5 |
| | Planning | $0.0_{\pm 0.0}$ | 6.5 |
| | Ref-Plan | $0.0_{\pm 0.0}$ | 6.5 |
| GPT-4o | Zero-shot | $40.5_{\pm 35.2}$ | 3 |
| | Reflection | $18.3_{\pm 5.2}$ | 4 |
| | Planning | $64.6_{\pm 23.7}$ | 2 |
| | Ref-Plan | $\mathbf{95.7}_{\pm 5.7}$ | 1 |

Table 33: Ablation study for agentic modules on *Stardew Valley*.

| Models | Input | Stardew | Rank |
|---|---|---|---|
| GPT-4o | Text | $\mathbf{81.4}_{\pm 4.8}$ | 1 |
| | Image | $0.0_{\pm 0.0}$ | 8.5 |
| | Both | $41.9_{\pm 22.1}$ | 6 |
| Gemini | Text | $59.2_{\pm 10.1}$ | 3 |
| | Image | $7.6_{\pm 9.0}$ | 7 |
| | Both | $60.0_{\pm 6.0}$ | 2 |
| Claude | Text | $53.6_{\pm 20.9}$ | 4 |
| | Image | $0.0_{\pm 0.0}$ | 8.5 |
| | Both | $49.8_{\pm 1.0}$ | 5 |

Table 34: Comparison across modalities on *Stardew Valley*.

## J.5 RESULT FOR STARDEW VALLEY

Table 32 presents a comparison of LLM performance in *Stardew Valley*. Among the models, GPT-4o presents the best performance, followed by Gemini-2.5-pro. Open-sourced LLMs fails to earn money, primarily for two reasons; (1) failure to perform valid actions (all models excepts Qwen-2.5-7B)

(2) poor crop scheduling, resulting in crops not being ready for harvest on Day 13 (Qwen-2.5-7B). Under the imposed energy constraints, the optimal strategy is to plant Parsnip Seeds every four days, as they yield the highest profit. The player should purchase and plant as many seeds as possible on Days 1 and 5, and exactly 24 seeds on Day 9. Planting more than 24 seeds on Day 9 depletes the player's energy during watering, leading to insufficient energy the next day and ultimately a failure to harvest on Day 13. None of the LLMs, including the API-based models, fully followed this optimal strategy. In particular, most models frequently selected suboptimal crops such as potato seeds, which contributed to their lower performance.

Table 33 shows the effect of different agentic modules on performance. In the case of GPT-4o, the planning module has a significant effect, as the task requires accurate seed selection and scheduling. In contrast, Llama-3.2-3B model fails to make a profit across four agent configuration, indicating that the underlying model's capabilities are a limiting factor regardless of agent design.

Table 34 summarizes game performance under different input modalities. Notably, all three proprietary LLMs fails to achieve strong performance when only vision input is available. This underperformance is mainly due to the difficulty of extracting structured information from a single screenshot (see figure 19 for an example). Although the screenshot contains rich contextual information, it also contains a lot of redundant content, and critical information occupies only a small portion of the image. For instance, watered plants appears slightly darker than dry soil, making it difficult to distinguish visually. Similarly, the current day, a crucial cue for planning, is indicated in a small font in the upper-right corner of screenshot. When both text and image inputs are given, GPT-4o and Claude-3.7-Sonnet exhibit a performance drop, while Gemini-2.5-pro shows improved performance. This suggests that only sufficiently capable models can effectively integrate multimodal information, while others may struggle with modality fusion or become distracted by noisy visual inputs.

# K  STARCRAFT II

## K.1  GAME DESCRIPTION FOR STARCRAFT II

**Environment.** *StarCraft II* is a real-time strategy game where players gather resources, construct buildings, train units, and command armies to defeat opponents. The environment features a partially observable map, requiring the agent to explore and gather information about the opponent's actions. For implementation, we adopt the `BurnySc2/python-sc2` environment (BurnySc2, 2017), a Python interface widely used in the reinforcement learning (RL) community. Note that the library supports the raw scripted interface without a graphics-based interface. The environment supports various official maps and game modes. For our *default evaluation* (as in Table 3), we use the 'Ancient Cistern LE' map with the agent playing as Protoss against the built-in AI bot (Zerg, Hard difficulty; employing a *timing* build order strategy). The environment allows testing across different races, maps, and difficulty settings. For instance, the 'Babylon LE' is used to assess *intra-game generalization* in Table 7.

**Observation-to-Text Conversion.** The `BurnySc2/python-sc2` environment provides the agent with observations capturing the current game state and context. Specifically, the observations include: {Resources, Buildings, Units, Research Progress, In-progress Actions, Enemy Information, Game Time}. We convert these observations into a concise text summary; an example summary is shown in Figure 20.

**Action Space.** Following the `BurnySc2/python-sc2` implementation, We define the action space as a discrete set of 72 high-level commands specifically for the Protoss race. These include unit training (e.g., Probes, Zealots, and Stalkers), building construction (e.g., Pylons and Gateways), research upgrades, scouting, multi-unit attacks or retreats, and special abilities (e.g., Chrono Boost). A complete list of these commands is provided in Figure 20. At each game step, the agent generates a list of five actions, which are executed in order.

## K.2  GAMEPLAY PROMPT FOR STARCRAFT II

Figure 20 shows the action inference prompt used by the 'zero-shot' agent for playing *StarCraft II*. The system prompt contains gameplay-specific knowledge and detailed instructions for action inference tailored to the Protoss race. It includes (1) the main task and game context, (2) a comprehensive

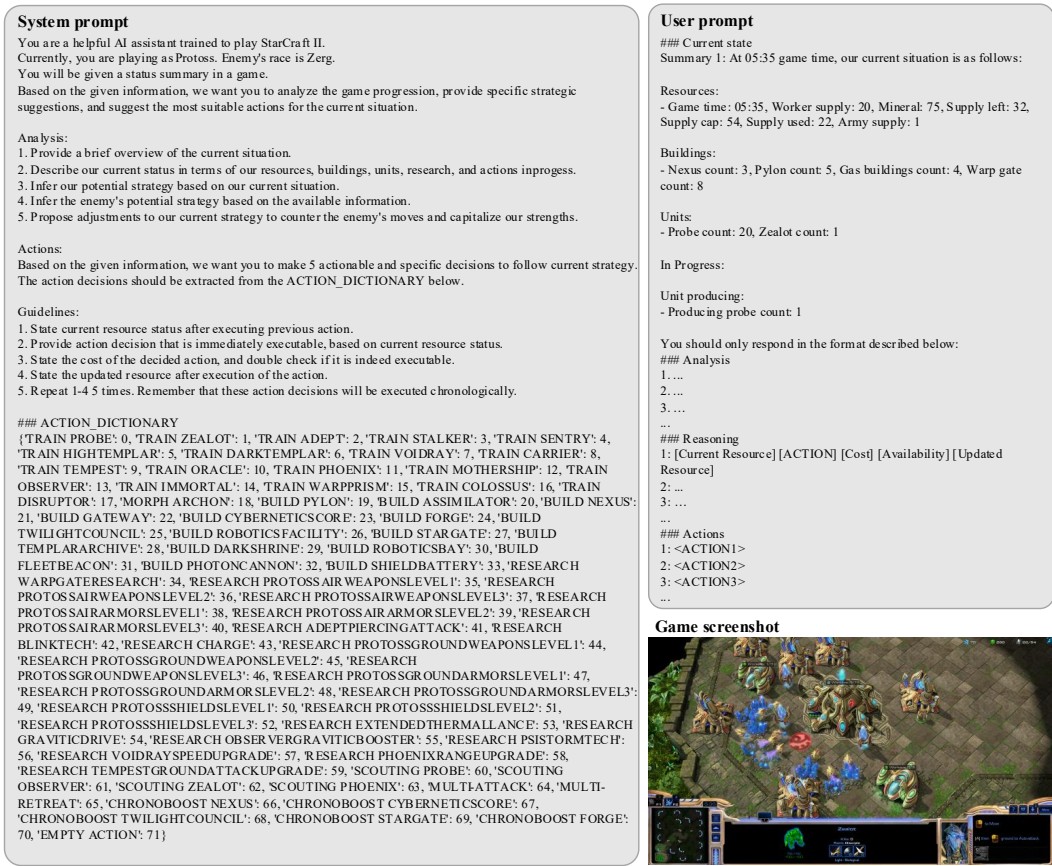

**System prompt**

You are a helpful AI assistant trained to play StarCraft II.
Currently, you are playing as Protoss. Enemy's race is Zerg.
You will be given a status summary in a game.
Based on the given information, we want you to analyze the game progression, provide specific strategic suggestions, and suggest the most suitable actions for the current situation.

Analysis:
1. Provide a brief overview of the current situation.
2. Describe our current status in terms of our resources, buildings, units, research, and actions inprogess.
3. Infer our potential strategy based on our current situation.
4. Infer the enemy's potential strategy based on the available information.
5. Propose adjustments to our current strategy to counter the enemy's moves and capitalize our strengths.

Actions:
Based on the given information, we want you to make 5 actionable and specific decisions to follow current strategy. The action decisions should be extracted from the ACTION_DICTIONARY below.

Guidelines:
1. State current resource status after executing previous action.
2. Provide action decision that is immediately executable, based on current resource status.
3. State the cost of the decided action, and double check if it is indeed executable.
4. State the updated resource after execution of the action.
5. Repeat 1-4 5 times. Remember that these action decisions will be executed chronologically.

### ACTION_DICTIONARY
{'TRAIN PROBE': 0, 'TRAIN ZEALOT': 1, 'TRAIN ADEPT': 2, 'TRAIN STALKER': 3, 'TRAIN SENTRY': 4, 'TRAIN HIGHTEMPLAR': 5, 'TRAIN DARKTEMPLAR': 6, 'TRAIN VOIDRAY': 7, 'TRAIN CARRIER': 8, 'TRAIN TEMPEST': 9, 'TRAIN ORACLE': 10, 'TRAIN PHOENIX': 11, 'TRAIN MOTHERSHIP': 12, 'TRAIN OBSERVER': 13, 'TRAIN IMMORTAL': 14, 'TRAIN WARPPRISM': 15, 'TRAIN COLOSSUS': 16, 'TRAIN DISRUPTOR': 17, 'MORPH ARCHON': 18, 'BUILD PYLON': 19, 'BUILD ASSIMILATOR': 20, 'BUILD NEXUS': 21, 'BUILD GATEWAY': 22, 'BUILD CYBERNETICSCORE': 23, 'BUILD FORGE': 24, 'BUILD TWILIGHTCOUNCIL': 25, 'BUILD ROBOTICSFACILITY': 26, 'BUILD STARGATE': 27, 'BUILD TEMPLARARCHIVE': 28, 'BUILD DARKSHRINE': 29, 'BUILD ROBOTICSBAY': 30, 'BUILD FLEETBEACON': 31, 'BUILD PHOTONCANNON': 32, 'BUILD SHIELDBATTERY': 33, 'RESEARCH WARPGATERESEARCH': 34, 'RESEARCH PROTOSSAIRWEAPONSLEVEL1': 35, 'RESEARCH PROTOSSAIRWEAPONSLEVEL2': 36, 'RESEARCH PROTOSSAIRWEAPONSLEVEL3': 37, 'RESEARCH PROTOSSAIRARMORSLEVEL1': 38, 'RESEARCH PROTOSSAIRARMORSLEVEL2': 39, 'RESEARCH PROTOSSAIRARMORSLEVEL3': 40, 'RESEARCH ADEPTPIERCINGATTACK': 41, 'RESEARCH BLINKTECH': 42, 'RESEARCH CHARGE': 43, 'RESEARCH PROTOSSGROUNDWEAPONSLEVEL1': 44, 'RESEARCH PROTOSSGROUNDWEAPONSLEVEL2': 45, 'RESEARCH PROTOSSGROUNDWEAPONSLEVEL3': 46, 'RESEARCH PROTOSSGROUNDARMORSLEVEL1': 47, 'RESEARCH PROTOSSGROUNDARMORSLEVEL2': 48, 'RESEARCH PROTOSSGROUNDARMORSLEVEL3': 49, 'RESEARCH PROTOSSSHIELDSLEVEL1': 50, 'RESEARCH PROTOSSSHIELDSLEVEL2': 51, 'RESEARCH PROTOSSSHIELDSLEVEL3': 52, 'RESEARCH EXTENDEDTHERMALLANCE': 53, 'RESEARCH GRAVITICDRIVE': 54, 'RESEARCH OBSERVERGRAVITICBOOSTER': 55, 'RESEARCH PSISTORMTECH': 56, 'RESEARCH VOIDRAYSPEEDUPGRADE': 57, 'RESEARCH PHOENIXRANGEUPGRADE': 58, 'RESEARCH TEMPESTGROUNDATTACKUPGRADE': 59, 'SCOUTING PROBE': 60, 'SCOUTING OBSERVER': 61, 'SCOUTING ZEALOT': 62, 'SCOUTING PHOENIX': 63, 'MULTI-ATTACK': 64, 'MULTI-RETREAT': 65, 'CHRONOBOOST NEXUS': 66, 'CHRONOBOOST CYBERNETICSCORE': 67, 'CHRONOBOOST TWILIGHTCOUNCIL': 68, 'CHRONOBOOST STARGATE': 69, 'CHRONOBOOST FORGE': 70, 'EMPTY ACTION': 71}

**User prompt**

### Current state
Summary 1: At 05:35 game time, our current situation is as follows:

Resources:
- Game time: 05:35, Worker supply: 20, Mineral: 75, Supply left: 32, Supply cap: 54, Supply used: 22, Army supply: 1

Buildings:
- Nexus count: 3, Pylon count: 5, Gas buildings count: 4, Warp gate count: 8

Units:
- Probe count: 20, Zealot count: 1

In Progress:

Unit producing:
- Producing probe count: 1

You should only respond in the format described below:
### Analysis
1. ...
2. ...
3. ...
...
### Reasoning
1: [Current Resource] [ACTION] [Cost] [Availability] [Updated Resource]
2: ...
3: ...
...
### Actions
1: <ACTION1>
2: <ACTION2>
3: <ACTION3>
...

**Game screenshot**

Figure 20: Action inference prompt for 'zero-shot' agent playing StarCraft II.

*action dictionary* listing all possible unit production, building construction, and research actions, (3) the current game status summary including resources, buildings, units, and ongoing actions, and (4) the expected output response format that guides the agent to provide a step-by-step analysis, reasoning, and 5 concrete actionable commands with cost and resource availability considerations.

The user prompt provides the current game state with detailed information on resources, supply, buildings, units, and ongoing unit production. Given this prompt, the agent infers the appropriate Protoss-specific actions to optimize the gameplay strategy against a Zerg opponent, focusing on resource management, army composition, and tech progression to counter the enemy effectively.

### K.3 EVALUATION METRIC FOR STARCRAFT II

**Single-Agent Play.** In the single-player mode, the agent competes in a series of matches against the AI bot opponent. It plays up to 4 matches, continuing until it either wins or loses. The evaluation metric is the win rate, calculated as:

$$\text{Score} = \frac{\text{Number of Wins}}{\text{Total Matches Played}} \times 100$$

**Multi-Agent Play.** In the multi-player mode, we do not use win rate as the performance metric, since only a single run is conducted per match. Instead, we measure the performance by the *army supply difference* between agents at the end of the match. The difference is calculated as the sum of each unit's count multiplied by its consumed resource cost, reflecting the effective army strength. The winning agent receives a positive score equal to this army supply difference, while the losing agent is assigned the negative of this value. This scoring method captures not only victory but also the margin of the win.

Using these difference-based scores, we then compute Elo ratings for all agents following the Bradley-Terry model (Bradley & Terry, 1952), following the approach used in *StreetFighter III* multi-agent evaluation in Section C.3.

The resulting *army supply difference* matrix and Elo scores are presented in Figure 4b.

### K.4   EXPERIMENTAL CONFIGURATION FOR STARCRAFT II

For all LLMs, we use a temperature of 0.1 and a repetition penalty of 1.0. During LLM inference, we pause the *StarCraft II* game environment. Interactions with the game environment are limited to a maximum of 1,000 steps. For single-agent play, we repeat the experiments 4 times and report the average score along with the standard deviation. For multi-agent play, we run a single experiment and report the Elo score.

### K.5   RESULT FOR STARCRAFT II

**Single-Agent Play.** We evaluate agent performances in a single-agent environment, all operating in a *ref-plan* setting. Table 35 shows a significant performance gap between open-source LLMs and proprietary LLMs. Among them, Llama-3.2-1B/3B models mostly repeat simple actions like scouting and mining minerals, without showing much strategic planning. On the other hand, proprietary LLMs, except for o3-mini, achieve over 50% win rate. Notably, GPT-4o and Gemini-2.5-pro won all four matches against the AI bot. They demonstrate strategic behavior by appropriately allocating resources over time in line with the game's progression.

Table 36 presents ablation studies on agentic modules. As previously mentioned, Llama-3.2-3B models fail to manage matches effectively regardless of ablation settings, resulting in a 0% win rate. In contrast, GPT-4o demonstrates remarkable planning abilities, which are critical in StarCraft II given its real-time strategy nature. In other words, long-term planning to sustain strategies over time plays a pivotal role in securing victories. Interestingly, models relying solely on reflection perform worse than zero-shot, suggesting that reflection without proper planning may actually degrade performance.

Table 37 presents the results of modality experiments. Surprisingly, for GPT-4o, using both text and image inputs results in decreased performance, and Gemini and Claude also show no improvement. This implies that agents can make sufficiently accurate decisions based on textual observations alone, and the addition of image input may introduce challenges in multimodal reasoning.

**Multi-Agent Play.** We evaluate agent performances in a multi-agent setting by conducting pairwise matches among seven LLMs (excluding Gemini-2.5-pro), all operating under a *ref-plan* configuration. The results are summarized in Figure 4(b). To ensure a fair comparison, both competing agents consistently use the Protoss race in every match.

Interestingly, unlike the single-agent evaluation where Claude achieved only a 50% win rate (Table 35), Claude outperforms GPT-4o and attains the highest Elo rating in the multi-agent arena. Even more surprising is that Minitron-8b, which had 0% win rate in single-agent play, defeated GPT-4o, o3-mini, and GPT-4o-mini, earning the second highest Elo rating. This discrepancy suggests that the presence of multiple intelligent agents can significantly alter game dynamics, potentially due to increased strategic diversity or emergent adversarial behaviors. We also acknowledge that a single evaluation episode may have introduced some bias in the observed rankings.

## L   SLAY THE SPIRE

### L.1   GAME DESCRIPTION FOR SLAY THE SPIRE

**Environment.** *Slay the Spire* is a deck-building rogue-like game where the player ascends a procedurally generated three-act tower. Each act consists of a branching map with various room types such as combat encounters, shops, treasure rooms, rest sites, and random events, ending in a boss fight.

Our goal is to evaluate an LLM agent's ability to reason over strategic choices in a stochastic, multi-step environment. Specifically, we task the agent with playing as the *Ironclad* character under standard rules (no ascension levels) and aim to defeat the final boss at floor 50. The LLM agent is responsible for two key decision types: (1) choosing which cards to play during combat, and (2)

| Models | StarCraft II | Rank |
|---|---|---|
| Llama-3.2-1B | $0.0_{\pm0.0}$ | 9.5 |
| Llama-3.2-3B | $0.0_{\pm0.0}$ | 9.5 |
| Qwen-2.5-3B | $0.0_{\pm0.0}$ | 9.5 |
| Qwen-2.5-7B | $0.0_{\pm0.0}$ | 9.5 |
| Minitron-4B | $0.0_{\pm0.0}$ | 9.5 |
| Minitron-8B | $0.0_{\pm0.0}$ | 9.5 |
| GPT-4o-mini | $75.0_{\pm50.0}$ | 3 |
| GPT-4o | $\mathbf{100.0}_{\pm0.0}$ | 1.5 |
| o3-mini | $25.0_{\pm50.0}$ | 6 |
| Gemini-2.5-pro | $\mathbf{100.0}_{\pm0.0}$ | 1.5 |
| Claude-3.7 | $50.0_{\pm57.7}$ | 4.5 |
| Deepseek-R1 | $50.0_{\pm57.7}$ | 4.5 |

Table 35: Gameplay score on *StarCraft II*.

| Models | Agent | StarCraft II | Rank |
|---|---|---|---|
| Llama-3B | Zero-shot | $0.0_{\pm0.0}$ | 6 |
| | Reflection | $0.0_{\pm0.0}$ | 6 |
| | Planning | $0.0_{\pm0.0}$ | 6 |
| | Ref-Plan | $0.0_{\pm0.0}$ | 6 |
| GPT-4o | Zero-shot | $50.0_{\pm57.7}$ | 2.5 |
| | Reflection | $0.0_{\pm0.0}$ | 6 |
| | Planning | $50.0_{\pm57.7}$ | 2.5 |
| | Ref-Plan | $\mathbf{100.0}_{\pm0.0}$ | 1 |

Table 36: Ablation study for agentic modules on *StarCraft II*.

| Models | Input | StarCraft II | Rank |
|---|---|---|---|
| GPT-4o | Text | $\mathbf{100.0}_{\pm0.0}$ | 2 |
| | Both | $50.0_{\pm57.7}$ | 5 |
| Gemini | Text | $\mathbf{100.0}_{\pm0.0}$ | 2 |
| | Both | $\mathbf{100.0}_{\pm0.0}$ | 2 |
| Claude | Text | $50.0_{\pm57.7}$ | 5 |
| | Both | $50.0_{\pm57.7}$ | 5 |

Table 37: Comparison across modalities on *StarCraft II*.

selecting rewards after battles. All other game decisions, such as map navigation, non-combat events, and potion usages, are handled by a simple rule-based policy.

The game runs on the *Steam* platform, and we use modding tools to enable communication between the game and an external agent. In particular, we use BaseMod (Bug Kiooeht, 2018a) and ModTheSpire (Bug Kiooeht, 2018b), along with a modified version of CommunicationMod (Forgotten Arbiter, 2019), which enables state extraction and action input via standard input/output streams. Our modified version also extracts detailed in-game information, including full card and relic descriptions.

To ensure consistency and reproducibility, we fix the game seed for all runs. This guarantees the same map layout, encounters, and card offerings. Additionally, since the original game unlocks card pools progressively as the player completes runs, we pre-unlock all cards using the *Unlock Everything* mod to make the full pool available from the start.

**Observation-to-Text Conversion.** After each action, we extract a JSON representation of the current game state, including player status, opponent status, current cards in hand, and relics. These states are serialized into a concise natural language summary that is passed to the LLM.

**Action Space.** Our environment defines two categories of actions: combat and card selection. During combat, the agent can either play a card (*PLAY*) or end its turn (*END*). In the card selection stage, the agent can choose a card reward (*CHOOSE*) or skip the reward (*SKIP*).

- `PLAY CARD_INDEX`: Play a non-target card from the hand at position `CARD_INDEX`.
- `PLAY CARD_INDEX TARGET_INDEX`: Play a targeted card from the hand at position `CARD_INDEX`, targeting opponent at `TARGET_INDEX`.
- `END`: End the current turn.
- `CHOOSE CARD_INDEX`: Select the card reward at position `CARD_INDEX`.
- `SKIP`: Skip the card reward.

## L.2 GAMEPLAY PROMPT FOR SLAY THE SPIRE

Figure 21 presents the action inference prompt utilized by the 'zero-shot' agent to play *Slay the Spire*. The system prompt describes the agent's role as a strategic player in the game and outlines key game mechanics, including block, energy, card draw, and enemy intents. Detailed game states, such as player status, relic, card, and opponents are provided in the user prompt.

## L.3 EVALUATION METRIC FOR SLAY THE SPIRE

The primary objective is to reach and defeat the final boss located on floor 50. Accordingly, the baseline score is determined by the highest floor cleared. For example, if the character dies on floor 43, the score would be 42. Since each floor has varying difficulty levels, and bosses—appearing at the end of each act—pose significant challenges, we assign bonus points to boss defeats. There are three major bosses and we grant an additional $50/3$ points for each boss defeated. Formally, the total score is defined as:

$$\text{Score} = (\text{\# of Cleared Floors}) + \frac{50}{3} \times (\text{\# of Bosses Defeated}) \tag{1}$$

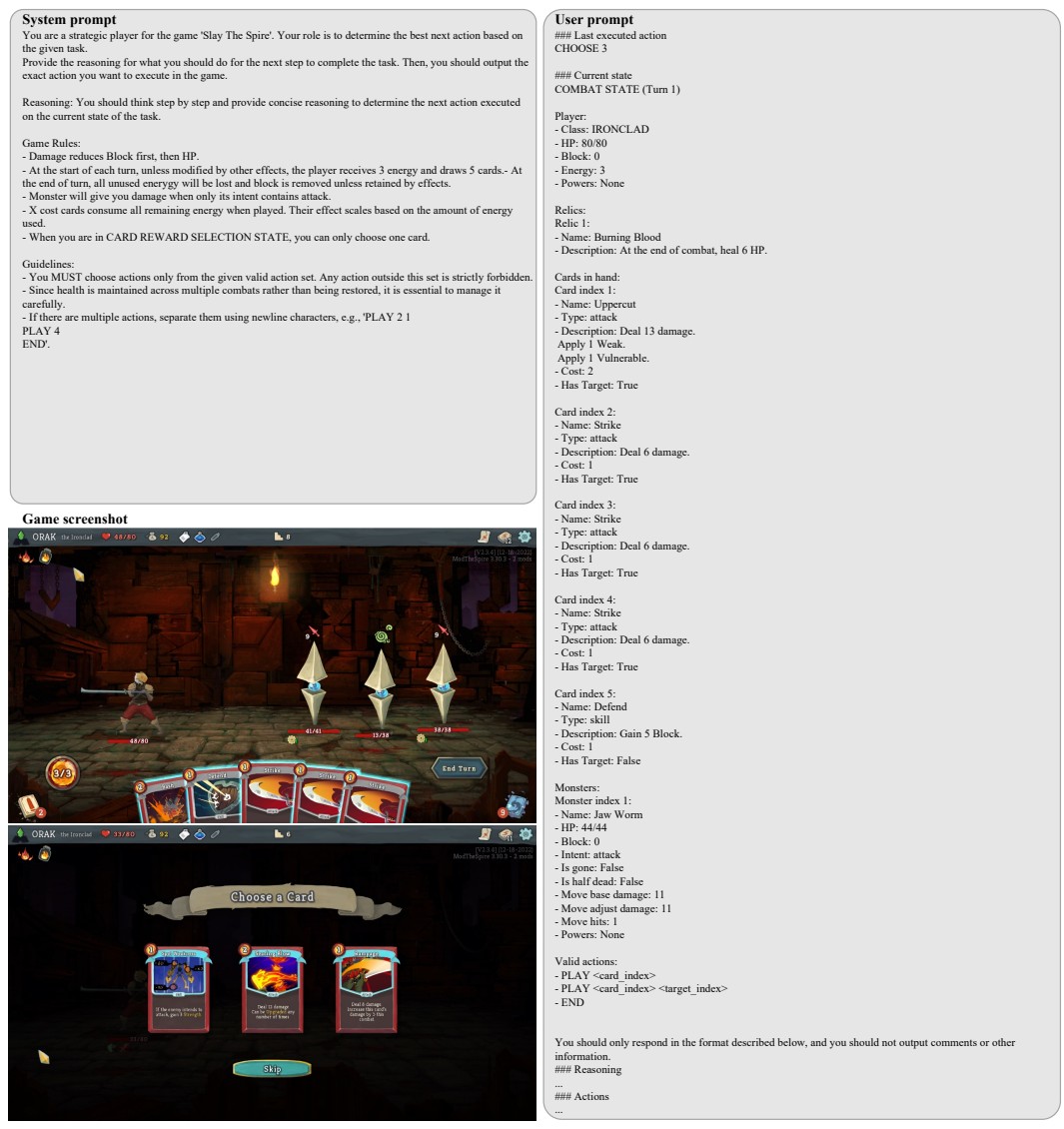

Figure 21: Action inference prompt for 'zero-shot' agent playing *Slay the Spire*.

### L.4 EXPERIMENTAL CONFIGURATION FOR SLAY THE SPIRE

For all LLMs, we use a temperature of 0.0 and a repetition penalty of 1. Interaction with the game environment is limited to a maximum of 200 steps. The game is not paused during LLM inference, as the game state does not change over time. All experiments carried out in three runs.

### L.5 RESULT FOR SLAY THE SPIRE

Table 38 shows a comparison in gameplay performance between open-source LLMs and proprietary LLMs in *Slay the Spire*. Most open-sourced models fail to make meaningful progress in the game, resulting in near-zero scores. In contrast, API-based LLMs demonstrate significantly stronger performance: Gemini-2.5-pro achieves the highest score, defeating the second boss in two out of three runs. The inherent stochasticity of the game contributes to high variance across all proprietary models.

Table 39 presents an ablation study evaluating the impact of agentic modules, reflection and planning, on performance in *Slay the Spire*. For the weaker model (Llama-3.2-3B), none of the agent variants achieve any meaningful progress, indicating that architectural changes alone are insufficient without

| Models | SlaySpire | Rank |
|---|---|---|
| Llama-3.2-1B | $0.0_{\pm 0.0}$ | 10 |
| Llama-3.2-3B | $0.0_{\pm 0.0}$ | 10 |
| Qwen-2.5-3B | $0.0_{\pm 0.0}$ | 10 |
| Qwen-2.5-7B | $5.0_{\pm 0.0}$ | 6 |
| Minitron-4B | $0.0_{\pm 0.0}$ | 10 |
| Minitron-8B | $0.0_{\pm 0.0}$ | 10 |
| GPT-4o-mini | $3.3_{\pm 2.9}$ | 7 |
| GPT-4o | $23.6_{\pm 22.1}$ | 3 |
| o3-mini | $15.0_{\pm 0.0}$ | 4.5 |
| Gemini-2.5-pro | $\mathbf{51.9}_{\pm 31.9}$ | 1 |
| Claude-3.7 | $15.0_{\pm 0.0}$ | 4.5 |
| Deepseek-R1 | $24.9_{\pm 17.1}$ | 2 |

Table 38: Gameplay score on *Slay the Spire*.

| Models | Agent | SlaySpire | Rank |
|---|---|---|---|
| Llama-3B | Zero-shot | $0.0_{\pm 0.0}$ | 6.5 |
| | Reflection | $0.0_{\pm 0.0}$ | 6.5 |
| | Planning | $0.0_{\pm 0.0}$ | 6.5 |
| | Ref-Plan | $0.0_{\pm 0.0}$ | 6.5 |
| GPT-4o | Zero-shot | $12.3_{\pm 4.6}$ | 4 |
| | Reflection | $\mathbf{26.2}_{\pm 19.4}$ | 1 |
| | Planning | $23.2_{\pm 14.2}$ | 3 |
| | Ref-Plan | $23.6_{\pm 22.1}$ | 2 |

Table 39: Ablation study for agentic modules on *Slay the Spire*.

| Models | Input | SlaySpire | Rank |
|---|---|---|---|
| GPT-4o | Text | $23.6_{\pm 22.1}$ | 3.5 |
| | Both | $23.6_{\pm 22.1}$ | 3.5 |
| Gemini | Text | $\mathbf{51.9}_{\pm 31.9}$ | 1 |
| | Both | $26.2_{\pm 19.4}$ | 2 |
| Claude | Text | $15.0_{\pm 0.0}$ | 5 |
| | Both | $9.7_{\pm 4.6}$ | 6 |

Table 40: Comparison across modalities on *Slay the Spire*.

strong base capabilities. In contrast, GPT-4o benefits substantially from added reasoning modules: the reflection variant achieves the highest score, while both planning and reflection-planning agents also outperform the zero-shot baseline. Notably, the planning module is relatively less effective, which may stem from the fact that optimal actions are highly sensitive to the opponent's intent, information that is only partially observable and difficult to predict multiple turns ahead. These results highlight that while base model capacity is a prerequisite, structured reasoning routines further enhance gameplay performance in complex decision-making environments.

Table 40 compares model performance in *Slay the Spire* when using either text-only input or a combination of text and vision inputs. In all three proprietary models, adding visual input does not improve performance—and in fact, often leads to degradation. This outcome is not entirely surprising, as crucial gameplay information, such as detailed card effects, relic descriptions, and power mechanics, is often absent in the game screenshot. As a result, the image input fails to provide meaningful utility and instead introduces ambiguity or redundancy, effectively acting as noise rather than useful context. These findings suggest that for structured, information-dense environments like *Slay the Spire*, high-quality textual representations remain the most reliable modality for LLM agents.

# M   BABA IS YOU

## M.1   GAME DESCRIPTION FOR BABA IS YOU

**Environment.** *Baba Is You* is a puzzle game in which players must discover and understand every rule and mechanic on their own, apart from the basic movement keys ('left', 'right', 'up', and 'down') (Hempuli, 2019). The game's defining feature is that the text tiles forming the rules can be pushed around, allowing the player to rewrite those rules on the fly. Every valid rule sentence must contain a verb (*e.g.*, 'Is' and 'Has'), and text tiles with a colored background (*e.g.*, 'You', 'Push' and 'Win') cannot serve as subjects. A level is cleared when the object designated by 'You' touches the object designated by 'Win'. For instance, with the rules 'Baba Is You' and 'Flag Is Win', the player wins as soon as Baba touches the flag. For imple-

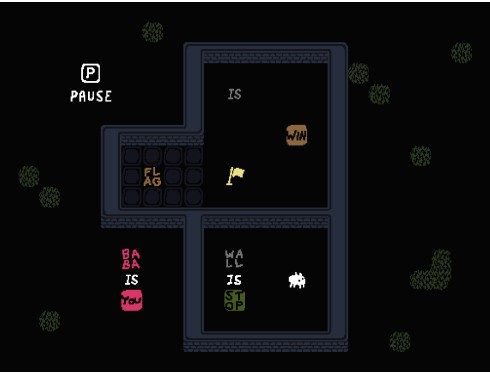

Figure 22: `Level 1` of *Baba Is You*.

mentation, we instrument the Steam edition with a lightweight Lua mod that, after every player move, dumps the full internal game state, *i.e.*, the coordinates of every object, to a JSON file. The Lua modding script uses mod hook functions provided by the game developer, which are triggered at specific points in the game's code. The agent outputs are delivered to the game via simulated key presses using the *pyautogui* library. We use `Level 1 - Where do I go?`, shown in Figure 22, as our *default evaluation setting* (for Table 3).

**Observation-to-Text Conversion.** We present the LLM with a textual description of the game state in two parts. First, we list the current $(x, y)$ coordinates of every object in the level. Second, we

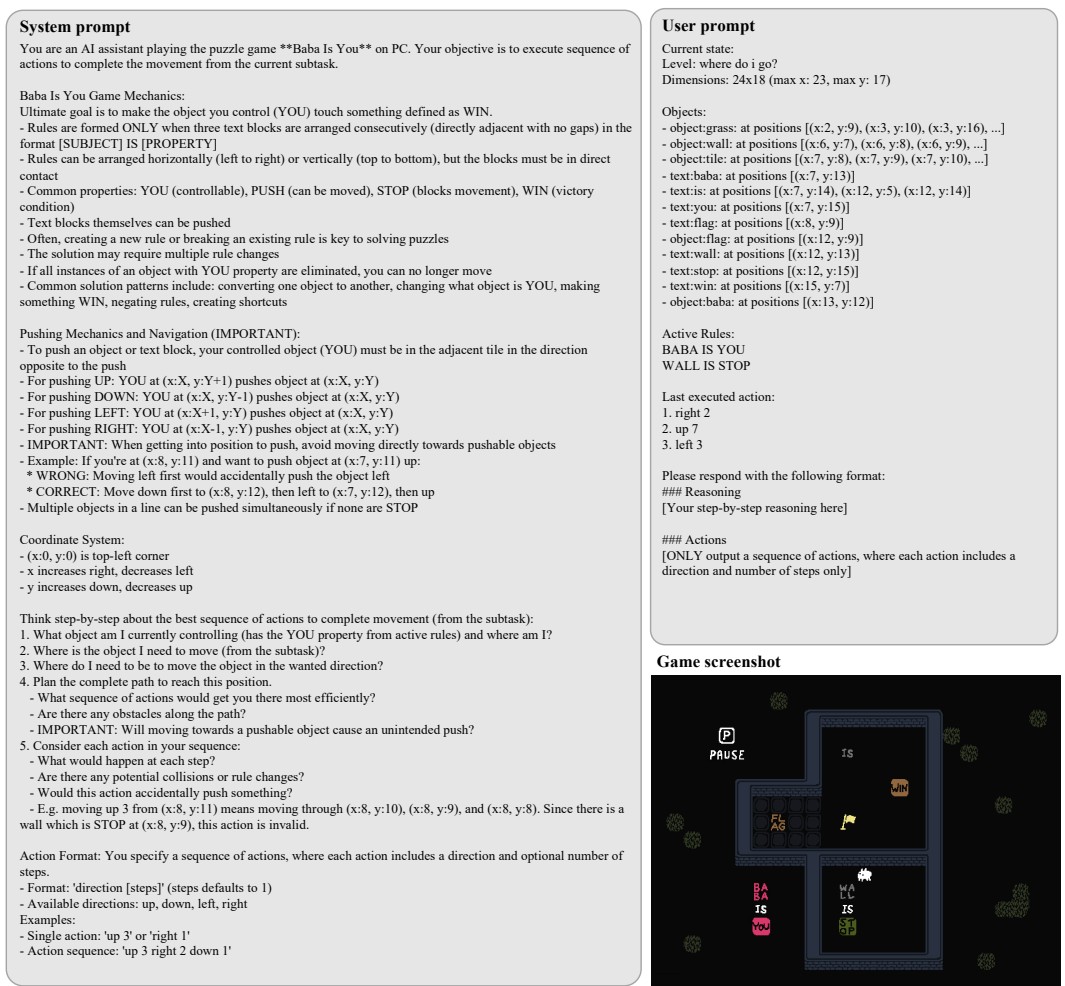

Figure 23: Action inference prompt for 'zero-shot' agent playing *Baba Is You*.

manually parse the state to extract all active rules and append those rules to the prompt. The combined description forms the model's observation.

**Action Space.** In the 2D grid environment, the agent can move 'left', 'right', 'up', or 'down'. At each step, the agent outputs a finite sequence of these directions, which we translate into consecutive key presses and send to the game.

## M.2   GAMEPLAY PROMPT FOR BABA IS YOU

Figure 23 shows the action inference prompt used by the 'zero-shot' agent for playing *Baba Is You*. The system prompt encodes key game-specific knowledge, including (1) the primary objective of touching a 'Win' object with a controllable ('You') object, (2) an explanation of how rules are formed and manipulated via text block arrangements, (3) crucial pushing mechanics with coordinate-based guidance to avoid unintended interactions, and (4) the expected input-output format for the LLM. The user prompt provides the current puzzle state, including the map dimensions, object locations, and active rules. Given this prompt, the LLM agent outputs a sequence of directional actions.

## M.3   EVALUATION METRIC FOR BABA IS YOU

To evaluate the performance of an LLM agent, we define a hierarchical scoring metric that rewards meaningful progress toward solving the puzzle. As shown in Figure 22, a key prerequisite is breaking the rule 'Wall Is Stop', which enables movement out of the closed area. The second subgoal is creating a winning condition, such as forming the rule 'Flag Is Win', but this is only possible once the

| Models | BabaIsYou | Rank |
|---|---|---|
| Llama-3.2-1B | $6.7_{\pm 11.5}$ | 12 |
| Llama-3.2-3B | $20.0_{\pm 0.0}$ | 6.5 |
| Qwen-2.5-3B | $13.3_{\pm 11.5}$ | 10.5 |
| Qwen-2.5-7B | $20.0_{\pm 0.0}$ | 6.5 |
| Minitron-4B | $20.0_{\pm 0.0}$ | 6.5 |
| Minitron-8B | $20.0_{\pm 0.0}$ | 6.5 |
| GPT-4o-mini | $13.3_{\pm 11.5}$ | 10.5 |
| GPT-4o | $20.0_{\pm 0.0}$ | 6.5 |
| o3-mini | $73.3_{\pm 46.2}$ | 1.5 |
| Gemini-2.5-pro | $73.3_{\pm 46.2}$ | 1.5 |
| Claude-3.7 | $46.7_{\pm 46.2}$ | 3 |
| Deepseek-R1 | $20.0_{\pm 0.0}$ | 6.5 |

Table 41: Gameplay score on *Baba Is You*.

| Models | Agent | BabaIsYou | Rank |
|---|---|---|---|
| Llama-3B | Zero-shot | $20.0_{\pm 0.0}$ | 4.5 |
| | Reflection | $20.0_{\pm 0.0}$ | 4.5 |
| | Planning | $20.0_{\pm 0.0}$ | 4.5 |
| | Ref-Plan | $20.0_{\pm 0.0}$ | 4.5 |
| GPT-4o | Zero-shot | $20.0_{\pm 0.0}$ | 4.5 |
| | Reflection | $20.0_{\pm 0.0}$ | 4.5 |
| | Planning | $20.0_{\pm 0.0}$ | 4.5 |
| | Ref-Plan | $20.0_{\pm 0.0}$ | 4.5 |

Table 42: Ablation study for agentic modules on *Baba Is You*.

| Models | Input | BabaIsYou | Rank |
|---|---|---|---|
| GPT-4o | Text | $20.0_{\pm 0.0}$ | 6 |
| | Image | $6.7_{\pm 13.7}$ | 9 |
| | Both | $20.0_{\pm 0.0}$ | 6 |
| Gemini | Text | $73.3_{\pm 46.2}$ | 2 |
| | Image | $20.0_{\pm 0.0}$ | 6 |
| | Both | $86.7_{\pm 23.1}$ | 1 |
| Claude | Text | $46.7_{\pm 46.2}$ | 3 |
| | Image | $20.0_{\pm 0.0}$ | 6 |
| | Both | $20.0_{\pm 0.0}$ | 6 |

Table 43: Comparison across modalities on *Baba Is You*.

wall constraint is removed. Each subgoal provides 20 points. If the agent clears the level by having the 'You' object touch a 'Win' object, it receives a full score of 100, overriding subgoal rewards. The final score is computed as:

$$
\text{Score} = \begin{cases} 100, & \text{if level is cleared} \\ 40, & \text{if 'Wall Is Stop' is broken and 'Win' rule is created} \\ 20, & \text{if 'Wall Is Stop' is broken only} \end{cases}
$$

### M.4 EXPERIMENTAL CONFIGURATION FOR BABA IS YOU

For all 6 open-source LLMs, we use a temperature of 0.7 and a repetition penalty of 1, and set the maximum number of game steps to 30. We run all experiments with 3 trials and report the average score with the standard deviation.

### M.5 RESULT FOR BABA IS YOU

The performance of different models on `Level 1` of *Baba Is You* using the 'reflection-planning' agent is shown in Table 41. We observe that o3-mini and Gemini-2.5-pro achieve the highest score of 73.3, significantly outperforming all other models. Apart from these two reasoning models, only Claude-3.7-sonnet, a hybrid reasoning model, scores above 20.0, indicating that it is the only other model capable of constructing a valid winning condition. In contrast, all non-reasoning models, including every open-source small language model we tested, typically only managed to break the 'Wall Is Stop' rule, consistently earning a score of 20.0. However, a qualitative analysis of the models' self-defined subtasks and reasoning traces suggests that these models often fail to infer that breaking 'Wall Is Stop' is a necessary prerequisite for constructing the winning condition. This implies that their success in breaking the rule was largely unintentional.

Table 42 presents an ablation study evaluating the impact of agentic modules on performance in *Baba Is You*. Across both Llama-3.2-3B and GPT-4o, we observe no measurable improvement over the zero-shot baseline, with all configurations achieving the same score of 20.0. This indicates that the agentic components do not significantly enhance the agent's ability to reach the winning condition for these two models. The task remains challenging, largely due to the model's limited spatial reasoning capabilities. While the models occasionally produce valid high-level plans, such as to form the rule 'Flag Is Win' at specific coordinates, they frequently fail to account for the game's pushing mechanics. As a result, they often push text tiles in unintended directions, breaking or misaligning the intended rule formation.

Table 43 presents an ablation study on input modalities across several multimodal models. We observe that relying solely on image input significantly degrades performance for all models. Adding image input on top of text yields only marginal improvements, if any, suggesting that the agents primarily rely on text-based representations to make decisions. Notably, Gemini-2.5-pro benefits slightly from the combined input, achieving the highest score of 86.7.

## N  2048

### N.1  GAME DESCRIPTION FOR 2048

**Environment.** *2048* (Cirulli, 2014) is a single-player sliding tile puzzle game played on a 4×4 grid. The objective is to combine numbered tiles by sliding them in one of four directions (i.e., up, down, left, or right) to create a tile with the value 2048. In this environment, the agent observes the current board state, represented as a 4×4 matrix of integers (each cell contains 0 for empty or a power of 2 for active tiles), and selects one of four discrete actions corresponding to directional moves. While the original game ends when no moves are available (i.e., the board is full and no adjacent tiles can be merged), we additionally terminate the episode if the agent performs five consecutive invalid moves (i.e., actions that result in no change to the board state). For implementation, we use an open-source, Pygame-based game environment. The `logic` module manages the board state, tile movements, merging, and win/loss conditions, while the interface leverages Pygame to render the board and handle user input. The implementation supports dynamic resizing and configurable parameters, and is designed to facilitate both human play and automated experiments.

**Observation-to-Text Conversion.** The environment's board state can be directly transformed into a textual description, formatting it as a 4×4 array of integers in which each element indicates the value of the corresponding tile.

**Action Space.** The action space comprises four discrete actions: 'up', 'down', 'left', and 'right', each representing a possible direction in which the agent can slide the tiles on the board.

### N.2  GAMEPLAY PROMPT FOR 2048

Figure 24 shows the action inference prompt and the corresponding game screenshot used by the zero-shot agent to play 2048. The system prompt includes (1) the main objective of the game, (2) detailed game rules, and (3) the expected input-output format between the LLM and the environment. The user prompt provides (1) the specific task for the 2048 game, (2) the previous board state and game score, (3) the last executed action, (4) the current board state and game score, and (5) the expected output format. Based on this information, the agent determines the next action to take.

### N.3  EVALUATION METRIC FOR 2048

The goal of *2048* is to create a tile with the value 2048. The game score increases as tiles are merged, with the value of the merged tile added to the total score. Although the score when the 2048 tile is created can slightly vary depending on the gameplay, it is generally estimated that the score is around 20,000 points. Therefore, we define the evaluation metric as the progress to the target score of 20,000, normalized to 100. Formally, the normalized score is defined as:

$$\text{Score} = \min\left(\frac{\text{Final Game Score}}{20{,}000} \times 100, 100\right).$$

where 'Final Game Score' denotes the total score at the end of the game. This metric reflects how close the agent came to achieving the primary objective of creating the 2048 tile.

### N.4  EXPERIMENTAL CONFIGURATION FOR 2048

For all 6 open-source LLMs, including Llama-3.2-1B/3B, Qwen-2.5-3B/7B, and Minitron-4B/8B, we use a temperature of 0.0 and a repetition penalty of 1.0. We set the maximum number of game steps to 10,000. However, the maximum number of game steps (10,000) was never reached in the experiments. The game typically terminated either when the board was full and no adjacent tiles could be merged, or when the agent failed to take an action that changed the board state for more than five consecutive steps. In the best gameplay episode of o3 zero-shot agent, which achieved a score of 57.32, a total of 685 steps were taken to reach this result.

### N.5  RESULT FOR 2048

All reported results in Tables 44–46 are computed as the mean and standard deviation over five independent runs, using a 'zero-shot' agent for each model configuration. To establish a baseline for

**System prompt**

You are an expert AI agent specialized in playing the 2048 game with advanced strategic reasoning. Your primary goal is to achieve the highest possible tile value while maintaining long-term playability by preserving the flexibility of the board and avoiding premature game over.

### 2048 Game Rules ###
1. The game is played on a 4×4 grid. Tiles slide in one of four directions: 'up', 'down', 'left', or 'right'.
2. Only two consecutive tiles with the SAME value can merge. Merges cannot occur across empty tiles.
3. Merging is directional:
   - Row-based merges occur on 'left' or 'right' actions.
   - Column-based merges occur on 'up' or 'down' actions.
4. All tiles first slide in the chosen direction as far as possible, then merges are applied.
5. A tile can merge only once per move. When multiple same-value tiles are aligned (e.g., [2, 2, 2, 2]), merges proceed from the movement direction. For example:
   - [2, 2, 2, 2] with 'left' results in [4, 4, 0, 0].
   - [2, 2, 2, 0] with 'left' results in [4, 2, 0, 0].
6. An action is only valid if it causes at least one tile to slide or merge. Otherwise, the action is ignored, and no new tile is spawned.
7. After every valid action, a new tile (usually 90 percent chance of 2, 10 percent chance of 4) appears in a random empty cell.
8. The game ends when the board is full and no valid merges are possible.
9. Score increases only when merges occur, and the increase equals the value of the new tile created from the merge.

### Decision Output Format ###
Analyze the provided game state and determine the single most optimal action to take next.
Return your decision in the following exact format:

### Reasoning
<a detailed summary of why this action was chosen>
### Actions
<up, right, left, or down>

Ensure that:
   - The '### Reasoning' field provides a clear explanation of why the action is the best choice, including analysis of current tile positions, merge opportunities, and future flexibility.
   - The '### Actions' field contains only one of the four valid directions.

**User prompt**

### Target task
Merge tiles to make a tile with the value of 2048

### Previous state
Board of 2048 Games:
[2, 0, 0, 0]
[4, 0, 0, 0]
[2, 2, 0, 0]
[16, 4, 2, 0]
Score: 52

### Last executed action
left

### Current state
Board of 2048 Games:
[2, 0, 0, 0]
[4, 0, 2, 0]
[4, 0, 0, 0]
[16, 4, 2, 0]
Score: 56

You should only respond in the format described below, and you should not output comments or other information.
Provide your response in the strict format:
### Reasoning
<a detailed summary of why this action was chosen>
### Actions
<direction>

**Game screenshot**

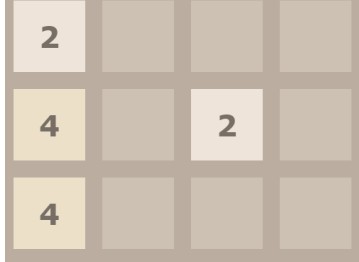

Figure 24: Action inference prompt for 'zero-shot' agent playing 2048.

| Models | 2048 | Rank |
|---|---|---|
| Random Agent | $5.5_{\pm 2.3}$ | 6 |
| Llama-3.2-1B | $0.0_{\pm 0.1}$ | 15 |
| Llama-3.2-3B | $0.3_{\pm 0.2}$ | 12 |
| Qwen-2.5-3B | $0.1_{\pm 0.1}$ | 13 |
| Qwen-2.5-7B | $0.6_{\pm 0.4}$ | 11 |
| Minitron-4B | $0.1_{\pm 0.0}$ | 14 |
| Minitron-8B | $0.7_{\pm 0.7}$ | 10 |
| GPT-4o-mini | $1.1_{\pm 1.0}$ | 9 |
| GPT-4o | $5.6_{\pm 1.5}$ | 5 |
| o3-mini | $25.3_{\pm 7.3}$ | 2 |
| o4-mini | $15.7_{\pm 6.7}$ | 3 |
| o3 | $\mathbf{34.9}_{\pm 23.4}$ | 1 |
| Gemini-2.5-pro | $5.1_{\pm 2.5}$ | 8 |
| Claude-3.7 | $5.3_{\pm 2.7}$ | 7 |
| Deepseek-R1 | $11.5_{\pm 3.4}$ | 4 |

Table 44: Gameplay score on *2048*.

| Models | Agent | 2048 | Rank |
|---|---|---|---|
| Random Agent | - | $5.5_{\pm 2.3}$ | 6 |
| Llama-3B | Zero-shot | $\mathbf{0.3}_{\pm 0.2}$ | 8 |
| | Reflection | $0.0_{\pm 0.0}$ | 11 |
| | Planning | $0.1_{\pm 0.1}$ | 10 |
| | Ref-Plan | $0.1_{\pm 0.2}$ | 9 |
| GPT-4o | Zero-shot | $5.6_{\pm 1.5}$ | 5 |
| | Reflection | $3.5_{\pm 2.9}$ | 7 |
| | Planning | $6.0_{\pm 5.5}$ | 4 |
| | Ref-Plan | $\mathbf{7.0}_{\pm 5.7}$ | 3 |
| o3-mini | Zero-shot | $\mathbf{25.3}_{\pm 7.3}$ | 1 |
| | Ref-Plan | $17.6_{\pm 9.5}$ | 2 |

Table 45: Ablation study for agentic modules on *2048*.

| Models | Input | 2048 | Rank |
|---|---|---|---|
| Random Agent | - | $5.5_{\pm 2.3}$ | 5 |
| GPT-4o | Text | $\mathbf{5.6}_{\pm 1.5}$ | 3 |
| | Image | $1.8_{\pm 1.1}$ | 10 |
| | Both | $5.4_{\pm 4.5}$ | 6 |
| Gemini | Text | $5.1_{\pm 2.5}$ | 8 |
| | Image | $\mathbf{5.5}_{\pm 2.4}$ | 4 |
| | Both | $3.1_{\pm 2.6}$ | 9 |
| Claude | Text | $5.3_{\pm 2.7}$ | 7 |
| | Image | $\mathbf{8.4}_{\pm 4.0}$ | 1 |
| | Both | $6.7_{\pm 0.9}$ | 2 |

Table 46: Comparison across modalities on *2048*.

comparison, we additionally included a random agent that selects one of the four possible actions (up, down, left, right) uniformly at random. This agent was evaluated over 50 episodes, and its performance is summarized in Tables.

**Gameplay score on 2048.** Beyond the 12 models presented in the main paper, we also include results for OpenAI's more recent models, o4-mini and o3 in Table 44. Interestingly, none of the open-source models (e.g., Llama, Qwen, Minitron) were able to correctly interpret the 2D array

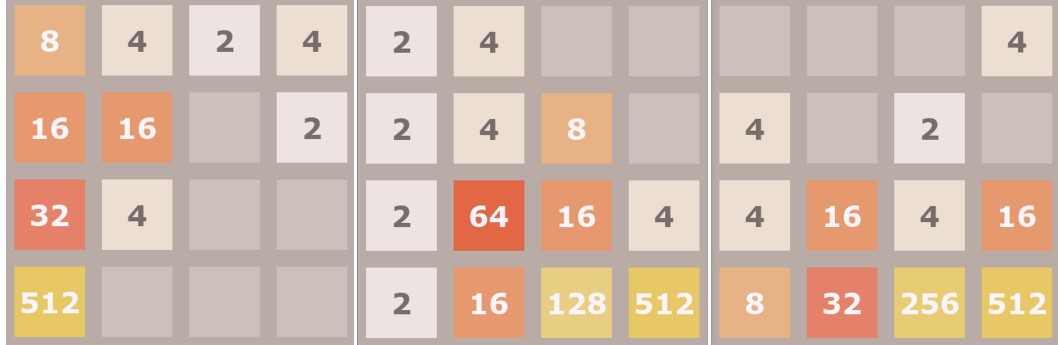

Figure 25: Three different runs for o3 zero-shot agent playing *2048*.

prompt representing the game board. Consequently, they repeatedly issued the same action even when no tiles could be merged, leading to premature termination of the game.

In contrast, GPT-4o, Gemini-2.5-pro, and Claude-3.7 were able to detect invalid moves and avoid them to some extent. However, these models failed to manage merged tiles into their planning, often repeating superficially valid actions that quickly led to a board lock-up. As a result, their gameplay performance was almost indistinguishable from that of the random agent.

Notably, the reasoning-capable models—o3-mini, o4-mini, o3, and DeepSeek-R1—exhibited meaningful gameplay performance. While the open-source and non-reasoning models were typically limited to producing a maximum tile of 64 or 128, o3-mini successfully generated the 512 tile in 4 out of 5 runs, and o3 achieved the 1024 tile in 2 out of 5 runs.

A particularly interesting observation is that, even without explicit strategic instructions (e.g., cornering high-value tiles or arranging tiles in a staircase pattern), the reasoning-based models implicitly discovered human-like strategies based solely on the basic game rules provided via system prompts. This behavior is illustrated in Figure 25, where o3 exhibits an emergent form of spatial organization akin to that used by experienced human players.

**Ablation study for agentic modules on 2048.** In addition to the models evaluated in the main paper, we conducted an extended ablation study using the o3-mini model to assess the impact of agentic modules—Reflection and Planning—on gameplay performance in the 2048 environment. Table 45 presents the results of this study across three representative models: Llama-3.2-3B, GPT-4o, and o3-mini, under various agent configurations.

For Llama-3.2-3B and GPT-4o, enabling the Reflection or Planning modules—individually or together—did not yield significant gains, suggesting that external agentic scaffolding alone is insufficient for improving performance, likely due to limited reasoning or architectural constraints.

In contrast, o3-mini exhibited a counter-intuitive trend: the zero-shot agent significantly outperformed both the Reflection and the Reflection-Planning agents. A plausible explanation for this phenomenon is that reasoning-centric models such as o3-mini may already perform internal reflection and planning when generating a response. As such, the introduction of additional agentic modules may introduce redundancy or even propagate errors in intermediate reasoning steps, degrading performance.

Supporting this hypothesis, Figure 26 shows the inference output of o3 shortly before achieving a 1024-tile. The output reveals that the model implicitly forecasts and sequences the next three actions, essentially performing internal planning without explicit agent modules. This emergent behavior highlights the potential of foundation models with strong reasoning capabilities to handle multi-step decision problems autonomously without external agentic augmentation.

**Comparison across modalities on 2048.** To assess the impact of input modality on gameplay performance, we evaluated models using either textual or visual (image-based) representations of the 2048 board state. The results are summarized in Table 46. Interestingly, GPT-4o demonstrated weaker performance with image inputs compared to text, suggesting that its image understanding capabilities—at least within the structured context of 2048—may lag behind its text comprehension. This is consistent with prior findings that GPT-4o, while multimodal, exhibits varying levels of alignment across modalities depending on task complexity and structure. In contrast, both Gemini

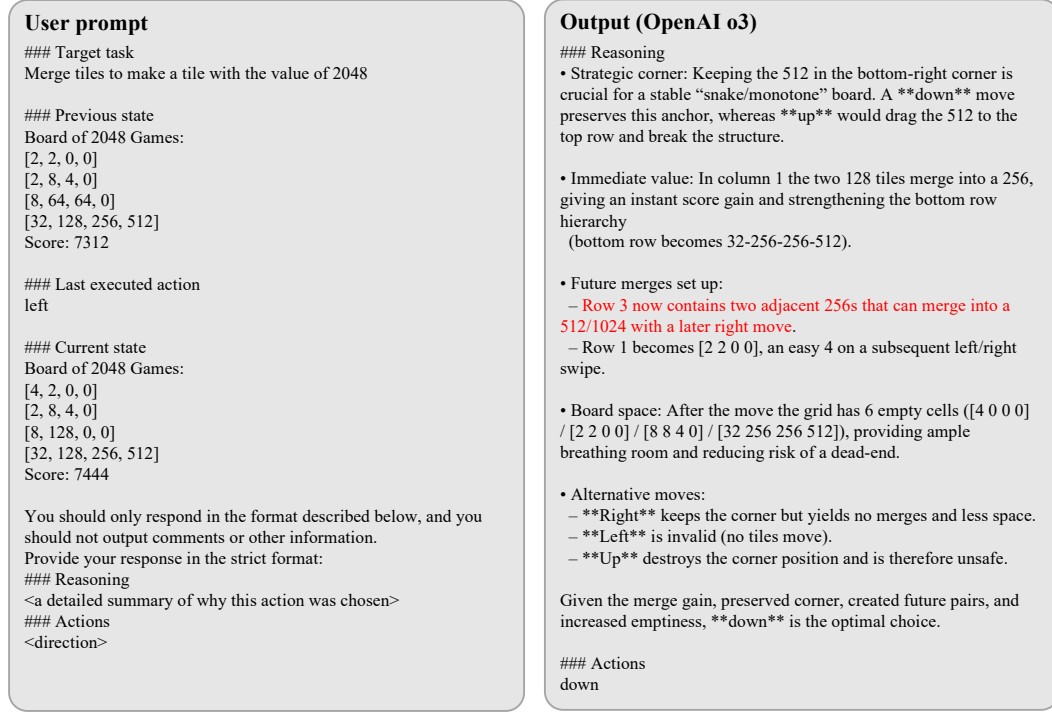

Figure 26: Output of OpenAI's o3 model right before producing the 1024 tile in *2048*.

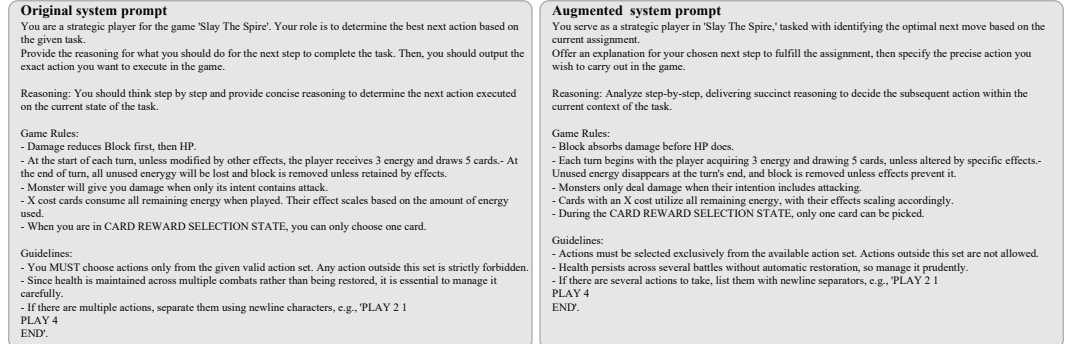

Figure 27: Example of original system prompt and augmented system prompt in *Slay the Spire*.

2.5 Pro and Claude 3.7 achieved better performance with image-based inputs than with text. This indicates a stronger visual reasoning capability in these models, particularly when parsing structured 2D spatial layouts such as the 2048 board. Their ability to interpret and act on visual patterns appears to be more robust than their capacity to process raw 2D arrays expressed as textual input.These findings highlight that multimodal models exhibit non-uniform modality strengths, and task-specific evaluations are crucial for selecting the appropriate input format to maximize agent performance.

## O   DETAILS FOR DATA AUGMENTATION

**Prompting Details.** To increase the diversity of our training dataset and avoid overfitting risks, we applied a data augmentation strategy focused on the system prompt. The original dataset was constructed by rolling out the same game prompt multiple times in the environment to collect assistant responses. The system prompt remains static, and the user prompt exhibits variations due to changes in game state, though these were constrained to a fixed format where only specific values changed.

To address this issue, we performed augmentation on the system prompt, which contains general information such as the LLM's role, game rules, and behavioral guidelines. Since the user prompt was

dependent on dynamic game states, modifying it risked introducing inconsistencies or hallucinations. Therefore, we chose to keep the user prompt fixed during augmentation.

We use GPT-4o to generate paraphrased versions of the original system prompt. Specifically, we prompted the model to produce 10 alternative phrasings of the original system prompt that preserved its semantics while varying its linguistic expression. Each paraphrased system prompt was then paired with the original user prompt and assistant response, resulting in 10 augmented versions of each original data point. Including the original, this expands the dataset by a factor of 11. Figure 27 presents an example of the original and augmented prompt in *Slay the Spire*.

This augmentation strategy helped increase the syntactic and lexical diversity of the dataset while preserving semantic fidelity and coherence, thereby supporting more robust fine-tuning of the LLM.

| # of Augs. | SuperMario | 2048 |
|:---:|:---:|:---:|
| 0 | 24.2 | 1.4 |
| 3 | 26.8 | 3.5 |
| 10 | 26.7 | 2.8 |
| 20 | 25.9 | 2.4 |

Table 47: Performance across different numbers of augmentations.

**Effect of The Number of Augmentations.** Table 47 shows the fine-tuned model performance across different numbers of augmentations. We use the same fine-tuning setup as the intra-game generalization in Table 7. Overall, increasing the number of augmentations improves the performance of fine-tuned models compared to the one with no augmentation. Interestingly, the small number of augmentations (3 or 10) is quite effective to ensure linguistic diversity in fine-tuning, and the effectiveness diminishes as the number of augmentations grows.

## P    IMPLEMENTATION DETAILS

### P.1    ASYNCHRONOUS INFERENCES FOR MULTI-AGENT ENVIRONMENT

Here we provide implementation details for multi-agent game environment that enables efficient, scalable, and realistic agent interaction.

**Overview.** The core design principle is to execute the game loop on a frame-by-frame basis while allowing each agent to asynchronously infer and initiate its next high-level action upon completion of the previous one. This approach is particularly well-suited for real-time, frame-based games such as *Street Fighter III* and *StarCraft II*, where individual actions may span a variable number of frames (e.g., a 'Move Away' action takes 4 frames, whereas a 'Super Attack' requires 7 frames to complete). In this system, each agent operates independently, without waiting for other agents to complete their actions. As soon as an agent completes its current action, it observes the current game state and determines its next high-level action. This asynchronous execution allows the game to progress fluidly and continuously, closely mimicking the dynamics of real-time multiplayer games.

**Illustrative Example.** Consider a multi-agent scenario in *Street Fighter III*. Agent 1 selects a 'Super Attack' that spans 7 frames, while Agent 2 chooses a 'Move Away' action that lasts 4 frames. The game orchestrator initiates both actions simultaneously. After 4 frames, Player 2's action concludes, triggering the agent to observe the updated state and select a new action, *e.g.*, a 'Medium Punch' lasting 2 frames. The orchestrator then integrates this new action into the ongoing simulation, even as Agent 1's "Super Attack" continues. This frame-by-frame orchestration proceeds iteratively. Each agent re-enters the decision-making process immediately upon completing its current action, independent of the progress or status of other agents.

**Advantages.** This asynchronous, non-blocking scheduling model provides several key advantages. It enables agents to operate with varying action durations without artificial synchronization barriers, facilitating a more natural and responsive interaction dynamic. The resulting system supports overlapping actions, better reflects the timing complexities of real-time games, and can be readily extended to support environments with more than two agents.

## P.2   FINE-TUNING

We conducted supervised fine-tuning using collected gameplay data to adapt the LLM agent to game-specific reasoning and interactions with environments.

| Task | LLaMA-3.2-1B(h) | LLaMA-3.2-3B(h) |
|---|---|---|
| Intra-game/Non-game | 1.7 | 4.2 |
| OOD-game | 1.7 | 4.3 |

Table 48: Elapsed GPU time for fine-tuning.

**Training Configuration.** We fine-tune two models: Llama-3.2-1B and Llama-3.2-3B. Training is conducted using 4 NVIDIA A100 GPUs with 80GB of memory each. We use a learning rate of 1e-6, a per-device batch size of 4 with gradient accumulation steps set to 4, resulting in an effective batch size of 64. The models were trained for 1 epoch with 100 warm-up steps. Table 48 summarizes the GPU time taken for fine-tuning.

**Data Statistics.** In total, we use 105,502 data points for fine-tuning. Only data points containing fewer than 4,096 tokens are used for training to ensure compatibility with model input length limitations. All data points from *Pokémon* and 329 out of 9900 data points from *Minecraft* are discarded due to length constraint. For the out-of-distribution (OOD) game generalization experiments, we exclude the games 2048 and Super Mario, resulting in 87,660 data points used. Apart from the number of data points, the training configuration remained identical.

## P.3   UNSEEN SCENARIOS

To evaluate the intra-game generalization capability of fine-tuned LLM agents, we define a separate scenario that is not used during the training dataset collection. Unseen scenarios differs from the seen ones by featuring a different character, map, or stage. Figure 28 presents example screenshots of the seen and unseen scenarios across six games.

**Street Fighter III.** While the character `Ken` was used during training data collection, a different character, `Chun-Li`, was introduced at evaluation time to assess intra-game generalization. Chun-Li features a distinct set of moves and hitboxes, posing a significantly different control and tactical challenge compared to `Ken`. The game environment and match settings remained unchanged.

**Darkest Dungeon.** While the first expedition used during training data collection featured a party composed of a `Plague Doctor`, `Vestal`, `Highwayman`, and `Crusader`, the unseen scenario involved a different party composition: one `Man-At-Arms`, two `Grave Robbers`, and one `Vestal`. This change introduced significantly different combat dynamics, synergies, and positional requirements. Additionally, the enemy pool in this dungeon included the `Madman`—a monster not encountered during training—known for its stress-inducing attacks and erratic behavior. All other gameplay parameters, including the dungeon type (short dungeon in Ruins area), remained unchanged.

**StarCraft II.** During training, agents were exposed to the 'Ancient Cistern LE' map, while evaluation was conducted on a different map, 'Babylon LE' to assess intra-game generalization. All other game settings, including player race (Protoss), opponent race (Zerg), opponent's build order strategy (Timing), and difficulty level (Hard), were kept constant.

**Slay the Spire.** We used game runs with different random seeds as unseen scenarios. Each game seed determines the layout of the map, including the sequence of opponents, bosses, events, and card reward options. All other game parameters, such as character choice (*IronClad*) and starting deck, were held constant.

**Baba Is You.** In the case of *Baba Is You*, the second stage, `Level 2 - Now what is this?`, is used as an unseen scenario. This stage features new rule combinations and object interactions that are absent from the training set, thereby testing the agent's ability to generalize to novel logic structures. The game mechanics and control scheme remained unchanged.

# Q  ANALYSIS OF THE GAP BETWEEN IN-GAME AND OOD-GAMES IN FINE-TUNING

| Genre | Action | Adventure | RPG | Simulation | Strategy | Puzzle |
|---|---|---|---|---|---|---|
| Games | SuperMario | AceAttorney | DarkestD | MineCraft | SlaySpire | 2048 |
| Llama-3.2-1B (Pretrain) | 18.7 | 1.3 | 0.0 | 0.0 | 0.0 | 0.1 |
| Llama-3.2-1B (Fine-tune) | 23.5 | 5.5 | 0.0 | 0.0 | 0.0 | 0.3 |
| Llama-3.2-3B (Pretrain) | 31.8 | 4.6 | 47.5 | 0.0 | 0.0 | 0.1 |
| Llama-3.2-3B (Fine-tune) | 24.9 | 9.1 | 40.1 | 0.0 | 0.0 | 3.1 |
| Avg. Improvement (%) | 2.0 | 210.5 | -7.8 | 0.0 | 0.0 | 1600.0 |

Table 49: Performance of LLaMA models with varying in-ood game gap.

**Setup.** For studying the in-OOD gap in generalization, we fine-tune Llama-3.2 only on *Baba Is You*. Other hyperparameter configurations are exactly the same as in Appendix P.

**Results.** As shown in Table 49, fine-tuning yields the most improvements for OOD games in the same genre, but can also provide benefits across different genres in some cases. For example, 2048, another puzzle game, showed the largest improvement after fine-tuning. Interestingly, Ace Attorney, though a different genre, also benefited from fine-tuning, likely due to shared requirements in logical reasoning (see qualitative examples below). In contrast, for games like *Darkest Dungeon* and *Super Mario*, which involve different gameplay dynamics and action spaces, the effect was mixed. Note that, as shown in Table 7, unlike the current experiment (fine-tuned on *Baba Is You* only), LLMs fine-tuned on 10 games (without *Super Mario*) showed consistent improvements on Super Mario (OOD game). This implies our fine-tuning set, constructed from a diverse set of games across genres, contains knowledge broadly beneficial for gameplay/decision-making and supports cross-genre generalizability.

# R  UNIFIED PERFORMANCE COMPARISON

Table 50 compares the overall LLM/VLM agent results across agentic modules and modalities. Interestingly, one of the LLM agent that are using text-only states performs the best for each game.

# S  STATISTICAL SIGNIFICANCE ANALYSIS

This section reports three complementary statistical analyses: (1) per-game confidence intervals, (2) per-game significance tests using Welch's t-test (Welch, 1947), and (3) across-game paired t-tests (Fisher, 1925).

**Per-Game Confidence Intervals**. For each game, we report the sample mean $\bar{x}$, sample standard deviation $s$, the number of seeds $n$, and the two–sided $95\%$ confidence interval half–width ("CI (95%)") computed as

$$\text{CI}_{95\%} = t_{0.975,\,n-1} \cdot \frac{s}{\sqrt{n}}.$$

Table 51 summarizes the results for GPT-4o.

**Per-Game Significance Tests**. We compare GPT-4o against Llama-3.2-3B and Qwen-2.5-72B on each game using Welch's t-test (Welch, 1947). As shown in Table 52, GPT-4o and Llama-3.2-3B shows significant or deterministic differences in 10 out of 12 games, while GPT-4o and Qwen-2.5-72B shows significant or deterministic differences in 7 out of 12 games. Here, "Deterministic" indicates both models have zero empirical variance and unequal means; "deterministic tie" indicates both zero variance and equal means.

**Across–Game Paired Tests**. To assess aggregate differences across games, we conduct paired t-tests (Fisher, 1925) by pairing per-game means for each model. As shown in Table 53, GPT-4o significantly outperforms both Llama-3.2-1B and Qwen-2.5-72B.

| Model | Action | | Adventure | | RPG | | Simulation | | Strategy | | Puzzle | |
|---|---|---|---|---|---|---|---|---|---|---|---|---|
| | SF3 | SuperMario | AceAttorney | HerStory | Pokémon | DarkestD | Minecraft | Stardew | StarCraft2 | SlaySpire | BabaIsYou | 2048 |
| **Main Benchmark Results (Text-only, Default Agent)** | | | | | | | | | | | | |
| Llama-3.2-1B | 0.0±0.0 | 18.7±8.6 | 1.3±2.2 | 2.1±1.2 | 0.0±0.0 | 0.0±0.0 | 0.0±0.0 | 0.0±0.0 | 0.0±0.0 | 0.0±0.0 | 6.7±11.5 | 0.0±0.1 |
| Llama-3.2-3B | 13.3±5.8 | 31.8±10.1 | 4.6±1.3 | 4.2±1.1 | 0.0±0.0 | 47.5±39.2 | 0.0±0.0 | 0.0±0.0 | 0.0±0.0 | 0.0±0.0 | 20.0±0.0 | 0.3±0.2 |
| Qwen-2.5-3B | 20.0±0.0 | 23.4±14.1 | 20.0±17.4 | 1.2±1.1 | 0.0±0.0 | 44.8±22.2 | 0.0±0.0 | 0.0±0.0 | 0.0±0.0 | 0.0±0.0 | 13.3±11.5 | 0.1±0.1 |
| Qwen-2.5-7B | 16.7±11.5 | 27.2±9.6 | 9.3±0.2 | 8.5±2.0 | 0.0±0.0 | 88.8±2.0 | 0.0±0.0 | 0.0±0.0 | 0.0±0.0 | 5.0±0.0 | 20.0±0.0 | 0.6±0.4 |
| Minitron-4B | 16.7±11.5 | 24.4±6.0 | 35.7±4.5 | 4.6±2.3 | 0.0±0.0 | 0.0±0.0 | 0.0±0.0 | 0.0±0.0 | 0.0±0.0 | 0.0±0.0 | 20.0±0.0 | 0.1±0.0 |
| Minitron-8B | 23.3±5.8 | 31.3±12.8 | 29.9±3.6 | 8.2±1.8 | 0.0±0.0 | 63.8±30.4 | 0.0±0.0 | 0.0±0.0 | 0.0±0.0 | 0.0±0.0 | 20.0±0.0 | 0.7±0.7 |
| Llama-3.3-70B | **33.3**±28.9 | 32.0±11.8 | 53.9±23.4 | 46.4±10.8 | 16.7±14.4 | 87.2±8.2 | 0.0±0.0 | 40.9±37.2 | 0.0±0.0 | 5.0±0.0 | 20.0±0.0 | 1.4±1.3 |
| Qwen2.5-72B | 26.7±5.8 | 29.9±8.7 | 14.9±8.5 | 41.0±1.7 | 27.8±9.6 | 84.0±6.6 | 0.0±0.0 | 18.0±16.7 | 0.0±0.0 | 9.0±5.3 | 20.0±0.0 | 0.2±0.1 |
| GPT-4o-mini | 16.7±11.5 | 28.8±8.8 | 28.4±2.8 | 21.2±5.6 | 0.0±0.0 | 81.3±5.8 | 46.0±7.0 | 16.1±27.8 | 75.0±50.0 | 3.3±2.9 | 13.3±11.5 | 1.1±1.0 |
| GPT-4o | 29.7±14.3 | 34.1±14.2 | **85.3**±1.5 | 64.2±5.2 | 38.9±9.6 | 93.4±1.5 | 71.0±7.0 | 81.4±4.8 | **100.0**±0.0 | 23.6±22.1 | 20.0±0.0 | 5.6±1.5 |
| GPT-5 | 19.5±11.0 | 31.6±10.5 | 59.1±26.5 | **74.9**±1.7 | **88.9**±4.8 | 92.9±0.7 | 73.0±7.0 | **92.3**±8.6 | 25.0±50.0 | 26.2±19.4 | **100.0**±0.0 | 10.2±2.1 |
| o3-mini | **33.3**±15.3 | 34.9±14.6 | 91.7±1.5 | 66.5±3.6 | 0.0±0.0 | 89.0±2.1 | **75.0**±0.0 | 55.1±16.0 | 25.0±50.0 | 15.0±0.0 | 73.3±46.2 | **25.3**±7.3 |
| Gemini-2.5-pro | 13.3±11.5 | **38.0**±14.4 | 55.7±3.4 | 67.5±3.3 | 83.3±0.0 | **93.7**±1.6 | **75.0**±0.0 | 59.2±10.1 | **100.0**±0.0 | **51.9**±31.9 | 73.3±46.2 | 5.1±2.5 |
| Claude-3.7 | 16.7±11.5 | 31.7±8.2 | 81.9±1.6 | 62.9±2.6 | 63.9±19.2 | 89.9±2.5 | **75.0**±0.0 | 53.6±20.9 | 50.0±57.7 | 15.0±0.0 | 46.7±46.2 | 5.3±2.7 |
| Deepseek-R1 | 20.0±0.0 | 28.7±13.2 | 83.3±1.5 | 67.2±3.9 | **75.0**±0.0 | 91.7±1.1 | 41.7±0.0 | 66.1±11.5 | 50.0±57.7 | 24.9±17.1 | 20.0±0.0 | 11.5±3.4 |
| **Agentic Module Ablation (LLaMA-3B / GPT-4o)** | | | | | | | | | | | | |
| LLaMA-3B (Zeroshot) | 13.3±5.8 | 21.2±8.2 | 5.7±3.2 | 4.2±1.1 | 0.0±0.0 | 47.5±39.2 | 0.0±0.0 | 0.0±0.0 | 0.0±0.0 | 0.0±0.0 | 20.0±0.0 | 0.3±0.2 |
| LLaMA-3B (Reflection) | 30.0±17.3 | 32.4±8.6 | 4.6±1.3 | 4.4±1.3 | 0.0±0.0 | 47.3±39.0 | 0.0±0.0 | 0.0±0.0 | 0.0±0.0 | 0.0±0.0 | 20.0±0.0 | 0.0±0.0 |
| LLaMA-3B (Planning) | 20.0±0.0 | 27.0±8.4 | 4.6±1.3 | 5.2±1.0 | 0.0±0.0 | 56.3±23.6 | 0.0±0.0 | 0.0±0.0 | 0.0±0.0 | 0.0±0.0 | 20.0±0.0 | 0.1±0.1 |
| LLaMA-3B (Ref-Plan) | 16.7±20.8 | 31.8±10.1 | 3.8±0.0 | 5.4±0.4 | 0.0±0.0 | 57.0±31.6 | 0.0±0.0 | 0.0±0.0 | 0.0±0.0 | 0.0±0.0 | 20.0±0.0 | 0.1±0.2 |
| GPT-4o (Zeroshot) | 29.7±14.3 | 29.6±9.2 | 49.9±1.3 | 64.2±5.2 | 33.3±0.0 | 93.4±1.5 | 0.0±0.0 | 34.5±29.9 | 50.0±57.7 | 24.7±9.2 | 20.0±0.0 | 5.6±1.5 |
| GPT-4o (Reflection) | 23.3±20.8 | 32.3±15.5 | **85.3**±1.5 | 61.5±0.4 | 36.1±4.8 | 85.2±10.3 | 50.0±0.0 | 15.6±4.5 | 0.0±0.0 | 41.3±19.6 | 20.0±0.0 | 3.5±2.9 |
| GPT-4o (Planning) | 30.0±26.5 | 29.4±12.5 | 52.7±0.5 | 59.4±5.7 | 33.3±0.0 | 82.0±8.6 | 13.0±0.0 | 54.9±20.2 | 50.0±57.7 | 35.3±9.2 | 20.0±0.0 | 6.0±5.5 |
| GPT-4o (Ref-Plan) | 23.3±20.8 | 34.1±14.2 | 52.8±0.5 | 62.0±4.9 | 38.9±9.6 | 91.6±2.5 | 50.0±0.0 | 81.4±4.8 | **100.0**±0.0 | 36.0±25.5 | 20.0±0.0 | 7.0±5.7 |
| **Modality Ablation (Text / Image / Both)** | | | | | | | | | | | | |
| Qwen2.5-VL-7B (Text) | 0.0±0.0 | 26.0±8.2 | 10.0±0.0 | 3.2±1.5 | 2.8±4.8 | 82.7±1.5 | 0.0±0.0 | 0.0±0.0 | 0.0±0.0 | 0.0±0.0 | 13.3±11.5 | 0.1±0.1 |
| Qwen2.5-VL-7B (Image) | 3.3±5.8 | 25.1±10.1 | – | – | – | – | – | 0.0±0.0 | – | – | 0.0±0.0 | 0.2±0.3 |
| Qwen2.5-VL-7B (Both) | 3.3±5.8 | 25.6±8.0 | 17.6±13.1 | 2.5±0.8 | 2.8±4.8 | 81.8±3.0 | – | 0.0±0.0 | 0.0±0.0 | 0.0±0.0 | 0.0±0.0 | 0.4±0.5 |
| Qwen2.5-VL-32B (Text) | 16.7±5.8 | 32.0±10.2 | 71.9±21.5 | 15.4±2.8 | 27.8±9.6 | 89.9±1.5 | 0.0±0.0 | 15.2±22.7 | 0.0±0.0 | 0.0±0.0 | 20.0±0.0 | 0.2±0.3 |
| Qwen2.5-VL-32B (Image) | **33.3**±20.8 | 25.8±10.7 | – | – | – | – | – | 0.0±0.0 | – | – | 13.3±11.5 | 0.4±0.3 |
| Qwen2.5-VL-32B (Both) | **33.3**±15.3 | 29.7±6.6 | 68.6±10.0 | 17.0±4.6 | 25.0±14.4 | 91.0±3.0 | – | 26.9±13.9 | 0.0±0.0 | 0.0±0.0 | 20.0±0.0 | 0.2±0.3 |
| GPT-4o (Text) | 29.7±14.3 | 34.1±14.1 | **85.3**±1.5 | 64.2±5.2 | 38.9±9.6 | 93.4±1.5 | 71±0.0 | 81.4±4.8 | **100.0**±0.0 | 23.6±22.1 | 20.0±0.0 | 5.6±1.5 |
| GPT-4o (Image) | 23.7±15.9 | 27.1±13.7 | – | – | – | – | – | 0.0±0.0 | – | – | 6.7±11.5 | 1.8±1.1 |
| GPT-4o (Both) | 24.3±14.5 | 27.1±10.2 | 53.5±1.7 | 40.6±29.5 | 41.7±8.3 | 92.2±3.0 | – | 41.9±22.1 | 50.0±57.7 | 23.6±22.1 | 20.0±0.0 | 5.4±4.5 |
| Gemini-2.5 (Text) | 13.3±11.5 | **38.0**±13.4 | 55.7±3.4 | 67.5±3.3 | 83.3±0.0 | **93.7**±1.6 | **75.0**±0.0 | 59.2±10.1 | **100.0**±0.0 | **51.9**±31.9 | 73.3±46.2 | 5.1±2.5 |
| Gemini-2.5 (Image) | 16.7±11.5 | 28.8±10.7 | – | – | – | – | – | 7.6±9.0 | – | – | 20.0±0.0 | 5.5±2.4 |
| Gemini-2.5 (Both) | 20.0±10.0 | 40.9±9.6 | 52.6±0.8 | 64.9±2.4 | 83.3±0.0 | 92.2±1.8 | – | 60.0±6.0 | **100.0**±0.0 | 26.2±19.4 | 86.7±23.1 | 3.1±2.6 |
| Claude-3.7 (Text) | 16.7±11.5 | 28.7±13.2 | 81.9±1.6 | 62.9±2.6 | 63.9±19.2 | 89.9±2.5 | **75.0**±0.0 | 53.6±20.9 | 50.0±57.7 | 15.0±0.0 | 46.7±46.2 | 5.3±2.7 |
| Claude-3.7 (Image) | 23.3±11.5 | 25.6±6.4 | – | – | – | – | – | 0.0±0.0 | – | – | 20.0±0.0 | 8.4±4.0 |
| Claude-3.7 (Both) | **33.3**±5.8 | 22.6±6.3 | 71.3±17.3 | 63.6±3.1 | 72.2±4.7 | 90.1±5.7 | – | 49.8±1.0 | 50.0±57.7 | 9.7±4.6 | 20.0±0.0 | 6.7±0.9 |
| **Human Novice** | | | | | | | | | | | | |
| Human (Novice) | 20.0±20.0 | 100.0±0.0 | 87.8±2.6 | 72.6±2.8 | 86.1±12.7 | 90.3±2.0 | 70.8±0.0 | 69.8±19.9 | 33.3±57.7 | 52.9±23.2 | 100.0±0.0 | 22.7±10.4 |

Table 50: Unified LLM/VLM results of Orak: (Top) Main performance (all models), (Middle) Agentic module ablation (LLaMA-3B / GPT-4o), (Bottom) Modality ablation (Text/Image/Both). The best values for each game are marked in bold, and missing Image results are shown as "–".

| Model | Metric | SF3 | SuperMario | AceAttorney | HerStory | Pokémon | DarkestD | Minecraft | Stardew | StarCraft2 | SlaySpire | BabaIsYou | 2048 |
|---|---|---|---|---|---|---|---|---|---|---|---|---|---|
| – | # seed | 5 | 20 | 3 | 3 | 3 | 3 | 3 | 3 | 4 | 3 | 3 | 5 |
| GPT-4o | mean | 29.7 | 34.1 | 85.3 | 64.2 | 38.9 | 93.4 | 71.0 | 81.4 | 100.0 | 23.6 | 20.0 | 5.6 |
| | std | 14.3 | 14.2 | 1.5 | 5.2 | 9.6 | 1.5 | 7.0 | 4.8 | 0.0 | 22.1 | 0.0 | 1.5 |
| | CI (95%) | 17.76 | 6.65 | 3.73 | 12.92 | 23.85 | 3.73 | 17.39 | 11.92 | 0.00 | 54.90 | 0.00 | 1.86 |

Table 51: Per-game statistics for GPT-4o. "CI (95%)" is the two-sided confidence interval.

# T  DISCUSSION FOR INTEGRATING ORAK WITH MULTI-TURN RL FINE-TUNING FRAMEWORKS

Orak can be readily integrated with existing multi-turn RL fine-tuning frameworks for LLM agents (Wang et al., 2025b; Zeng et al., 2025). In game-based environments, rewards are often *sparse*, being provided only at the end of an episode (e.g., final score). Therefore, *careful reward design* and *dynamic data extraction* are crucial for stable and effective optimization.

**Reward Design.** We discuss two types of reward designs.

(1) Reward Discounting. Following common practices in reinforcement learning, we apply reward discounting to estimate intermediate rewards during gameplay:

$$R_t = \gamma^{T-t} S_{\text{final}}, \tag{2}$$

where $0 < \gamma < 1$ denotes the discount factor and $S_{\text{final}}$ is the final game score.

(2) Turn-Based Rewards (when available). For games that provide intermediate feedback, such as *2048* or *Super Mario*, a turn-level reward can be incorporated as:

$$R_t = \lambda S_t + (1 - \lambda)\gamma^{T-t} S_{\text{final}}, \tag{3}$$

| Game | t-stat | p-value | Interpretation |
|---|---|---|---|
| SF3 | 4.644 | 0.0097 | **significant** |
| Super Mario | 4.149 | 0.0002 | **significant** |
| Ace Attorney | 54.641 | $<0.0001$ | **significant** |
| Her Story | 20.155 | 0.0015 | **significant** |
| Pokémon | 7.018 | 0.0197 | **significant** |
| Darkest Dungeon | 107.849 | 0.0001 | **significant** |
| Minecraft | 17.568 | 0.0032 | **significant** |
| Stardew Valley | 29.373 | 0.0012 | **significant** |
| StarCraft II | – | – | **deterministic** |
| Slay the Spire | 1.850 | 0.2056 | not significant |
| Baba Is You | 2.003 | 0.1831 | not significant |
| 2048 | 8.329 | 0.0011 | **significant** |

(a) GPT-4o vs. Llama-3.2-3B (Welch's t-test).

| Game | t-stat | p-value | Interpretation |
|---|---|---|---|
| SF3 | 0.435 | 0.681 | not significant |
| Super Mario | 1.128 | 0.268 | not significant |
| Ace Attorney | 14.127 | 0.0039 | **significant** |
| Her Story | 7.345 | 0.0104 | **significant** |
| Pokémon | 1.416 | 0.230 | not significant |
| Darkest Dungeon | 2.406 | 0.126 | not significant |
| Minecraft | 17.568 | 0.0032 | **significant** |
| Stardew Valley | 6.320 | 0.0164 | **significant** |
| StarCraft II | – | – | **deterministic** |
| Slay the Spire | 1.113 | 0.371 | not significant |
| Baba Is You | – | – | **deterministic tie** |
| 2048 | 8.032 | 0.0013 | **significant** |

(b) GPT-4o vs. Qwen-2.5-72B (Welch's t-test).

Table 52: Per-game significance tests (Welch's t-test).

| Comparison | t-stat | p-value | Interpretation |
|---|---|---|---|
| GPT-4o vs. Llama-3.2-1B | 5.227 | 0.00028 | **significant** |
| GPT-4o vs. Qwen-2.5-72B | 3.122 | 0.00971 | **significant** |

Table 53: Cross-game significance tests (paired t-test).

where $\lambda$ is a balancing coefficient and $S_t$ is the turn-level score. This formulation allows smoother credit assignment across gameplay turns.

**Dynamic Data Extraction.** During fine-tuning, the model repeatedly performs *data rollouts*, *reward estimation*, and *multi-turn RL optimization* across multiple episodes. As the policy improves, it naturally encounters increasingly challenging game scenarios, promoting *progressive data extraction* and richer experience replay. After each episode, validation accuracy is compared between the current and previous policy models, and the superior one is retained. This iterative selection procedure contributes to stable and robust policy improvement over time.

## U    LIMITATIONS AND FUTURE WORK

**Cost Consideration.** Among all games in Orak, six games, *i.e.*, *Ace Attorney*, *Her Story*, *Darkest Dungeon*, *Stardew Valley*, *Slay the Spire*, and *Baba is You*, require a one-time purchase, typically priced ranging from $9.99 to $24.99. While this represents a non-negligible upfront cost, it is relatively minor compared to the recurring cost associated with proprietary LLM API calls. From a cost-efficiency perspective, the benchmark remains accessible and practical for sustained research.

**License Issue.** We have made considerable efforts to comply with licensing requirements in designing our benchmark. (1) Users must purchase the commercial games themselves for evaluation. (2) We do not distribute any commercial game executables. (3) We do not modify any game assets during gameplay. Moreover, we explicitly state the following legal compliance guidelines "*This project is strictly for research purposes. Researchers are required to use our framework solely as a benchmark for evaluating LLMs, and any models derived from our benchmark are strictly prohibited from commercial use*".

**Real-time Gameplay.** *Street Fighter III*, *Super Mario*, and *StarCraft II* inherently require real-time gaming, unlike other turn-based or simulation games in Orak. However, in our current evaluation setup, the game is paused during LLM inference to remove the impact of real-time constraints on agent performance. While this allows for more stable evaluation of reasoning capabilities, real-time responsiveness is critical in many gaming contexts, so it should be handled for practical needs. We leave latency-aware evaluation protocols for building real-time gaming LLM agents as future work.

**Study on RL-based Fine-tuning.** Although RL-based fine-tuning has demonstrated strong performance in many domains, such as mathematics and programming, we did not explore it in this study. Given the interactive nature of the Orak environments, it would be natural to derive dynamic, context-aware rewards from in-game feedback and apply RL fine-tuning methods such as DPO (Rafailov et al., 2023) or GRPO (Shao et al., 2024). Unlike domains such as mathematics or programming,

where problems typically have static correct solutions derived through logical reasoning, gameplay requires *strategic reasoning* that adapts dynamically to the actions of other agents. In games, the optimal action is often contingent on the evolving behavior of the player or opponents, reflecting the complex, interactive nature of multi-agent environments. This distinction is particularly relevant to real-world domains such as business, economics, and negotiation, where strategic decision-making is frequently modeled using game-theoretic frameworks (Zhang et al., 2024). Therefore, leveraging Orak as an environment for RL-based fine-tuning may significantly enhance an LLM's capacity to reason strategically and operate effectively in multi-agent settings. This line of research holds promise for improving LLM performance in a broad range of real-world applications where understanding multi-agent dynamics is essential.

**Support for Diverse Modalities.** Orak supports evaluation of LLMs and VLMs, but it does not extend to other modalities often essential in real-world gameplay. One key example is *sound*. For instance, in *Minecraft*, players rely on zombie audio cues to avoid danger, while in *Street Fighter III* and *StarCraft II*, alert sounds indicate some specific attacks by the enemy. Also, beyond games in Orak, First-Person Shooter (FPS) games usually use gunfire sounds for spatial awareness, and horror games often rely on audio to signal the proximity of threats. Consequently, incorporating other modalities such as audio remains an open challenge, and benchmarking the performance of emerging Speech-Language Models (Chu et al., 2023; Cui et al., 2024) in gaming scenarios could be an important step toward broader multimodal agent development.

**Societal Impact.** Simulation games, *Minecraft* and *Stardew Valley*, offer rich environments where players can explore, mine resources, and craft items, enabling life-like simulations of human behavior. These games offer a valuable testbed for analyzing long-horizon behavior of LLM agents by systematically comparing gameplay trajectories of humans and LLM agents, which enables a rigorous assessment of whether LLMs exhibit human-like decision-making patterns. Introducing multiple agents into these environments allows for the study of emergent social behaviors among LLM agents. We believe such settings are particularly well-suited for precisely measuring the social impact of complex agent behaviors, offering valuable insights into the dynamics of LLM-based agents.

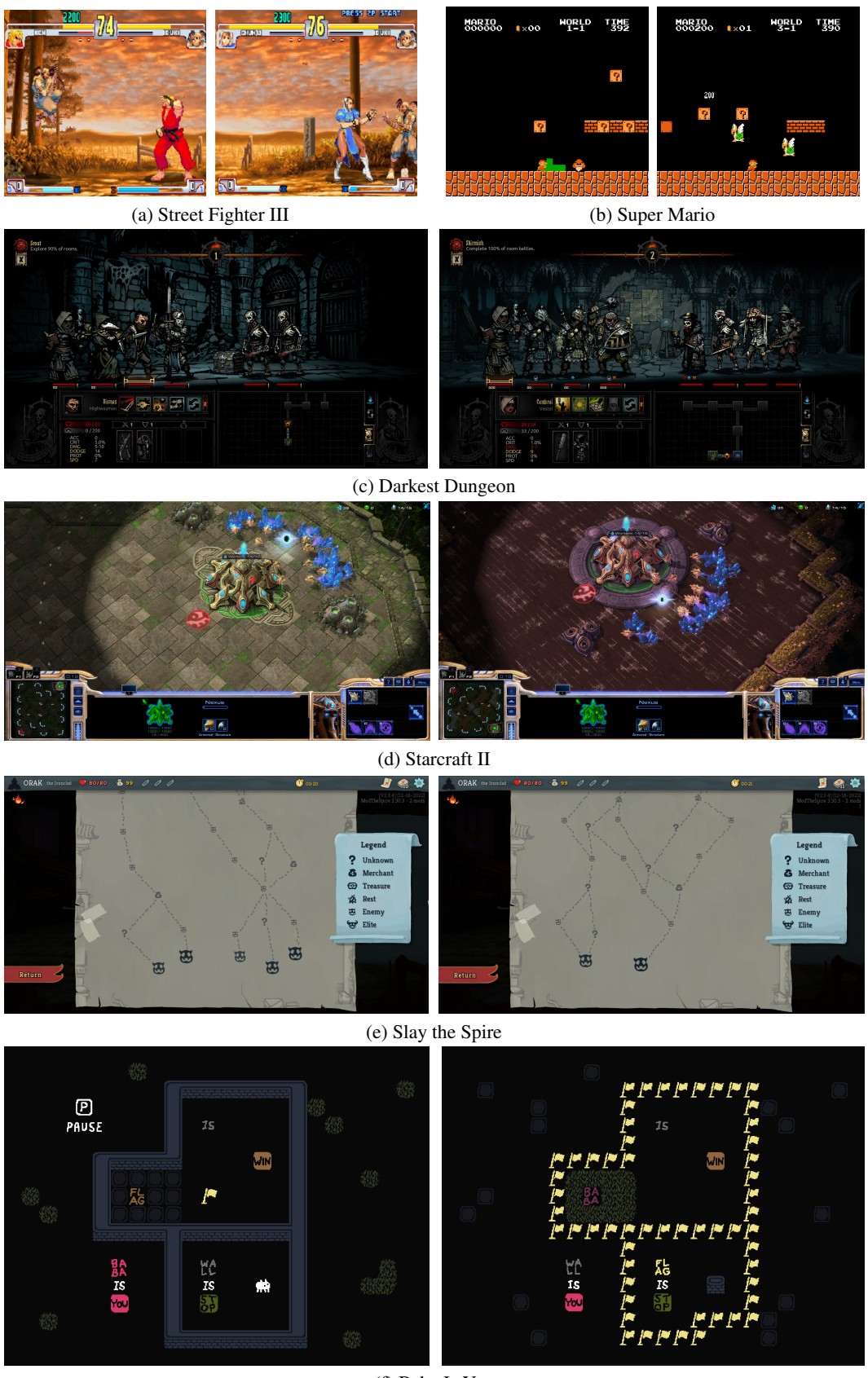

(a) Street Fighter III

(b) Super Mario

(c) Darkest Dungeon

(d) Starcraft II

(e) Slay the Spire

(f) Baba Is You

Figure 28: Comparison of seen (left) and unseen (right) scenarios for six games.

