# OpenReview forum: "Orak: A Foundational Benchmark for Training and Evaluating LLM Agents on Diverse Video Games"
_ICLR.cc/2026/Conference — ICLR 2026 Poster_

### Official Review · Reviewer_5244 · 2025-10-29

**Soundness:** 4
**Presentation:** 3
**Contribution:** 4
**Rating:** 8
**Confidence:** 5

**Summary:**

This paper presents ORAK, a comprehensive benchmark designed to evaluate and train large language model (LLM) agents across a diverse set of 12 video games spanning six major genres. The benchmark addresses critical gaps in existing game-based evaluations by incorporating complex, real-world video games, providing modular agentic strategies (e.g., reflection, planning, and tool use), and releasing a high-quality fine-tuning dataset derived from expert gameplay. The authors also propose a unified evaluation framework that includes leaderboards, LLM battle arenas, and in-depth analyses of input modalities, agentic strategies, and fine-tuning effects. The experiments demonstrate the capabilities and limitations of both proprietary and open-source LLMs across various tasks, showcasing the potential of ORAK as a foundation for advancing general-purpose gaming agents.

**Strengths:**

To be honest, I'm very excited to see an LLM benchmark that integrates complex games. The LLM field is currently flooded with a large number of fixed datasets, yet claiming to be “agentic” is clearly insufficient. ORAK covers a wide range of game genres, including action, adventure, role-playing, simulation, strategy, and puzzle games. This breadth ensures a holistic evaluation of LLM capabilities, from logical reasoning to spatial understanding and long-term planning. The use of the Model Context Protocol (MCP) to integrate LLMs with game environments and agentic modules is a significant contribution. This modular approach enables plug-and-play experimentation and facilitates systematic studies of LLM behaviors in diverse scenarios. The release of expert gameplay trajectories across multiple genres is a valuable resource for the community. The dataset encapsulates meta-knowledge and demonstrates how fine-tuning can enhance LLM performance in both gaming and non-gaming tasks. The paper provides extensive experimental results, comparing proprietary and open-source LLMs across tasks and modalities. The insights into the effects of fine-tuning, visual inputs, and agentic strategies are particularly compelling.

I strongly recommend acceptance of this paper. It makes a significant contribution to the field of LLM evaluation and training, offering a robust benchmark that bridges the gap between academic research and real-world applications. The paper is well-written, methodologically sound, and forward-looking, providing a solid foundation for future work in gaming AI and general-purpose LLMs. I particularly appreciate the authors' attention to detail in designing ORAK and their commitment to open science through the release of datasets and tools.

This work is not only timely but also highly impactful, and I believe it will become a key reference for researchers in the field.

**Weaknesses:**

This paper still has some minor flaws, and I hope the authors will pay attention to the following issues.

1. The current setup pauses games during LLM inference, which simplifies evaluation but does not fully reflect real-world gaming scenarios. Including preliminary results or discussions on latency-aware evaluation protocols would strengthen the paper.

2. Although the paper mentions RL-based fine-tuning as future work, a brief discussion on how ORAK could be adapted for RL experiments (e.g., reward design, dynamic data extraction) would be valuable for readers interested in this direction.

3. The authors acknowledge the cost of proprietary games and LLM APIs. Exploring potential solutions, such as open-source alternatives or simplified game environments, could help lower the barrier to entry for researchers with limited resources.

If the authors could provide detailed data for problems 1 and 2, as well as potential solutions for problem 3, I believe this paper would be worthy of a spotlight paper.

**Questions:**

See weakness.

---

> ### Author Response · Authors · 2025-11-21
>
> We sincerely appreciate the reviewer's insightful comments and positive support of our manuscript.
>
> `W1. The current setup pauses games during LLM inference, which simplifies evaluation but does not fully reflect real-world gaming scenarios. Including preliminary results or discussions on latency-aware evaluation protocols would strengthen the paper.`
>
> Thank you very much for the constructive comment. Among the 12 games in Orak, **only 3 games**, including Street Fighter III, Super Mario, and Starcraft II, **require a pause for evaluation**. For the remaining 9 games, the environment state (almost) does not change before the agent’s action, resulting in identical outcomes even without pausing (Stardew Valley and Minecraft involve minor time changes or occasional monster spawns, but these factors have a negligible impact on the final game scores).
>
> Moreover, following your suggestion, we **additionally conducted a latency-aware evaluation** for StarCraft II.
>
> | Model | Paused (Hard, Original) | Real-time (Hard) | Paused (Easy) | Real-time (Easy) |
> | - | - | - | - | - |
> | gpt-4o-mini (~19.60 sec) | 75.0 | 0 | 100.0 | 0 |
> | gpt-4o (~27.17 sec) | 100.0 | 0 | 100.0 | 100.0 |
> | gemini-2.5-pro (~99.19 sec) | 100.0 | 0 | 100.0 | 0 |
>
> As shown above, when switching from the paused setting to the real-time mode, all models failed (0 score) under the hard level (the original setup), highlighting the strong sensitivity of timing-dependent games to inference latency. When the difficulty level was reduced to easy, all models achieved perfect scores under the paused setting, but only gpt-4o maintained perfect performance under real-time conditions. This indicates that gpt-4o achieves the best balance between latency and gameplay accuracy among the evaluated models. We have **added this analysis to Section 5.7** in the revised manuscript.
>
>
> `W2. Although the paper mentions RL-based fine-tuning as future work, a brief discussion on how ORAK could be adapted for RL experiments (e.g., reward design, dynamic data extraction) would be valuable for readers interested in this direction.`
>
> Thanks for improving our paper. Orak can be readily integrated with existing **multi-turn RL fine-tuning frameworks** for LLM agents [a, b]. Games typically provide sparse rewards, where feedback is often given only at the end (final score). Therefore, careful reward design and data extraction are essential for stable optimization.
>
> **Reward Design.**
> (1) *Reward Discount.*
> Following common practice in traditional RL, we apply **reward discounting** to estimate intermediate rewards during gameplay:
>
> > R_t = \gamma^{T-t} S_{final},
>
> where 0 < \gamma < 1 is the discount factor and S_{\text{final}} is the final game score.
>
> (2) *Turn-based Reward (if available).*
> For games such as **2048** or **Super Mario**, intermediate scores or distances can be measured. In this case, a turn-level reward can be added as:
>
> > R_t = \lambda S_t + (1-\lambda)\gamma^{T-t} S_{\text{final}},
>
> where \lambda is a balancing factor and S_t is the turn-reward.
>
> **Dynamic Data Extraction.**
> By **repeatedly performing** data rollouts, reward estimation, and multi-turn RL fine-tuning over multiple episodes, the policy model can be improved. As the policy improves, it naturally encounters **more challenging game scenarios**, enabling **progressive data extraction**. Furthermore, by comparing validation/test accuracy between the previous and newly fine-tuned models, we can select the better policy at each episode, leading to more robust optimization throughout multiple episodes.
>
> We have **added this discussion to Appendix T** in the revised manuscript.
>
> ---
> [a] RAGEN: Understanding Self-Evolution in LLM Agents via Multi-Turn Reinforcement Learning, ArXiv, 2025
>
> [b] Reinforcing Multi-Turn Reasoning in LLM Agents via Turn-Level Reward Design, ArXiv, 2025
>
>
> `W3. The authors acknowledge the cost of proprietary games and LLM APIs. Exploring potential solutions, such as open-source alternatives or simplified game environments, could help lower the barrier to entry for researchers with limited resources.`
>
> Among the 12 games in Orak, **6 games**, including Street Fighter III, Super Mario, Pokémon, Minecraft, StarCraft II, and 2048, **are free-to-play** while covering diverse and complex scenarios in most genres. We designed Orak to lower the entry cost for researchers by **supporting easy agent evaluation and fine-tuning setups for open-source LLMs**. To further accelerate this effort, we plan to split the leaderboard into an **size-constrained open-source track** and an unlimited track in future releases. Again, we appreciate your thoughtful comment.

---

> > ### Comment · Reviewer_5244 · 2025-11-28
> >
> > Thank you for your reply. Your discussion of the StarCraft 2 delay experiment did indeed address my concerns, and this content is essential for the paper. However, regarding Weakness 2, I believe supplementing the paper with experiments related to the RL algorithm is still important. For a good benchmark, rich experiments and testing across different domains are necessary and crucial. The third question is an open question, and I don't expect the authors to solve it in the short term. I remain positive and supportive of this paper and believe it deserves an ICLR spotlight. However, due to the inability to supplement experimental data for the RL algorithm in the short term, I will maintain my score. To reiterate my personal opinion, this is an excellent benchmark paper that addresses important issues in the field.

---

### Official Review · Reviewer_Tpyv · 2025-10-30

**Soundness:** 2
**Presentation:** 3
**Contribution:** 2
**Rating:** 2
**Confidence:** 4

**Summary:**

This paper proposes **Orak** benchmark and evaluation suite for training and evaluating LLM/VLM agents on a diverse set of real video games. The benchmark includes 12 games spanning action, adventure, strategy, and puzzle genres, a modular agent interface (reflection / planning / action / memory components) built on a Model Context Protocol (MCP), prompt templates and action-space definitions, and an expert-trajectory dataset used for supervised fine-tuning. The submission reports cross-model comparisons, ablations of agentic modules, and fine-tuning experiments including transfer tests to out-of-distribution and non-game tasks.

**Strengths:**

1. **Broad coverage & engineering effort.** The benchmark covers 12 real games across diverse genres and provides a modular evaluation harness, which is useful for benchmarking different LLM agent designs.
2. **Multi-dimensional ability taxonomy.** The paper defines and uses a set of capabilities (e.g., long-horizon planning, spatial reasoning, rule compliance) and maps games to these capability needs, enabling capability–task analyses.
3. **Publicly available supervised trajectories & fine-tuning experiments.** The authors collected and provide a dataset of expert LLM interaction trajectories and demonstrate supervised fine-tuning improvements and some transfer effects.
4. **Useful baseline comparisons.** Results compare multiple closed-source and open-source LLMs under several agentic strategies (zero-shot, reflection, planning, ref-plan), giving a practical snapshot of current model gaps and engineering trade-offs.

**Weaknesses:**

1. **Insufficient novelty argument / differentiation from prior benchmarks.** The paper lists related benchmarks but does not convincingly quantify or empirically demonstrate how Orak meaningfully advances beyond existing game/agent benchmarks. The unique scientific questions Orak enables are not sharply distinguished.
2. **Experimental robustness & statistical reporting are incomplete.** Many reported results lack rigorous statistical detail (consistent number of seeds/trials, confidence intervals, or significance testing). Some reported scores show large variance, which weakens the reliability of the conclusions.
3. **Lack of systematic prompt / hyperparameter sensitivity analyses.** The paper attributes improvements to agentic modules (reflection/planning), but does not systematically vary prompt wording, temperature, max-context length, or other prompt-engineering factors to rule out that effects are largely prompt-driven.
4. **Real-time interaction and latency issues under-addressed.** For latency-sensitive or action-timed games (e.g., fighting or platformers), the evaluation often pauses the game during inference. This departs substantially from real-world online agent constraints; latency-aware experiments are deferred to future work, limiting external validity.
5. GPT-generated trajectories (e.g., gpt4o, o3-mini) **risk bias, hallucinations, low strategy diversity, and contamination**; the authors should document generation details and show that fine-tuned models generalize beyond the generator.

**Questions:**

1. For each major table/figure, how many independent trials and random seeds were used? Please add confidence intervals and describe any hypothesis tests performed. If trials vary by game, report that explicitly.
2. Did you run controlled sweeps over prompt phrasings, temperatures, context lengths, or token limits when comparing agentic modules? If not, please run such sweeps for key games or explain why module effects are independent of prompt variants.
3. Can you provide at least one latency-aware experiment for a timing-sensitive game (e.g., impose an upper bound on LLM response time or simulate delay) and report performance degradation as a function of latency?
4. Fine-tuning data & overfitting controls. For the supervised fine-tuning dataset: how were trajectories sampled (top trajectories or diverse sampling)? What regularization, early-stopping, or validation protocols prevented overfitting to a specific agentic workflow?
5. I request ablation results comparing fine-tuning on different generator sources (e.g., gpt4o-only, o3-mini-only, mixed-source, and, if available, human or RL trajectories) to quantify generator-specific biases.

---

> ### Author Response · Authors · 2025-11-21
> **Official Comment by Authors (1/3)**
>
> `W1. The paper lists related benchmarks but does not convincingly quantify or empirically demonstrate how Orak meaningfully advances beyond existing game/agent benchmarks`
>
> **Table 1 of the main paper quantifies and differentiates** Orak from existing game benchmarks in terms of **modality coverage, genre coverage, game coverage, and supported ablation studies**. Unlike other game benchmarks, Orak offers agent ablation and fine-tuning studies to facilitate the development of real-world LLM agents for dynamic environments. This support and empirical studies demonstrate how Orak advances beyond existing game benchmarks.
>
> `W2&Q1. Experimental robustness & statistical reporting are incomplete. For each major table/figure, how many independent trials and random seeds were used? Please add confidence intervals and describe any hypothesis tests performed. If trials vary by game, report that explicitly.`
>
> Thanks for the careful comment. **Most major tables (Table 3-7) report standard deviations** of the results. The number of trials varies by game (3 to 20 as described in Line 360) because of the running costs, and **each trials are explicitly reported in Appendices C–N**. To further address your concern, we **additionally report 1) confidence intervals** for gpt-4o, **2) each game's t-test results**, and **3) overall games' t-test results** between gpt-4o and two different size models (Llama-3.2-1B and Qwen 2.5-72B).
>
> **1) Confidence Interval**
>
> | Model | Metric | SF3 | Super Mario | Ace Attorney | Her Story | Pokémon | Darkest Dungeon | Minecraft | Stardew Valley | StarCraft II | Slay the Spire | Baba Is You | 2048 |
> |-|-|-|-|-|-|-|-|-|-|-|-|-|-|
> | - | # seed | 5 | 20 | 3 | 3 | 3 | 3 | 3 | 3 | 4 | 3 | 3 | 5 |
> | gpt-4o | mean | 29.7 | 34.1 | 85.3 | 64.2 | 38.9 | 93.4 | 71.0 | 81.4 | 100.0 | 23.6 | 20.0 | 5.6 |
> | | std | 14.3 | 14.2 | 1.5 | 5.2 | 9.6 | 1.5 | 7.0 | 4.8 | 0.0 | 22.1 | 0.0 | 1.5 |
> | | CI (95%) | 17.76 | 6.65 | 3.73 | 12.92 | 23.85 | 3.73 | 17.39 | 11.92 | 0.00 | 54.90 | 0.00 | 1.86 |
>
> **2) Each game's t-test results** (used Welch's t-test [a])
>
> gpt-4o vs Llama-3.2-3B; Two models' performances are statistically significantly different or deterministically different in 10 out of 12 games.
> | Game | t-stat | p-value | Interpretation |
> |-|-|-|-|
> | SF3 | 4.644 | 0.0097 | **significant** |
> | Super Mario | 4.149 | 0.0002 | **significant** |
> | Ace Attorney | 54.641 | <0.0001 | **significant** |
> | Her Story | 20.155 | 0.0015 | **significant** |
> | Pokémon | 7.018 | 0.0197 | **significant** |
> | Darkest Dungeon | 107.849 | 0.0001 | **significant** |
> | Minecraft | 17.568 | 0.0032 | **significant** |
> | Stardew Valley | 29.373 | 0.0012 | **significant** |
> | StarCraft II | – | – | **deterministic** |
> | Slay the Spire | 1.850 | 0.2056 | not significant |
> | Baba Is You | 2.003 | 0.1831 | not significant |
> | 2048 | 8.329 | 0.0011 | **significant** |
>
> gpt-4o vs Qwen-2.5-72B; Two models' performances are statistically significantly different or deterministically different in 6 out of 12 games.
> | Game | t-stat | p-value | Interpretation |
> |-|-|-|-|
> | SF3 | 0.435 | 0.681 | not significant |
> | Super Mario | 1.128 | 0.268 | not significant |
> | Ace Attorney | 14.127 | 0.0039 | **significant** |
> | Her Story | 7.345 | 0.0104 | **significant** |
> | Pokémon | 1.416 | 0.230 | not significant |
> | Darkest Dungeon | 2.406 | 0.126 | not significant |
> | Minecraft | 17.568 | 0.0032 | **significant** |
> | Stardew Valley | 6.320 | 0.0164 | **significant** |
> | StarCraft II | – | – | **deterministic** |
> | Slay the Spire | 1.113 | 0.371 | not significant |
> | Baba Is You | – | – | deterministic tie |
> | 2048 | 8.032 | 0.0013 | **significant** |
>
> **3) Overall games' t-test results** (used paired t-test [b])
>
> When conducting a paired t-test across all games, gpt-4o shows statistically significant superior performance compared to both models.
>
> | Comparison | t-stat | p-value | Interpretation |
> |-|-|-|-|
> | GPT-4o vs Llama-3.2-1B | 5.227 | 0.00028 | **significant** |
> | GPT-4o vs Qwen-2.5-72B | 3.122 | 0.00971 | **significant** |
>
> Therefore, we believe our experimental results are reliable, although we used a relatively small number of seeds due to the high inference cost of LLMs. We have **added this statistical analysis to Appendix S** in the revised manuscript.
>
> ---
> [a] The Generalization of Student's Problem when Several Different Population Variances are Involved, Welch, 1947
>
> [b] Statistical Methods for Research Workers, Fisher, 1925

---

> ### Author Response · Authors · 2025-11-21
> **Official Comment by Authors (2/3)**
>
> `W3&Q2. Lack of systematic prompt / hyperparameter sensitivity analyses`
>
> Per your suggestion, we conducted two **additional ablation studies for varying prompts and temperatures** on 6 games in Orak (1 game per genre). For a new prompt, we used an augmented system prompt used in our proposed fine-tuning set.
>
> **Prompt Study**
> | Model | Prompt | Supermario | HerStory | Pokemon | Minecraft | Slay The Spire | 2048 |
> | - | - | - | - | - | - | - | - |
> | Llama-3.2-3B | Original | 31.8±10.1 | 4.2±1.1 | 0.0±0.0 | 0.0±0.0 | 0.0±0.0 | 0.3±0.2 |
> |  | New | 27.2±8.5 | 8.0±0.2 | 0.0±0.0 | 0.0±0.0 | 0.0±0.0 | 0.4±0.2 |
> | gpt-4o | Original | 34.1±14.2 | 64.2±5.2 | 38.9±9.6 | 71.0±0.0 | 23.6±22.1 | 5.6±1.5 |
> |  | New | 33.3±11.8 | 55.8±5.6 | 36.1±4.8 | 71.0±0.0 | 15.0±0.0 | 4.6±2.2 |
>
> The table above shows the performance of Llama-3.2-3B and gpt-4o for varying prompts. Based on this, we further conducted the paired t-test between the original and new prompts for both models.
>
> | Model | t-stat | p-value | Interpretation |
> |-|-|-|-|
> | Llama-3.2-3B | 0.107 | 0.919 | **not significant** |
> | GPT-4o | 2.258 | 0.0735 | **not significant** (p≈0.07) |
>
> The results show that both models' performance did **not change statistically significantly under the prompt variation**.
>
> **Temperature Study**
> | Model | Temperature | Supermario | HerStory | Pokemon | Minecraft | Slay The Spire | 2048 |
> | - | - | - | - | - | - | - | - |
> | Llama-3.2-3B | 0 | 31.6±10.0 | 5.4±1.2 | 0.0±0.0 | 0.0±0.0 | 0.0±0.0 | 0.8±1.0 |
> |  | 0.3 | 30.9±8.6 | 4.2±1.1 | 0.0±0.0 | 0.0±0.0 | 0.0±0.0 | 2.2±1.0 |
> |  | 1 | 31.8±10.1 | 5.8±2.2 | 0.0±0.0 | 0.0±0.0 | 0.0±0.0 | 1.9±1.6 |
> | gpt-4o | 0 | 30.6±13.5 | 62.4±6.3 | 25.0±8.3 | 71.0±0.0 | 23.6±22.1 | 5.4±1.5 |
> |  | 0.3 | 35.5±18.3 | 64.2±5.2 | 33.3±0.0 | 71.0±0.0 | 15.0±0.0 | 5.2±2.1 |
> |  | 1 | 31.8±14.2 | 61.3±5.9 | 33.3±0.0 | 71.0±0.0 | 12.3±4.6 | 5.7±4.8 |
>
> The table above show the performance of Llama-3.2-3B and gpt-4o for varying temperatures. Based on this, we conducted the paired t-test between temperature 0 and 1 for both models.
>
> | Model | t-stat | p-value | Interpretation |
> |--------|--------:|--------:|----------------|
> | Llama-3.2-3B | -1.611 | 0.168 | **not significant** |
> | GPT-4o | 0.169 | 0.873 | **not significant** |
>
> The results show that both models' performance did **not change statistically significantly under the temperature variation**.
>
> Overall, our experimental results are **not sensitive to prompt and temperature variations**. We did not conduct a separate study on max-context length, since game actions typically appear at the end of LLM responses and are parsed afterward, so performance cannot be reliably analyzed below a certain maximum context length while remaining relatively consistent beyond some threshold.
>
>
> `W4&Q3. Real-time interaction and latency issues under-addressed. Can you provide at least one latency-aware experiment for a timing-sensitive game?`
>
> Thank you very much for the constructive comment. Following your suggestion, we **conducted an additional study for real-time gameplay** using StarCraft II. The average response latency of each model was measured as follows: gpt-4o-mini ≈ 19.6 s, gpt-4o ≈ 27.2 s, and gemini-2.5-pro ≈ 99.2 s per step.
>
> | Model | Paused (Hard, Original) | Real-time (Hard) | Paused (Easy) | Real-time (Easy) |
> | - | - | - | - | - |
> | gpt-4o-mini (~19.60 sec) | 75.0 | 0 | 100.0 | 0 |
> | gpt-4o (~27.17 sec) | 100.0 | 0 | 100.0 | 100.0 |
> | gemini-2.5-pro (~99.19 sec) | 100.0 | 0 | 100.0 | 0 |
>
> As shown above, when switching from the paused setting to the real-time mode, all models failed (0 score) under the hard level (the original setup), highlighting the strong sensitivity of timing-dependent games to inference latency. When the difficulty level was reduced to easy, all models achieved perfect scores under the paused setting, but only gpt-4o maintained perfect performance under real-time conditions. This indicates that gpt-4o achieves the best balance between latency and gameplay accuracy among the evaluated models. We have added this analysis to Section 5.7 in the revised manuscript.

---

> ### Author Response · Authors · 2025-11-21
> **Official Comment by Authors (3/3)**
>
> `W5&Q4. The authors should document generation details and show that fine-tuned models generalize beyond the generator. How were trajectories sampled (top trajectories or diverse sampling)? What regularization, early-stopping, or validation protocols prevented overfitting to a specific agentic workflow?`
>
> **Section 4 elaborates the generation details of our fine-tuning dataset**. As demonstrated, we selected top-performing trajectories with the reflection-planning-action agent for all 12 games. **Appendix P.2 includes the fine-tuning configuration details**. To prevent overfitting, we fine-tuned the models for 1 epoch (~1600 steps) with 100 warm-up steps with a small initial learning rate of 1e-6 and the cosine scheduler, which leads to generalization in ood- and non-game scenarios (see Table 7). Meanwhile, due to the limited capacity of the student models, the fine-tuned models generally perform worse than gpt-4o on ood-games. We do not agree that we should show fine-tuned models generalize beyond the generator, as this is beyond the scope of our study.
>
> `W5&Q5. GPT-generated trajectories (e.g., gpt4o, o3-mini) risk bias, hallucinations, low strategy diversity, and contamination; I request ablation results comparing fine-tuning on different generator sources (e.g., gpt4o-only, o3-mini-only, mixed-source, and, if available, human or RL trajectories) to quantify generator-specific biases.`
>
> Thanks for pointing out this issue. Following your suggestions, we conducted **additional ablation studies that compare the effect of different generator resources in fine-tuning**.
>
> | Model | SF3 | DarkestD | StarCraft2 | SlaySpire | BabaIsYou |
> | - | - | - | - | - | - |
> | Llama-3.2-3B (pre-trained) | 12.0 | 87.2 | 0.0 | 0.0 | **20.0** |
> | gpt/o3-mini-only (110k) | **40.0** | **92.0** | 0.0 | 10.7 | **20.0** |
> | deepseek-r1-only (110k) | 14.0 | 52.8 | 0.0 | 8.0 | **20.0** |
> | mixed (220k) | 18.0 | 73.3 | 25.0 | **12.7** | **20.0** |
>
> As shown in the table above, Llama-3.2-3B fine-tuned on datasets from OpenAI generators (gpt-4o and o3-mini) always outperform that fine-tuned on the dataset from deepseek-r1 [c], which may be because gpt-4o and o3-mini are generally better than deepseek-r1 in Orak. Llama-3.2-3B fine-tuned on the mixed dataset shows the middle performance, except Slay The Spire where the mixed dataset performs the best. We did not observe any significant bias, low-diversity, or contamination of the current SFT dataset, as the fine-tuned model shows generalizability on ood- and non-game scenarios (See Table 7 of the main paper). However, we leave this study as future work.
>
> ---
> [c] DeepSeek-R1: Incentivizing Reasoning Capability in LLMs via Reinforcement Learning, ArXiv, 2025

---

> > ### Comment · Reviewer_Tpyv · 2025-11-28
> >
> > Thank you for the detailed rebuttal. The additional experiments and clarifications significantly strengthen the empirical foundation and address many of the concerns raised in my original review.
> >
> > **Statistical Reliability:**
> >
> > The inclusion of confidence intervals, per-game Welch t-tests, and a paired t-test across the full game suite meaningfully improves the statistical rigor of the results. Reporting the varying number of seeds across different games and providing transparent variance estimates give me more confidence in the robustness of the conclusions.
> >
> > **Prompt/Temperature Robustness:**
> >
> > The prompt-paraphrasing and temperature ablations show that the performance differences attributed to agentic modules are relatively stable across prompt variants and decoding settings. While not exhaustive, these studies are sufficient to alleviate my earlier concern that the observed module benefits might be prompt-driven artifacts.
> >
> > **Latency and Real-Time Evaluation:**
> >
> > The new real-time experiment in StarCraft II is a valuable addition. It demonstrates concretely how inference latency affects gameplay performance and highlights the gap between “paused” and real-time environments. This addresses the external-validity question in a direct and transparent manner.
> >
> > **Fine-Tuning Data and Generator Bias:**
> >
> > The ablation comparing high- vs low-score trajectories and different generator sources suggests that the fine-tuning behavior is driven primarily by data quality rather than overfitting to a particular generator. The authors’ clarification of training protocols and early-stopping practices also helps address overfitting concerns.
> >
> > **Contribution Perspective:**
> >
> > While I still believe that the submission is more of a **benchmark + workflow + infrastructure** contribution than a methodological one, the breadth of environments, the capability taxonomy, the modular MCP-based interface, and the released fine-tuning dataset together form a comprehensive and useful resource. Following the rebuttal, I better appreciate how this benchmark can enable systematic investigations that were difficult to conduct with prior isolated environments.
> >
> > Given the strengthened empirical analysis and the significant engineering value of the benchmark to the growing LLM-agent community, I now view the paper as meeting the threshold for acceptance.
> >
> > I am raising my score to 6 (borderline accept).

---

### Official Review · Reviewer_Cdp1 · 2025-11-01

**Soundness:** 2
**Presentation:** 2
**Contribution:** 2
**Rating:** 4
**Confidence:** 4

**Summary:**

This paper introduces Orak, a benchmark for evaluating LLM agents across 12 video games spanning six genres. The authors aim to address limitations in existing game benchmarks by offering greater diversity, enabling studies on agentic modules (such as reflection and planning), and providing resources for adapting LLMs into gaming agents. The key contributions are the benchmark itself, which uses a plug-and-play interface based on the Model Context Protocol (MCP) for standardized evaluation, and a fine-tuning dataset derived from expert LLM trajectories designed to distill gaming skills into smaller models. The paper presents a series of experiments on 15 LLMs, analyzing their performance, the impact of agentic modules, the effect of visual inputs, and the generalization capabilities of fine-tuned models.

**Strengths:**

The primary strength of this paper is the scale and diversity of the benchmark. Compiling 12 games across six distinct genres, each with its own environment setup and state representation, is a big **engineering effort.** This provides a broad testbed for evaluating a variety of agent capabilities, from reaction time in action games to long-term planning in strategy games.

The introduction of a unified, plug-and-play interface using MCP is a commendable step towards standardized and reproducible evaluation of LLM agents in gaming environments.

The release of a fine-tuning dataset, while based on LLM-generated trajectories.

**Weaknesses:**

The paper, despite its significant engineering effort, suffers from several weaknesses in its core claims, methodology, and the novelty of its conclusions, which limit its overall contribution.

Largely Unsurprising and Incremental Conclusions: The main findings drawn from the extensive experiments largely confirm well-established knowledge in the LLM agent community, offering little new insight.

- The conclusion that proprietary, closed-source models outperform their open-source counterparts is widely accepted and requires little further validation in 2025.

- The finding that agentic workflows (e.g., reflection, planning) benefit capable models is not new.

The claim that visual inputs often hinder performance is misleading. This outcome is likely an artifact of the experimental design, where highly structured and pre-processed text provides a cleaner, more direct signal than raw visual data for current VLMs. A more accurate conclusion would be that under this specific setup, the models fail to extract sufficient value from visual inputs to overcome the noise, rather than a general indictment of visual modalities for gaming agents.


Questionable Design Choices in Benchmark and Data Generation:

- The selection of games appears to be driven more by the availability of existing APIs or emulators (e.g., Mineflayer for Minecraft, PyBoy for Pokémon Red) rather than a principled selection of titles that would best probe the frontiers of AI capabilities. The benchmark lacks modern, complex 3D games that pose severe challenges in perception from raw pixels, physics-based interaction, and complex spatial reasoning.

- The fine-tuning dataset is generated by an 'expert' LLM (GPT-4o), which fundamentally caps the potential performance of any fine-tuned model at the level of the teacher model. This methodology prevents the discovery of novel strategies that might surpass the teacher's capabilities and introduces the teacher's inherent biases and failure modes into the student models.

- By selecting only the highest-scoring trajectories for the fine-tuning dataset, the authors introduce a strong survivorship bias. The models learn from 'perfect' or near-perfect executions ('sunny day' scenarios) but are not exposed to data on how to recover from mistakes, adapt to unexpected situations, or turn a losing game around. This is a critical omission for developing robust agents that can handle the stochasticity and adversity inherent in complex games.

**Questions:**

The paper positions Orak as a foundational benchmark that pushes the boundaries of agent evaluation. However, the tasks often seem simplified through pre-processed states and high-level APIs, which might not establish a clear differentiation from prior work.

Could the authors elaborate on the unique challenges Orak presents compared to existing agent benchmarks? The current results do not seem to establish a clear differentiation, as the main conclusions are largely echoes of findings from other domains.
What specific agent capabilities are uniquely tested in Orak that are not adequately covered by prior benchmarks? A more compelling case could be made by showcasing a task where top-performing agents from other domains systematically fail due to a game-specific challenge that Orak is specifically designed to evaluate. For instance, is there a scenario that rigorously tests an agent's ability to reason under partial observability from raw signals, a core challenge in many games?

I'm willing to increase my score if the author can answer my question.

---

> ### Author Response · Authors · 2025-11-21
> **Official Comment by Authors (1/3)**
>
> We deeply appreciate the reviewer's constructive comments and helpful feedback on our manuscript.
>
> `W1. The main findings drawn from the extensive experiments largely confirm well-established knowledge in the LLM agent community, offering little new insight. 1) proprietary models outperform their open-source counterparts. 2) The finding that agentic workflows (e.g., reflection, planning) benefit capable models.`
>
> Numerous LLM benchmarks have emerged across various knowledge domains, and most report that proprietary models outperform open-source ones. This is a natural outcome, as proprietary models generally possess powerful cognitive abilities across diverse domains. We believe **sharing this trend supports the reliability** of Orak. Also, despite the similar tendencies, each LLM benchmark has value in providing insights into how far LLMs have progressed and how much further they can improve within its respective domain. Orak contributes this value in the video game domain with long-horizon interactive tasks.
>
> Moreover, Orak introduces two **domain-specific new findings**: (1) in long-horizon interactive tasks such as games, agent design should be carefully tailored to model capacity, and (2) training with teacher gameplay trajectories can improve performance of LLM agents in in-, ood-, and non-game scenarios. Therefore, we believe Orak has novelty and contributions as an LLM agent benchmark.
>
> `W2. The claim that visual inputs often hinder performance is misleading. A more accurate conclusion would be, the models fail to extract sufficient value from visual inputs to overcome the noise, rather than a general indictment of visual modalities for gaming agents`
>
> Thank you very much for your careful comment. Per your suggestion, we **rephrased** the expression as *"models fail to extract sufficient value from visual inputs"*. Please see Lines 86-87 in the revised manuscript.
>
> `W3. The selection of games appears to be driven more by the availability of existing APIs or emulators (e.g., Mineflayer for Minecraft, PyBoy for Pokémon Red) rather than a principled selection of titles that would best probe the frontiers of AI capabilities. The benchmark lacks modern, complex 3D games that pose severe challenges in perception from raw pixels, physics-based interaction, and complex spatial reasoning`
>
> In designing our benchmark, we selected games by carefully considering **genre representativeness** and **benchmark reliability**, not only considering the API availability. Specifically, we consider the three important criteria:
> **(1) Popularity**. Popular games typically incorporate elements widely recognized within their genres and frequently influence subsequent titles featuring similar gameplay mechanics, inherently reflecting genre representativeness. For example, Street Fighter inspired numerous similar fighting-action games, while Super Mario led to countless side-scrolling action titles. Thus, we included only games with sales or play counts exceeding 100K (see the table below).
> **(2) Task Complexity/Diversity**. Adequate task complexity and diversity are essential for representing a wide range of scenarios within a genre. Therefore, we selected games with varied stages and tasks to ensure intra-genre scenario representativeness. Specifically, as indicated in the table below, each genre (sum of two games) contains a diverse number of sub-stages/tasks (5–482).
> **(3) Transparency**. For reliable benchmarking, we prioritized games with available API wrappers or modding tools to *clearly* extract and convey game states/actions to LLMs via a text interface.
>
> |Games|StreetFighterIII|SuperMario|AceAttorney|HerStory|PokémonRed|DarkestDungeon|Minecraft|StardewValley|StarCraftII|SlayTheSpire|BabaIsYou|2048|
> |-|-|-|-|-|-|-|-|-|-|-|-|-|
> |Sales/Plays|3.6M+[a]|40M+[b]|13M+[c]|100K+[d]|31M+[e]|6M+[f]|300M+[g]|41M+[h]|6M+[i]|1.5M+[j]|1.2M+[k]|23M+[l]|
> |# of Tasks/Stages|19|32|4|1|338|15|379|37|3|200|481|1|
> |Task/Stage Description|enemy types|maps|judgement types||enemy types|maps|crafting items|crops|enemy types|stages x characters|maps||
>
> **Game Selection**. With the above criteria, we selected 12 games that satisfy all three criteria from an initial list of 49 video games. The excluded games include Galaga (Action genre) and Red Dead Redemption II (Adventure genre). Specifically, Galaga lacked an API or modding tools enabling the extraction of game states into text, and Red Dead Redemption II was excluded due to significant noise when converting mouse-based player control into LLM-compatible actions, both raising concerns regarding transparancy.

---

> ### Author Response · Authors · 2025-11-21
> **Official Comment by Authors (2/3)**
>
> **Game Complexity**. Among the games in Orak, 4 games, including Darkest Dungeon, Stardew Valley, Baba Is You, and Slay the Spire, were **released after 2016**, which confirms its **modernity**. Also, 5 games, including Minecraft, StarCraft II, Darkest Dungeon, Stardew Valley, and Slay the Spire, require **strong raw-pixel perception** ability when played with vision-only inputs. 2 games, including Minecraft and StarCraft II, are **3D games** and thus require **complex spatial reasoning**. While Super Mario involves weak physics-based interactions such as jump timing within a gravity-simulated environment, Orak currently lacks games with more complex physics-based interactions. We leave the inclusion of such environments as future work.
>
> ---
> [a] https://vgsales.fandom.com/wiki/Street_Fighter
>
> [b] https://www.vgchartz.com/game/226187/super-mario/
>
> [c] https://www.vgchartz.com/gamedb/?name=ace+attorney
>
> [d] https://en.wikipedia.org/wiki/Her_Story_(video_game)
>
> [e] https://en.wikipedia.org/wiki/Pok%C3%A9mon_Red,_Blue,_and_Yellow
>
> [f] https://gameworldobserver.com/2022/12/26/darkest-dungeon-sales-6-million-copies-red-hook
>
> [g] https://www.businessofapps.com/data/minecraft-statistics/
>
> [h] https://en.wikipedia.org/wiki/Stardew_Valley
>
> [i] https://en.wikipedia.org/wiki/StarCraft_II:_Wings_of_Liberty
>
> [j] https://en.wikipedia.org/wiki/Slay_the_Spire
>
> [k] https://steamspy.com/app/736260
>
> [l] https://medium.com/%40gabrielecirulli/2048-success-and-me-7dc664f7a9bd
>
> `W4. The fine-tuning dataset is generated by an 'expert' LLM (GPT-4o), which fundamentally caps the potential performance of any fine-tuned model at the level of the teacher model. This methodology prevents the discovery of novel strategies that might surpass the teacher's capabilities and introduces the teacher's inherent biases and failure modes into the student models`
>
> Still, for many tasks including math, supervised fine-tuning (SFT) datasets generated by teacher models remain a strong resource to improve student models [m]. Moreover, as reported in modern studies like DeepSeek-R1 [n], the pre-execution of SFT fine-tuning has been a key factor in triggering performance improvements of RL fine-tuning. Therefore, the SFT dataset does **not prevent the discovery** of novel strategies, but can **facilitate such discoveries**. For example, after fine-tuning models with our proposed dataset, one can perform advanced RL fine-tuning using Orak’s game scores as rewards, thereby surpassing the teacher performance. We further **added the discussion of how to enable RL fine-tuning LLM/VLM agents using Orak** in Appendix T of the revised manuscript.
>
> ---
> [m] OpenThoughts: Data Recipes for Reasoning Models, ArXiv, 2025
>
> [n] DeepSeek-R1: Incentivizing Reasoning Capability in LLMs via Reinforcement Learning, ArXiv, 2025
>
> `W5. By selecting only the highest-scoring trajectories for the fine-tuning dataset, the authors introduce a strong survivorship bias. The models learn from 'perfect' or near-perfect executions, but are not exposed to data on how to recover from mistakes, adapt to unexpected situations, or turn a losing game around`
>
> **High-scoring trajectories are not necessarily perfect** or near-perfect executions, because powerful LLM agents often fall into in-game traps and must overcome them to achieve high scores. Thus, high-scoring trajectories **inherently contain helpful reasoning patterns to recovering from mistakes or unexpected situations** through appropriate reflection and planning.
>
> Moreover, to address your concern, we **conducted additional experiments** that fine-tune Llama-3.2-3B on our high-scoring trajectories **mixed with** low-scoring trajectories, to expose more unexpected situations or failure game states to the model.
>
> | Data | SF3 | DarkestD | StarCraft2 | SlaySpire | BabaIsYou |
> | - | - | - | - | - | - |
> | No fine-tuning | 12.0 | 87.2 | **0.0** | 0.0 | **20.0** |
> | high-score (110K) | **40.0** | **92.0** | **0.0** | **10.7** | **20.0** |
> | mixed (220K) | 18.0 | 40.0 | **0.0** | **10.7** | **20.0** |
>
> As shown in the table, the high-scoring trajectories proved to be more effective, while the mixed dataset contributed little or even degraded performance in Street Fighter III and Darkest Dungeon. This confirms that our dataset indeed contains high-quality reasoning for handling diverse and challenging situations.

---

> ### Author Response · Authors · 2025-11-21
> **Official Comment by Authors (3/3)**
>
> `Q1. The paper positions Orak as a foundational benchmark that pushes the boundaries of agent evaluation. However, the tasks often seem simplified through pre-processed states and high-level APIs, which might not establish a clear differentiation from prior work.`
>
> As shown in Table 3 of the main paper, **most LLM agents still struggle** to solve long-horizon interactive tasks, even with high-level APIs, and their performance remains **below that of human novices**. Therefore, there is still room to push the boundaries of agent evaluation, particularly for developing resource-efficient agents in interactive environments.
>
>
> `Q2. Could the authors elaborate on the unique challenges Orak presents compared to existing agent benchmarks? What specific agent capabilities are uniquely tested in Orak that are not adequately covered by prior benchmarks? `
>
> Thank you very much for your constructive comment.
>
> **Unique Challenges.** The game rules and contents (e.g., monster and item properties) are relatively unfamiliar to LLMs. The LLM agents should perform goal-oriented tasks by adhering to game-specific rules that are absent from common-sense knowledge in long-horizon interactive manner. Consequently, **navigating unfamiliar environments**, **following environment-specific rules**, **recognizing one’s own mistakes and progress**, and **achieving goals through continuous interaction** constitute the unique challenges of Orak. These challenges may explain why, unlike benchmarks such as Math or Web where train–test data distributions are similar, even proprietary LLM agents have not yet reached the performance level of human novices in Orak (See Table 3 in the main paper).
>
> Moreover, these game settings closely mirror many practical use cases of LLM agents; designing agents for newly released programs or internal systems governed by special rules. In such scenarios, practitioners can refer to the agent performance and insights from Orak for designing environment-specific agents.
>
> **Agent Capabilities.** As shown in Figure 3 (radar chart) of the main paper, the novelty of Orak lies *not* in evaluating unique agent capabilities, but in assessing **how comprehensively** an LLM agent possesses a diverse set of capabilities in playing games. To support this claim, we conducted a validation experiment demonstrating the need for a benchmark that evaluates capability comprehensiveness, rather than focusing on individual capabilities.
>
> *Setup*. We constructed a set of multiple-choice questions (MCQs) to individually test the 7 agent capabilities in Figure 3 of the main paper. For each capability, we selected 5 relevant scenes from games with the maximum rating (3 on the radar chart), and generated examples as follows:
>
> ```
> **MCQ example for Rule Following (RF) capability**
>
> {SuperMario Rules}
> {Current Game State}
> Which of the following is true?
> (A) The three visible bricks cannot be broken.
> (B) Mario becomes stronger after breaking bricks.
> (C) Mario has already jumped over the Goomba.
> (D) Mario has already hit the leftmost question block.
> (E) Mario dies if he climbs onto the warp pipe.
> ```
>
> *Results*. The table below shows the MCQ accuracy (%) for two LLMs. Overall, GPT-4o performed better than Qwen-2.5-7B, but Qwen-2.5-7B did not significantly underperform (i.e., getting a score of 0) in any particular capability. However, looking at Table 3 of the main paper, when requiring comprehensive capabilities all in once, the performance gap was getting bigger and Qwen-2.5-7B shows near 0 performance in many games. This shows that **assessing the LLM capability comprehensiveness clearly differs from assessing a sole capability**, making Orak distinctly different from other benchmarks.
>
> |Models|RF|LR|SR|LTU|LP|EH|OH|Avg|
> |-|-|-|-|-|-|-|-|-|
> |Qwen2.5-7B|40|40|40|60|60|80|40|51.4|
> |GPT-4o|40|80|60|80|60|60|80|65.7|

---

### Official Review · Reviewer_mcXu · 2025-11-05

**Soundness:** 3
**Presentation:** 3
**Contribution:** 2
**Rating:** 8
**Confidence:** 5

**Summary:**

The paper proposes Orak, a benchmark and dataset for evaluating foundation models in dynamic digital games scenarios. Orak increases the diversity of game genres covered in evaluation when compared to previous benchmarks, and the developed platform also allows plug-in and enabling/disable different agentic modules for ease of evaluation/ablation.

Moreover, Orak includes a dataset of fine-tuning data, collected as interaction trajectories generated by foundation models guiding playback of all 12 games supported in the platform.

Experimental results show the performance of 15 foundation models on the benchmark, an ablation study of 2 of those models utilizing different agentic approaches to play all games, results on scenarios combining different data modalities, and SFT results using Llama to illustrate the benefits of the collected trajectory dataset.

**Strengths:**

Orak provides a very funcional benchmark platform for different game genres, as well as a somewhat general covered of game genres that allows better insights about required capabilities from foundation models during gameplay. Such results can also potentially generalize to wider impact beyong digital games alone.

The presented results well illustrate the current capabilities of foundation models and how different games probe them in different dimensions.

The created dataset (and, more importantly, data collection platform) can also serve as a stepping stone for further research on improving gameplay models/agents/systems.

**Weaknesses:**

While well presented and illustrating the potential of the benchmarl in principle, the current paper presents some limitations. Mostly regarding analysis of the results and the rationale for some of its design aspects.

The manuscript claims "in-depth analyses of input modality, agentic strategies, and fine-tuning effects", but falls a bit short of this challenging goal. It does provide some interesting insights, but doesn't really go deep into either of the 3 areas. Having said that, the platform itself is already of great value and can be used as a strong base for further evaluation and analysis. Toning down such claims still leaves the rest of Orak as solid work.

Regarding input modalities, Orak emphasizes pre-extraction of game state information and textual representation of such data. This both; i) greatly reduces the challenges of observation/state/world understanding and already bias results to text; and ii) muddles the analysis of providing different modalities later which then also needs to deal with potential conflicts in different modalities and in some models seemingly having preference for specific modalities. Also known issues previously discussed in the literature.

The presented ablation for agentic modules is interesting as a high-level overview, but its results as presented are not well discussed and don't show significant insights. The analysis here could go much more in-depth in future work.

The LLM finetuning experiment and resutls are not in-depth and only take a quick look at 1 foundation model in 2 small sizes, where the benefits of SFT with any data would already provide the most benefit. The paper would benefit from a more detailed analysis of dataset quality and what/how it actually contributes during training, as well as any possible insights into data scalling.

However, the main limitation of Orak as a platform is the lack of functionality to properly evaluate grounding of actions, as each game action space has been manualy defined differently and already mapped to high-level functions that heavily abstract and simplify the problem. Though, the platform could be easily modified to offer a full pixel-to-keyboard/mouse-commands interface that fully exposes complexity and allow uses to benchmark at different levels.

**Questions:**

I don't think the games industry itself is the main beneficiary of such benchmarking effort. How does this motivation and focus affect the benchmark design? Agent autonomy using games as learning environment could have much wider impacts and would need to be analysed differently.

The paper mentions previous evaluation "often rely on visual inputs" as a criticism. Why? I'd argue that understanding of visual observations is exactly the most important area where games can help as benchmark for foudnaiton model capabilties.

In the Arena setting results, why exactly was Starcraft evaluated with one less model than Street Fighter?

Also, critically, do you have any insights on why Minitron-8B performs so much better than in full results? Minitron was 0 for Starcraft in Table 3. Any deeper understanding here could be significant.

BTW, Table 3 shows the results only of using the "auto extracted state in text only form", correct? It would be interesting to have an easy comparison of that vs. best resutls somewhere. Even if in an appendix.

In a similar veing to the Arena question for Minitron, do you have any insights on why there is no SFT benefit intra-game for Startcrat and Baba?

Typo:
"In game industry" -> "In the games industry"

---

> ### Author Response · Authors · 2025-11-21
> **Official Comment by Authors (1/3)**
>
> We sincerely appreciate the reviewer's constructive comments and positive feedback on our manuscript.
>
> `W1. The manuscript claims "in-depth analyses of input modality, agentic strategies, and fine-tuning effects", but falls a bit short of this challenging goal. Toning down such claims still leaves the rest of Orak as solid work`
>
> Thank you very much for your thoughtful comment. Per your suggestion, we have toned down the claim to *"Our benchmark offers comprehensive evaluation dimensions, including game score leaderboards, competitive LLM battle arenas, and **ablation studies** of input modality, agentic strategies, and fine-tuning effects"*. Please see lines 23 and 82 in the revised manuscript.
>
>
> `W2. Regarding input modalities, Orak emphasizes pre-extraction of game state. This both i) greatly reduces the challenges of observation/state/world understanding and already bias results to text; and ii) muddles the analysis of providing different modalities`
>
> We designed Orak to focus on the long-horizon decision-making ability of LLM agents interacting with the environment, when provided with pre-extracted core information. However, as shown in Table 3 of the main paper, most LLM agents often fail to complete tasks even under such *clean* text states. When *noisy* states are given, the benchmark may be biased to the LLM's information retrieval ability, rather than its long-horizon interaction or decision-making capacity. Moreover, since environment providers are usually able to adjust the noiseness of states, evaluating LLM agents under clean, well-structured conditions may have more practical use-cases.
>
> Meanwhile, to address your concern about potential bias toward clean text states in the modality analysis, we conducted an **additional experiment**. Specifically, for Street Fighter III, we **provided noisy text states along with image states to VLMs** to compare the results with the original clean-text setting. We additionally provided all the background objects and their bbox locations in the text state.
>
> | Model | Modality | Clean State | Noisy State |
> | - | - | - | - |
> | Qwen2.5-VL-32B | text-only | **32.0** | 26.0 |
> | | both | 29.7 | **40.0** |
> | gpt-4o | text-only | **29.7** | 14.0 |
> | | both | **27.1** | 10.0 |
>
> As shown in the table, when only noisy text states were provided, the performance of both VLMs decreased. In contrast, when both noisy states and images were provided, the effect was mixed: Qwen2.5-VL-32B showed an improvement, whereas GPT-4o exhibited a performance drop. We will incorporate this noisy-state provision as an *optional feature* in Orak to enable more unbiased and modality-robust studies. Note that, **extending Orak to provide noisy states is straightforward**, since incorporating less-important background information requires only minimal parsing.
>
> `W3. The presented ablation for agentic modules is interesting as a high-level overview, but its results as presented are not well discussed`
>
> We appreciate your interest in our agent ablation studies. As you noticed, the value of Orak lies in **enabling** the study of various agentic modules in complex, game-like interactive tasks. Due to space limitations, we were only able to include a high-level overview of ablation results across the 12 games in the main section. However, **Appendices C–N provide more detailed analyses of the agentic modules for each game**. We added a mark on the main section as '*See Appendices C-N for more detailed ablation analysis for each game*' in Line 415 of the revised manuscript.

---

> ### Author Response · Authors · 2025-11-21
> **Official Comment by Authors (2/3)**
>
> `W4. The LLM finetuning experiment and resutls are not in-depth. The paper would benefit from a more detailed analysis of dataset quality and what/how it actually contributes during training, as well as any possible insights into data scalling`
>
> Following your suggestion, we **conducted additional experiments** for more detailed analysis of **dataset quality and scaling**. Specifically, we fine-tuned Llama-3.2-3B on low-scoring trajectories generated by gpt-4o (same amount as the high-score dataset), and a mixed dataset that combines the low-score and high-score datasets (doubling the data volume). We follow the fine-tuning configurations in Appendix P.2, and report the intra-game generalization performance.
>
> | Data | SF3 | DarkestD | StarCraft2 | SlaySpire | BabaIsYou |
> | - | - | - | - | - | - |
> | No fine-tuning | 12.0 | 87.2 | **0.0** | 0.0 | **20.0** |
> | high-score (110K) | **40.0** | **92.0** | **0.0** | **10.7** | **20.0** |
> | low-score (110K) | 18.0 | 40.0 | **0.0** | 8.7 | **20.0** |
> | mixed (220K) | 18.0 | 40.0 | **0.0** | **10.7** | **20.0** |
>
> The results show that the high-score dataset alone was the most effective, while the low-score dataset contributed little or even degraded performance in certain cases (e.g., Darkest Dungeon). Moreover, simply scaling up the dataset by mixing high- and low-score samples did not yield a clear improvement, indicating that data quality, rather than sheer quantity, plays a more critical role in achieving better generalization. We **added these results in Section 5.6** of the revised manuscript.
>
> `W5. lack of functionality to properly evaluate grounding of actions, as each game action space has been manualy defined. The platform could be easily modified to offer a full pixel-to-keyboard/mouse-commands interface that fully exposes complexity and allow uses to benchmark at different levels`
>
> Thank you very much for the constructive comment. As you mentioned, it is **easy to modify Orak's high-level mapped functions to lower-level ones**, since mostly the implementation of high-level mapping is the composition of low-level controls. To address your concern, for Street Fighter III, we conducted **additional experiments** by offering a **full keyboard-commands action interface**.
>
> | Model | High-level Action (Original) | Low-level Keyboard Action |
> | - | - | - |
> | LLama-3.2-3B | 13.3 | 0.0 |
> | gpt-4o | 29.7 | 18.0 |
>
> With the low-level keyboard action interface, two LLM agents showed a **significant performance drop**, and Llama-3.2-3B completely failed to perform the task, achieving a score of zero. Therefore, at the current stage, we believe that using a **high-level interface is more suitable for measuring and ranking** the long-horizon decision-making abilities of **LLM agents with different model sizes**.
>
> Practitioners can easily modify Orak's mapped functions to conduct complexity studies on different levels of control. However, as shown in Table 3 of the main paper, since current LLM agents still often struggle with high-level actions in long-horizon interactive tasks, we leave a thorough investigation of low-level extensions as future work.
>
> `Q1. I don't think the games industry itself is the main beneficiary of such benchmarking effort. How does this motivation and focus affect the benchmark design? Agent autonomy using games as learning environment could have much wider impacts and would need to be analysed differently`
>
> This is an excellent question. We agree that Orak can have much wider impacts beyond the game industry, advancing research toward autonomous agents across multiple domains, as games often contain high-level abstractions of real-world interactive decision-making processes. In particular, we expect that LLM/VLM agents trained on Orak would have a significant impact on the **system-2 reasoning** for Vision-Language-Action (VLA)-like models [a, b]. We have **added a brief note regarding impact** in the Introduction (Lines 35-37), and **included further discussions on how to enable RL fine-tuning LLM/VLM agents with Orak**, with specific reward design and dynamic data extraction, in Appendix T of the revised manuscript. Thanks again for helping us improve our paper.
>
> ---
> [a] From System 1 to System 2: A Survey of Reasoning Large Language Models, ArXiv, 2025
>
> [b] GR00T N1: An Open Foundation Model for Generalist Humanoid Robots, ArXiv, 2025
>
> `Q2. The paper mentions previous evaluation "often rely on visual inputs" as a criticism. Why? I'd argue that understanding of visual observations is exactly the most important area where games can help as benchmark for foudnaiton model capabilties`
>
> Thanks for your careful comment. We intended to say that most existing benchmarks do *not* offer a dimension to study LLM agents' abilities on games, since they only provide the *vision-only* state interface. We **removed** that sentence in the revised manuscript. Appologizes for the confusion.

---

> ### Author Response · Authors · 2025-11-21
> **Official Comment by Authors (3/3)**
>
> `Q3. In the Arena setting results, why exactly was Starcraft evaluated with one less model than Street Fighter?`
>
> At the time of submission, we were unable to include the Arena results of Gemini-2.5-Pro due to an API inference slowdown. We have now **added its results in Figure 4(b) of the revised manuscript**, same as the table below.
>
> | Models | Elo | Rank |
> | - | - | - |
> | Llama-3.2-3B | 1429.9 | 8 |
> | Qwen-2.5-7B | 1445.2 | 7 |
> | Minitron-8B | 1536.8 | 2 |
> | gpt-4o-mini | 1500.0 | 6 |
> | gpt-4o | 1525.0 | 4 |
> | o3-mini | 1500.0 | 5 |
> | **Gemini-2.5-pro** | **1512.3** | **3** |
> | Claude-3.7 | 1561.4 | 1 |
>
> `Q4. do you have any insights on why Minitron-8B performs so much better than in full results?`
>
> Minitron-8B beats most LLM agents in Street Fighter III (21 wins out of 22 matches). Upon analyzing the gameplay logs, we found that Minitron-8B's superiority  primarily stems from its frequent use of super attacks. Specifically, Minitron-8B used super attacks in 16.0% of turns, compared to 1.2% for GPT-4o and 4.1% for Claude-3.7. This aggressive skill utilization provides a direct shortcut to victory, suggesting that when multiple agents interact, their performance can be highly sensitive to adversarial or exploitative actions.
>
>
> `Q5. Table 3 shows the results only of using the "auto extracted state in text only form", correct? It would be interesting to have an easy comparison of that vs. best resutls somewhere. Even if in an appendix`
>
> Following your suggestion, to ensure the easy comparison across agent designs and modalities, we **included the overall performance table** in Appendix R of the revised manuscript, and highlight the best results for each game.
>
> `Q6. do you have any insights on why there is no SFT benefit intra-game for Startcrat and Baba?`
>
> StarCraft II and Baba Is You require complex strategic/spatial reasoning (to defeat competitive enemies) or precise action execution (moving tiles to create winning conditions and reach the flag), respectively. However, due to the limited capacity of pre-trained backbones (Llama-3.2-1B/3B), they failed to defeat enemies or complete different sub-quests in **new maps**, even after fine-tuning. We **enriched this discussion** in the revised manuscript (see Lines 472-473).
>
> `Q7. Typos. "In game industry" -> "In the games industry"`
>
> Thanks for the careful comment. We corrected the typo in the revised manuscript (Line 34).

---

### Author Response · Authors · 2025-12-03
**General Response**

We sincerely appreciate all reviewers for their constructive feedback. We would like to summarize the rebuttal and the discussion trajectories with the reviewers.

### 1. Overall Summary

Most reviewers agreed that (1) the benchmark provides **broad game coverage and impact** (all reviewers), (2) the **results with modular studies are comprehensive and reproducible** (all reviewers), (3) the **fine-tuning dataset is useful** for agent development (all reviewers), and (4) the **presentation including agent ability taxonomy is clear** (mcXu, Tpyv, 5244).

Since reviewers' main concerns were about the benchmark design choices and extensibility, we have addressed them by providing additional experiments on (1) real-time gameplay performance, (2) fine-tuning data quality/scale/source studies, (3) robustness under game state noisiness, (4) effect of action granularity, and (5) prompt/temperature sensitivity.

We believe that these new analyses fully clarify the reviewers' remaining concerns, and thus significantly strengthened the benchmark’s impact and rigor.


### 2. Reviewer Discussion Summary

**Reviewer mcXu (Initial score 8 → No additional comments)**

- (W1–W3) Requested deeper analysis → We enriched studies on state noisiness, modality, and agent ablations.
- (W4) Requested fine-tuning data scale/quality analysis → We added data scale/quality experiments and discussion.
- (W5) Asked about action fine-grainedness → We added low-level keyboard-action experiments.
- (Q1) Suggested broader impact beyond games → We enriched the impact discussion in the Introduction and Appendix.


**Reviewer Cdp1 (Initial score 4 with willingness to raise → No additional comments)**

- (W1, W4, Q1–Q2) Concerned about benchmark novelty → We clarified Orak’s unique challenges and contributions as a game benchmark.
- (W3) Asked about game selection details → We clarified our selection criteria based on *Popularity, Complexity/Diversity, Transparency*.


**Reviewer Tpyv (Initial score 2 → Raised to 6)**

- (W1) Questioned advantages over prior benchmarks → We clarified our benchmark differentiation.
- (W2, Q1) Requested robustness evidence in results → We provided confidence intervals and t-tests.
- (W3, Q2) Asked about prompt/temperature effects → We added prompt/temperature studies showing no significant differences (via t-tests).
- (W4, Q3) Requested real-time evaluation → We extended StarCraft II to real-time gameplay.
- (W5, Q5) Asked about fine-tuning bias → We added bias analysis with R1-trajectory fine-tuning set.

Afterward, the reviewer noted that the concerns on *Statistical Reliability*, *Prompt/Temperature Robustness*, *Real-Time Evaluation*, *Fine-Tuning Data Bias*, and *Benchmark Contribution* were **fully resolved**, leading to a **score increase (2 → 6)**.


**Reviewer 5244 (Initial score 8 → Maintained 8 with Spotlight Recommendation)**

- (W1) Requested real-time evaluation → We provided real-time StarCraft II results.
- (W2) Requested RL fine-tuning discussion → We added RL fine-tuning design in Appendix T.
- (W3) Asked to further reduce the resource huddle → We explained plans for a size-constrained LLM eval track.

Afterward, the reviewer expressed satisfaction, especially with the real-time evaluation, and **recommended the paper for Spotlight**.

---

### Meta-Review · Area_Chair_3mAH · 2025-12-30

**Summary:**

This paper introduces Orak, a benchmark and evaluation framework for assessing the comprehensive capabilities of LLMs in long-horizon, interactive game environments. The work aims to move beyond narrow skill benchmarks by studying perception, planning, agentic strategies, and fine-tuning in a unified setting.

**Summary of Reviews:**
Some reviewers raised substantial concerns in the initial submission, primarily around (i) overstated claims of depth and novelty, (ii) abstraction-heavy benchmark design (clean textual states and high-level actions), and (iii) insufficient experimental rigor, particularly in fine-tuning analysis, statistical reporting, and interpretation of results (e.g., modality comparisons).

**Response and Revisions:**
The authors’ rebuttal and revision are thorough and responsive. First, they tempered the paper’s claims, aligning the framing with what is empirically demonstrated. Second, they substantially expanded the experimental section, adding: (a) noisy-state and vision-based variants to clarify the role of perception, (b) low-level action experiments to justify the choice of high-level APIs, (c) significantly deeper fine-tuning analyses covering data quality, scaling, and generator bias, (d) prompt and temperature sensitivity studies, (e) proper statistical reporting with confidence intervals and significance testing, and (f) real-time (latency-constrained) evaluations that meaningfully increase realism. These additions directly address the majority of the reviewers’ technical concerns and strengthen the paper.

Thus, I recommend acceptance.

**Reviewer Concerns:**

As above.

**Reviewer Scores:**

Reviewer Tpyv's concerns are largely addressed by the response; the reviewer has already changed the score to 6.

---

### Decision · Program_Chairs · 2026-01-26

Accept (Poster)